# Atmospheric methane control mechanisms during the early Holocene

Ji-Woong Yang[1], Jinho Ahn[1], Edward J. Brook[2], and Yeongjun Ryu[1]

[1]School of Earth and Environmental Sciences, Seoul National University, Seoul 08826, South Korea

[2]College of Earth, Ocean, and Atmospheric Sciences, Oregon State University, Corvallis, OR 97331, USA

*Correspondence to*: Jinho Ahn (jinhoahn@snu.ac.kr)

**Abstract.** Understanding processes controlling the atmospheric methane ($CH_4$) mixing ratio is crucial to predict and mitigate future climate changes in this gas. Despite recent detailed studies of the last ~1000 to 2000 years, the mechanisms that control atmospheric $CH_4$ still remain unclear, partly because the late Holocene $CH_4$ budget may be comprised of both natural and anthropogenic emissions. In contrast, the early Holocene was a period when human influence was substantially smaller, allowing us to elucidate more clearly the natural controls under interglacial conditions more clearly. Here we present new high resolution $CH_4$ records from Siple Dome, Antarctica, covering from 11.6 to 7.7 thousands of years before 1950 AD (ka). We observe four local $CH_4$ minima on a roughly 1000-year spacing, which correspond to cool periods in Greenland. We hypothesize that the cooling in Greenland forced the Intertropical Convergence Zone (ITCZ) to migrate southward, reducing rainfall in northern tropical wetlands. The inter-polar difference (IPD) of $CH_4$ shows a gradual increase from the onset of the Holocene to ~9.5 ka, which implies growth of boreal source strength following the climate warming in the northern extratropics during that period.

## 1 Introduction

Methane ($CH_4$) is a potent greenhouse gas whose atmospheric mixing ratio has increased more than 2.5 times since the Industrial Revolution (Dlugokencky et al., 2009). Although lower in abundance compared to carbon dioxide ($CO_2$), $CH_4$ has ~28 times higher global warming potential (GWP) on a 100 year time scale and even higher GWP on shorter time scales due to its short atmospheric lifetime (Stocker et al., 2013). Hence understanding the controls on atmospheric $CH_4$ is important to predict and mitigate future climate and environmental changes.

Naturally, $CH_4$ is mainly produced from microbial decomposition by methanogens in anaerobic environments, such as waterlogged soil, wetlands, or sediments of lakes and rivers. Even though a part of $CH_4$ is oxidized, and can be emitted in the form of $CO_2$, a considerable amount of $CH_4$ is still released into the atmosphere through vascular plants, diffusion and ebullition processes (e.g., Joabsson and Christensen, 2001). Other, more minor sources include geological $CH_4$ released from mud volcanoes and gas seepages through faults (e.g., Etiope et al., 2008 and references therein), pyrogenic sources such as wildfire and biomass burning (Andreae and Merlet, 2001; Ferretti et al., 2005; Hao and Ward, 1993), and microbial digestion by wild animals and termites (e.g., Sanderson, 1996). The $CH_4$ flux from the ocean to the atmosphere is considered as too small to create a significant change in global budget compared to the other sources (e.g., Rhee et al., 2009). The major sink of atmospheric $CH_4$ is

photochemical reactions (oxidation) with the hydroxyl radical (OH), which is mainly controlled by atmospheric
temperature, humidity, and the mixing ratio of $CH_4$ itself and non-methane volatile organic compound (NMVOC)
(e.g., Levine et al., 2011 and references therein). Air temperature affects humidity thereby limiting the production
of OH. Oxidation of both NMVOCs and $CH_4$ compete for OH, that is, an increase in NMVOC emission reduces
the available OH, and increases the atmospheric lifetime of $CH_4$ (Valdes et al., 2005). Further, since the OH is
produced by photo-dissociation reaction, the $CH_4$ sink strength is affected by light availability and tropospheric
ozone (e.g., Levy, 1971). However, recent model studies suggested that $CH_4$ changes between glacial- and
interglacial conditions were driven mostly by source changes, rather than sink changes (Weber et al., 2010; Levine
et al., 2011).
Polar firn and ice are the unique archives that preserves the ancient atmosphere for the research of fossil air
older than the 20[th] century. Paleoatmospheric $CH_4$ levels have been reconstructed for the last 800 ka from
Antarctic- and Greenland ice cores (Loulergue et al., 2008). Given the relatively long lifetime in troposphere (11.2
± 1.3 years at present, e.g., Prather et al., 2012) compared to atmospheric mixing time, ice core $CH_4$ records
represent well-mixed global signatures. The 800 ka record, shows that past $CH_4$ change generally followed
glacial-interglacial cycles, with low concentrations during glacial periods and high concentrations in interglacials,
as well as the shorter orbital cycles of obliquity and precession (e.g., Spahni et al., 2005; Loulergue et al., 2008).
Those earlier studies suggested that the changes in climate and hydrology in the tropics induced by orbital forcing
controlled $CH_4$ emissions. The resemblance between water stable isotope records from Greenland ice cores, a
proxy for Greenland temperature change, and global $CH_4$ mixing ratios on millennial time scales is also well
known. This implies that local temperature change around Greenland is linked to the major $CH_4$ sources in low
latitudes (e.g., Brook et al., 1996; Chappellaz et al., 1993; Huber et al., 2006; EPICA Community Members, 2006;
Grachev et al., 2007, 2009).
Intensive precipitation changes in the low latitude summer monsoon regions, caused by insolation changes (e.g.,
Asian monsoon) have been suggested as an important $CH_4$ control during the glacial period (e.g., Chappellaz et
al., 1990). From time series analysis of past $CH_4$ records, Guo et al. (2012) found that the tropical monsoon
circulation is a primary control of relatively shorter (millennial) time scale variability, while long-term (multi-
millennial to orbital scale) variations are dominated by solar insolation changes. It has been found that tropical
monsoon activity is closely related to orbital-scale $CH_4$ change (e.g., Brook et al., 1996; Chappellaz et al., 1990),
especially Asian monsoon (e.g., Loulergue et al., 2008) and South American monsoon (e.g., Cruz et al., 2005).
However, no direct correlation between $CH_4$ and tropical monsoon signals has been reported for the early
Holocene, although positive relationships between Greenland climate and tropical monsoon intensity (e.g., Chiang
et al., 2008), as well as between Greenland climate and $CH_4$ (e.g., Spahni et al., 2005; Wang et al., 2005; Mitchell
et al., 2011) have been reported.
The relationship between the latitudinal shift of the ITCZ and $CH_4$ emissions varies with time scales. Landais
et al. (2010) and Guo et al. (2012) suggested that ITCZ migration is not a dominant control of glacial-interglacial
$CH_4$ cycle because long-term $CH_4$ trend does not follow the precessional insolation change in the northern
hemisphere (NH) well. Modelling studies found the southward shift of the ITCZ coincides with reduced $CH_4$ in
Last Glacial Maximum (LGM) and Heinrich Stadial (HS) events, even though changes in wetland area and surface
hydrology were limited (Weber et al., 2010; Hopcroft et al., 2011). These authors instead suggested that changes
in temperature and/or plant productivity affected $CH_4$ production during those events. ITCZ migration does appear

to be related to millennial- or sub-millennial scale $CH_4$ change, however. Brook et al. (2000) found that sub-millennial $CH_4$ minima during the last deglaciation correspond with reduced precipitation recorded in Cariaco Basin sediment data, which indicates southward displacement of ITCZ (Hughen et al., 1996). This hypothesis is supported by spectral analysis of $CH_4$ during the past 800 ka record that found that ITCZ change becomes an important driver of millennial scale $CH_4$ change (Tzedakis et al., 2009; Guo et al., 2012).

For the Holocene, high-resolution $CH_4$ records from Law Dome and West Antarctic Ice Sheet (WAIS) Divide ice cores in Antarctica show characteristic variability on multi-decadal to centennial time scale during the late Holocene (MacFarling-Meure et al., 2006; Mitchell et al., 2011). The high-resolution records have been compared with various temperature- and precipitation proxies, but previous work found no strong correlations that explain the observed decadal- to centennial scale variabilities. This may be because the late Holocene $CH_4$ budget was comprised of both natural and anthropogenic terms, making it difficult to distinguish between them. Mitchell et al. (2011) pointed out that some of the abrupt $CH_4$ decreases could have had anthropogenic causes. Later, Mitchell et al. (2013) made simultaneous measurement of Antarctic (WAIS Divide) and Greenland (Greenland Ice Sheet Project 2; GISP2) ice cores to derive an IPD record, and extended their high-resolution records back to ~4 ka. They used eight-box atmospheric methane model (EBAMM) and anthropogenic- and natural emission scenarios to investigate $CH_4$ control factors. Their results showed that the late Holocene $CH_4$ evolution can be explained by a combination of natural- and anthropogenic emissions. In principle, stable isotope ratios of $CH_4$ help us to distinguish the types of sources – biogenic, pyrogenic, and geologic. Sowers (2010) reconstructed the $CH_4$ mixing ratio and stable isotopic composition ($\delta^{13}C$-$CH_4$ and $\delta D$-$CH_4$) throughout the entire Holocene. He suggested several possible control factors, such as boreal wetlands and thermokarst lakes, changing $C_3/C_4$ plant ratio of $CH_4$-emitting ecosystems, and changing composition of methanogenic communities. Previous studies have shown reduction of pyrogenic emission and increased agricultural emission during the last millennium (Ferretti et al., 2005; Mischler et al., 2009). In later work using $\delta^{13}C$-$CH_4$ records from North Greenland Eemian Ice Drilling (NEEM) ice core, Sapart et al. (2012) found that the centennial-scale variations during the last two millennia were caused by changes in pyrogenic- and biogenic emissions. Ruddiman et al. (2011) and Sapart et al. (2012) estimated $CH_4$ emission change due to anthropogenic land use changes, which shows a good agreement with the trends from ice core measurement. Since there is no high-resolution reconstruction of past population and land use area, and consequently large uncertainties of $CH_4$ emission from land use change impede identification of any shorter scale changes.

The early Holocene is a suitable period to study natural $CH_4$ controls under Holocene interglacial climate condition. Since there was only negligible human population and relevant $CH_4$-emitting anthropogenic activities (e.g., Goldewijk et al., 2010; Kaplan et al., 2011) during this time, the early Holocene $CH_4$ changes must have occurred mostly due to natural causes. Understanding natural controls could contribute to better constraints on human-induced $CH_4$ changes. However, high-resolution studies that covers the entire early Holocene have not been carried out extensively so far, except for studies of the prominent cooling event at 8200 years BP (Spahni et al., 2003; Kobashi et al., 2007; Ahn et al., 2014). Despite Rhodes et al. (2015) reported a very high-resolution record from WAIS Divide ice core that extends from the last glacial period to the earliest Holocene (~9.8 ka), the authors do not deal with the early Holocene $CH_4$ variability. Earlier studies mainly focused on long-term change, attributing the major control to low latitude hydrology based on regional climate records that show wetter climate in tropics during the early Holocene (Blunier et al., 1995; Brook et al., 2000; Chappellaz et al., 1993, 1997).

Therefore, in this study we present a new high-resolution $CH_4$ record from the early Holocene and investigate
natural control mechanisms under interglacial condition. It should be noted that environmental boundary
conditions of the early Holocene were not identical to those of the late Holocene. Global sea level rose throughout
the early Holocene while remnant ice sheets in North America disappeared.
**2 Materials and Methods**
In this study we used ice samples from the Siple Dome deep ice core (SDMA) drilled from 1997 to 1999 on the
Siple Coast, West Antarctica (81.65°S, 148.81°W; 621 m elevation) (Taylor et al., 2004). The SDMA samples
were collected and cut at National Ice Core Laboratory (NICL, Denver, Colorado, USA) from January to February
of 2013. The brittle zone of SDMA ice starts below 400 m and continues to the bottom of the core at 1004 m
(Gow and Meese, 2007) and samples from this region are more likely to be fractured. Hence, the samples were
carefully collected from unbroken subsections during sample preparation at NICL. The samples were packed in
insulated foam boxes with numerous eutectic gels, and shipped to South Korea via expedited airfreight.
Temperature loggers showed the temperatures were maintained below -25°C. The boxes were picked up directly
just after custom clearance at the airport and then the ice samples were stored in a walk-in freezer at Seoul National
University (SNU, Seoul, South Korea) that was maintained below -20°C. We measured 295 individual ice samples
from 156 depth intervals from 518.87 to 718.83 m, covering from 8.36 to 20.25 ka after synchronizing to the
Greenland Ice Core Chronology 2005 (GICC05, Rasmussen et al., 2006), of which 256 ice samples from 120
depth intervals from 518.87 to 623.38 m are used in this study. All samples were duplicated, so that our final $CH_4$
data were presented by averaging the results of duplicate analysis from the same depth. The analytical uncertainty
of our data set is estimated by the uncertainty of individual ice measurement divided by square root of 2 (see
below). We rejected data that show difference between duplicate measurements larger than 10 ppb, and 9 data
points were rejected in the studied period. The results of SNU measurement (111 points) are plotted in Figure 1.
The 16 samples from 8 depths were used for reproducibility check on different days (Table 1). The air occluded
in ice was extracted by a melting and refreeze process under vacuum. Ice samples were prepared in a walk-in
freezer in the morning of each experiment day, the outermost >2 mm was trimmed off to eliminate potential
contamination by ambient air during the storage. The samples were then moved to the laboratory and placed in
glass sample containers. The sample flasks were custom-made glass flasks welded to stainless steel flanges, and
attached to the vacuum line with copper gaskets. The sample flasks were partially submerged in a chilled ethanol
bath while being attached to the vacuum line. Flasks were evacuated for at least 40 minutes, then the ice samples
were melted by submerging the sample flasks in a warm water bath. Melting was usually completed within 30
minutes. The sample flasks were then submerged in the cold ethanol bath chilled to around -82°C for more than
an hour to refreeze. During refreezing, we carried out daily calibration of the gas chromatograph system, normally
taking ~90 minutes. The ethanol temperature normally rose to -55°C just after submerging the flasks, and
recovered to -65°C before expansion of the air in the flasks. The extracted air in the headspace was expanded into
the evacuated vacuum line and sample loop of a gas chromatograph (GC) equipped with a flame ionization
detector (FID) to measure $CH_4$ mixing ratio. After detecting the $CH_4$ peak in the GC chromatogram (retention
time of ~1.6 minutes), the vacuum line and sample loop is evacuated again prior to the next injection. The GC
linearity was tested by a series of inter-tank calibration using four working standard air cylinders (395.5, 721.3,

895.0, and 1384.9 ppb $CH_4$ on NOAA04 scale, Dlugokencky et al., 2005). A daily GC calibration curve was determined by measurements of a working standard having the closest $CH_4$ mixing ratio of expected value from the samples; in this study, we used the 721.3 ppb $CH_4$ standard for samples of the early Holocene. We calibrated with a standard air six times before and after sample measurements. The detailed configuration of the vacuum line and GC is described in another paper (Yang et al., in preparation).

Different solubilities of each air component cause preferential dissolution during melting procedure. As the solubility of $CH_4$ is higher than the other major components of air – nitrogen ($N_2$), oxygen ($O_2$), Argon (Ar), the $CH_4$ mole fraction of the extracted air is lower than originally enclosed air (solubility effect). The $CH_4$ mole fraction of air enclosed in ice sample is estimated from residual gas fraction and $CH_4$ mixing ratio in air remained in refrozen meltwater (retrapped air). Residual gas fraction is a measure of how much air is retrapped during refreeze, which is defined as ratio of amount (pressure) of air extracted from the 2nd gas extraction to the 1st extraction. The 2nd gas extraction was carried out using leftover refrozen meltwater samples after the 1st extraction finished. Mean residual gas fraction is $1.05 \pm 0.13\%$ ($1\sigma$, n = 60) for SDMA ice samples and $0.38 \pm 0.08\%$ ($1\sigma$, n = 40) for bubble-free ice. The test with ice samples from Styx glacier, Antarctica revealed that $CH_4$ mixing ratio in retrapped air is enriched 3.1 times (n = 12) for glacial ice and 3.0 times (n = 7) for bubble-free ice. Then the solubility effect is corrected by using a simple mass balance calculation.

Daily systematic offset correction was applied to account for the daily-varying system condition. To do this, we measured four bubble-free ice samples every day with SDMA ice samples. The experimental procedures for the bubble-free ice were identical to the SDMA ice. After the sample flasks are evacuated, standard air is injected into the flasks containing bubble-free ice, so that it returns similar air pressure to the typical size of SDMA ice when the extracted air inside the bubble-free ice flasks is expanded into the sample loop. The solubility correction for the bubble-free ice was done by the same formula as SDMA ice samples, but using different residual gas fraction. After corrected for solubility effect, the daily systematic offset is calculated by difference between $CH_4$ mixing ratio of the injected standard air and results from the four flasks containing bubble-free ice. The systematic offset ranges from 5 to 15 ppb during the SDMA measurement period. A daily offset is subtracted from the ice samples corrected for gas solubility effect. This is one of the major differences with OSU wet extraction system, where the systematic offset is interpolated from the results of blank tests carried out between several days (Mitchell et al., 2011).

The bubble-free ice was made by chilling the degassed ultrapure water (resistivity >18.2 MΩ·cm at 25°C) slowly from the bottom in a closed stainless steel chamber. From gas extraction test using our bubble-free ice without injecting standard air, we observed that no significant pressure increase at the pressure gauge with a detection limit of 0.01 Torr (corresponding to less than 0.03% of sample air pressure in the extraction line) after melting-refreezing the bubble-free ice. Mass dependent (gravitational) fractionation within the firn (Craig et al., 1988; Schwander, 1989) was corrected by using the nitrogen isotope ratio ($\delta^{15}N$) of atmospheric $N_2$ occluded in bubbles. Siple Dome $\delta^{15}N$ records show a mean enrichment of $0.23 \pm 0.01$‰ during the early Holocene (Severinghaus et al., 2009) and result in a slight decrease of $CH_4$ by $1.97 \pm 0.15$ ppb, which we applied to all of our measurements.

Here we consider two types of uncertainty sources: uncertainty in (1) estimating daily systematic offset and (2) other causes. The former indicates uncertainty of the daily systematic offset ($e1$). As the daily systematic offset is calculated from the mean of the four flasks with bubble-free ice and standard air, scattering of the bubble-free ice

samples can induce uncertainty in the systematic offset correction. The daily $e1$ is estimated with standard error of the mean (SEM, n=4), because the daily systematic offset is calculated from the mean of the four bubble-free ice samples. The average of daily e1 is 1.9 ppb. The latter ($e2$) includes uncertainty due to solubility correction and inhomogeneous distribution of $CH_4$. Given our solubility correction uses the mean value of residual gas fraction and the ratio at which $CH_4$ enriches in retrapped air, different solubility effect and/or inhomogeneous $CH_4$ distribution in individual ice causes offset between adjacent duplicate ice samples analysed on the same day. As the duplicates from same depths were measured on the same day, we estimated the $e2$ with pooled standard deviation (PSD) between duplicate measurements from entire depths, which yields 3.3 ppb. Taking the $e1$ and $e2$ into account together, the final uncertainty of individual measurement is given as 3.8 ppb by error propagation. The uncertainty for the mean of duplicate results is obtained by dividing the individual uncertainty by square root of 2, yielding 2.7 ppb. Further details on the correction method is found in our manuscript in preparation (Yang et al., in preparation).

We made additional measurements using adjacent samples (depth difference of 10 cm) at randomly selected 8 depth intervals to examine reproducibility and long-term stability of our system. The second measurements of duplicates were performed 8 to 80 days after the first analysis. Table 1 displays quadruplicate results at each depth. PSD between the mean of duplicate analyses of the first and second measurements on different days yields 1.1 ppb. The good agreement between duplicate means indicates good reproducibility of our system. In the meanwhile, PSD of the quadruplicate measurements is 3.0 ppb, which is similar to PSD of duplicate samples for the entire data set (3.3 ppb).

To check reliability of the record we compared our data set with previous SDMA measurements at Oregon State University (OSU) for 8.4 to 9.1 ka period when the two records overlap. The OSU $CH_4$ record was measured with a temporal resolution of 8 years with precision of 2.8 ppb (Mitchell et al., 2011; Ahn et al., 2014). The average offset between the two data sets is 0.1 ppb, which lies within analytical uncertainty range of data sets. Therefore, we created a composite record by using the OSU data for 499.49 – 537.20 m interval (7.6 to 9.0 ka) because mean temporal resolution of OSU data (~22 years) is lower than SNU data (~37 years) during this period (Fig. 1). Our new SDMA $CH_4$ composite data have mean temporal resolution of ~26 years. The WAIS Divide continuous $CH_4$ records show much higher resolution (~2 years), but does not cover the entire early Holocene period (Rhodes et al., 2015).

**3. Result and Discussion**

**3.1 Millennial scale variability**

We carried out spectral analysis of SDMA composite record using the REDFIT program (Schulze and Mudelsee, 2002). Moderate (over 90% significance level) spectral power was found at ~1340, 401, 309, and 96-year periods. Given the ~42 years of gas age distribution of SDMA (Ahn et al., 2014), it would not reliable to study centennial scale variability. Therefore, we smoothed the data by a 250-year running average to remove centennial- to multi-centennial scale components and then detrended by a high-pass filter with a cut off period of 1800 years to isolate millennial scale variability. For comparison, the same processing scheme was applied to WAIS Divide time series

and we observed that Siple Dome and WAIS Divide $CH_4$ anomalies share similar millennial scale variability,
confirming the reliability of both our data and observed millennial scale changes (Fig. 2).

3       The high-pass filtered $CH_4$ time series demonstrates millennial scale minima at ~8.2, 9.3, 10.2 and 10.9 ka,

which occurred with nearly 1000-year spacing. The REDFIT results for 7.6 to 11.2 ka interval that excludes PBO
shows moderate (80% significance level) powers at ~731 and 430 (860)-year periods. Each minimum is
accompanied by depletion of water stable isotope ratio ($\delta^{18}O_{ice}$) from North Greenland Ice Core Project (NGRIP)
ice core, which implies climate cooling in Greenland. A close relationship between $CH_4$ and Greenland $\delta^{18}O_{ice}$
has been previously reported in glacial-interglacial cycles and Dansgaard-Oeschger (DO) events during the last
glacial period (e.g., Brook et al., 1996, 2000; Blunier and Brook, 2001; Chappellaz et al., 1993, 2013; EPICA
Community Members, 2006). However, it has not been confirmed for interglacial climate conditions during the
Holocene. Mitchell et al. (2011) found no significant correlation with Greenland climate in multi-decadal scale
during the late pre-industrial Holocene (LPIH), possibly because LPIH $CH_4$ budget is also affected substantially
by anthropogenic emissions (e.g., Ferretti et al., 2005; Mischler et al., 2009; Mitchell et al., 2013; Sapart et al.,
2012). In contrast, we observe a significant positive correlation (r = 0.57, p = 0.06) between the millennial-scale
change of Siple Dome $CH_4$ and NGRIP $\delta^{18}O_{ice}$ during the early Holocene. The correlation coefficient between the
smoothed- and filtered time series of SDMA $CH_4$ (before synchronization to GICC05) and NGRIP $\delta^{18}O_{ice}$ was
calculated for the 7.8 - 11.5 ka by interpolating to the original ages of SDMA $CH_4$ composite, with a reduced
degree of freedom.
The gas chronology of SDMA was developed based on $CH_4$ and $\delta^{18}O$ of air ($\delta^{18}O_{atm}$) correlation (Severinghaus
et al., 2009). In this study, we improved the chronology by synchronization of the previous chronology to GICC05
age scale by setting 3 age tie-points with stable water isotope ($\delta^{18}O$) record from the North Greenland Ice Core
Project (NGRIP) ice cores during the abrupt climate change events of the Preboreal Oscillation (PBO) and the 8.2
ka event, given that both events have been proved to be synchronous with $CH_4$ change (Kobashi et al., 2007,
2008). Ages between tie-points were inferred by linear interpolation of the age offset of nearest tie-points, which
range from -114 to 28 years. After synchronizing to the GICC05 scale, the correlation coefficient between SDMA
$CH_4$ composite and the NGRIP $\delta^{18}O_{ice}$ increases to r = 0.74 (p < 0.01) It implies that natural $CH_4$ budget is closely
connected with Greenland climate on millennial timescales, even though this conclusion is less robust as there is
no age tie-points between the 8.2 ka episode and PBO (Fig. 3). The positive correlation implies that the natural
$CH_4$ budget is connected with Greenland climate on millennial timescales.
The uncertainty of the modified chronology was examined by comparing with a tentative age scale determined
by $CH_4$ correlation with NEEM $CH_4$ discrete measurement data. NEEM $CH_4$ data follow GICC05modelext-
NEEM-1 scale (Rasmussen et al., 2013). The detailed method for $CH_4$ correlation is described in Section 3.2. The
age difference between the two chronologies is plotted in Figure 4, showing the maximum age difference of 105
years. In addition, we include the maximum layer counting uncertainty of 99 years (Rasmussen et al., 2006) and
delta-age uncertainty of 30 years (Rasmussen et al., 2013) during the early Holocene. Therefore, error propagation
of the above three errors indicate that the maximum error of SDMA gas age used in this study is ~147 years.
According to atmospheric modelling studies, abrupt cooling in the North Atlantic regions can alter atmospheric
circulation and to cause southward migration of the mean latitudinal position of the ITCZ (e.g., Chiang and Bitz,

2005; Broccoli et al., 2006; Cvijanovic and Chiang, 2012). Climate proxies demonstrate the climatic teleconnection between northern North Atlantic and low latitude regions. Sediment reflectance record from Cariaco Basin shows increased rainfall and humidity – which is due to southward displacement of ITCZ – corresponding to the 8.2, 9.3, and 10.9 ka abrupt cooling event, as revealed in previous studies for the different time periods (Peterson et al., 2000; Haug et al., 2001; Fleitmann et al., 2007; Deplazes et al., 2013). The southward displacement of the ITCZ leads further weakening of Asian and Indian summer monsoons and probably reduces $CH_4$ emission from northern tropical wetlands. The $^{18}O$ enrichment in speleothems from Dongge Cave (China), Qunf Cave (Oman), and Hoti Cave (Oman, not shown, Neff et al., 2001) occurred at similar timing with abrupt cooling in Greenland at 8.2, 9.3, and 10.9 ka, which indicates the reduction of monsoonal rainfall in northern tropical wetlands. The speleothem records from Chinese and Oman caves seem to lag by ~100 – 200 years after the $CH_4$ change at ~9.3 ka, but this lies within chronological uncertainties of ~200 – 400 years at around ~9.0 ka (Dykoski et al., 2005; Fleitmann et al., 2007). Moreover, sediment Ba/Ca ratio from Gulf of Guinea demonstrates concurrent decrease of West African monsoon (Weldeab et al., 2007). In contrast, an inverse relationship is observed from the Eastern Brazilian speleothem data (Lapa Grande Cave, Strikis et al., 2011) that suggest an increase in precipitation at the time of abrupt $CH_4$ decreases. Rhodes et al. (2015) pointed out that strong southward migration of the ITCZ could induce an abrupt $CH_4$ increase from southern hemisphere (SH) during the HS 1, 2, 4, and 5 events. Sperlich et al. (2015) also suggested that a sharp $CH_4$ peak at Greenland Interstadial 21.2 (~85 ka) was caused by emission from Asian and Amazon wetlands. However, considering the orbital parameters that indicate maximum summer insolation in NH while minimum in SH during the early Holocene, it can be inferred that contribution of SH wetland emission was relatively weak and overcompensated by reduction of NH emission.

The possibility that the observed $CH_4$ minima were caused by reduction of northern extra-tropical sources is not supported by previous modelling studies. Zürcher et al. (2013) found that abrupt cooling in Greenland and northern high latitudes by large freshwater input to the North Atlantic causes boreal peatland $CH_4$ emission to decrease substantially, which can explain ~23% of abrupt $CH_4$ decrease (~80 ppb) during the 8.2 ka event. Given the meltwater pulses during the early Holocene before the 8.2 ka event were probably much weaker (Teller and Leverington, 2004) than that corresponding to the 8.2 ka event, we suggest that boreal emission change is not the major cause of the $CH_4$ local minima.

Previously, Björck et al. (2001) found that climate cooling in the northern Atlantic and Santa Barbara Basin occurred associated with a change in solar-forcing at ~10.3 ka. However, the proxy data in Figure 2 show no clear indication of southward migration of the ITCZ and changes in Asian, Indian, African, and South American summer monsoon intensity associated with the ~10.2 ka cooling and $CH_4$ decrease. (Fig. 2b-f). Furthermore, speleothem $\delta^{18}O$ records from Mawmluh Cave (not shown) show no weakening of the Indian monsoon (Berkelhammer et al., 2012), and there was no distinct change in $\Delta\varepsilon_{LAND}$ , a proxy of global terrestrial respiratory fractionation of atmospheric $O_2$ at this time, which is affected by low latitude surface hydrology (Severinghaus et al., 2009). These evidences suggest that precipitation and surface hydrology in the northern tropics may have not changed significantly during around the 10.2 ka. Instead, there are two small decreases at ~9.9 and ~10.6 ka as shown in Dongge cave deposit record (Fig. 2d), but it is difficult to tell, given dating uncertainties, if these events correlate with the 10.2 ka cooling. Although there appears to have been no strong change in low latitude hydrology at 10.2 ka, the amplitude of $CH_4$ decrease at 10.2 ka is similar order to the other millennial events. Given that no

clear reduction of the Asian, Indian, and African monsoon intensity is observed, it is possible that the $CH_4$ decrease at 10.2 ka was controlled by other processes, outside of the northern tropics.

Previous studies have suggested an important role of solar forcing during the Holocene (e.g., Björck et al., 2001; Bond et al., 1997, 2001). Bond et al. (1997) reported four large ice-rafted debris (IRD) drifts occurred at ~8.1, 9.4, 10.3 and 11.1 ka caused by surface cooling of North Atlantic Ocean. They found that the ocean surface cooling and the IRD events are closely related to cooling over the Greenland. Figure 2 shows that each IRD event (maxima in hematite stained grain) occurred concurrently with minima of NGRIP $\delta^{18}O_{ice}$ record within age uncertainty. We postulate that the Greenland cooling leads to southward shift of the ITCZ and in turn it changes wetland $CH_4$ emission in low latitudes. Bond et al. (2001) found that IRD maxima during the Holocene coincide with solar activity minima and suggested that solar forcing could affect the climate change around the North Atlantic Ocean (and Greenland), through amplification by changes in sea ice and/or deep water formation. A close interplay between solar activity and monsoon intensity has been observed in previous studies using the Chinese and Oman speleothem records during the Holocene (Neff et al., 2001; Wang et al., 2005; Gupta et al., 2005), even on multi-decadal time scales (Agnihotri et al., 2002). However, the forcing mechanism of solar activity on the North Atlantic and global climate is not well understood. Jiang et al. (2015) found positive correlations between North Atlantic SST and solar forcing inferred from paleo-proxies ($^{14}C$ and $^{10}Be$) for the last 4000 years, although the correlation disappears during the mid- and early Holocene. They hypothesized that climate sensitivity to solar forcing is high for cooler climate. The above evidence suggests that the early Holocene $CH_4$ minima may be linked to anomalies in solar activity, but future study is needed to make it more conclusive.

Meanwhile, a shift to an El Niño-like SST state was suggested as another mechanism that changes tropical rainfall patterns (Marchitto et al., 2010). According to modern atmospheric observations, El Niño conditions lead to drying conditions in low latitude wetlands in Africa, Asia, and the Americas (e.g., Dai and Wigley, 2000; Lyon and Barnston, 2005; Hodson et al., 2011), which reduces tropical $CH_4$ emissions. Thus, we could speculate that both the ITCZ migration and El Niño-like SST change affected the tropical surface hydrology and $CH_4$ emission. According to Holocene ENSO activity reconstructions by Moy et al. (2002), no ENSO event was recorded during the early Holocene until around 7 ka, except weak ENSO events during 10.4 – 10.1 ka, where we observe a $CH_4$ drop apparently unrelated to monsoon proxies. Mitchell et al. (2011) observed a significant positive correlation between $CH_4$ and Pacific Decadal Oscillation (PDO) variability during the late Holocene. It has been reported that PDO modulates the wet/dry impact of ENSO depending on phase relationship between ENSO and PDO (e.g., Wang et al., 2014 and references therein). Using a Holocene PDO reconstruction from sediment grain size analysis by Kirby et al. (2010) shows PDO-related drying intervals in North America during 9.5 – 9.1, 8.9 – 8.6, and 8.3 – 7.8 ka, which overlap the $CH_4$ minima at 8.2 and 9.3 ka present in this study.

**3.2 Inter-polar difference of $CH_4$ during the early Holocene**

We calculated the inter-polar difference (IPD) of $CH_4$ to trace the latitudinal source distribution change during the early Holocene. The currently available high-resolution $CH_4$ records covering the early Holocene are SDMA discrete (this study), WAIS Divide discrete (WAIS Divide project members, 2015), WAIS Divide continuous (Rhodes et al., 2015), NEEM discrete (Chappellaz et al., 2013) and NEEM continuous data (Chappellaz et al., 2013). Among the Antarctic records, we consider WAIS continuous records most reliable from ~9.9 to 11.5 ka

interval. For the rest of the studied period, SDMA discrete records are better constrained than WAIS discrete data, because SDMA records have better analytical precision, as well as comparison with OSU measurements reveals a minimal offset for the early Holocene interval. Before IPD calculation, WAIS continuous data were calibrated to SDMA data, given the discrete measurements generally have better accuracy than continuous ones. Regarding the Greenland side, we use NEEM discrete records because not only there are discrepancies between continuous- and discrete data in some intervals, but also because NEEM discrete records were measured by similar wet extraction technique at OSU (Chappellaz et al., 2013).

Precise synchronization is crucial for direct comparison between data sets which have high frequency variations. For synchronizing between Antarctic (Siple Dome and WAIS Divide continuous) and NEEM records, the NEEM $CH_4$ record (~11 years resolution on average) is chosen as reference. Synchronization was done by two steps: First, we made initial synchronization between the Antarctic and NEEM data by setting match points at the midpoint of abrupt $CH_4$ change, and then we linearly interpolated the age offset of each match point for the rest of data points. Then we applied a Monte Carlo simulation to find a maximum correlation. Both data sets were resampled every 30 years, and each point was randomly perturbed (assuming a normal distribution with 1 sigma of 30 years). By doing so 1000 different time series were created, and the set having a maximum correlation with NEEM data was chosen. Criteria for "best fit" is correlation coefficient of 0.8 with NEEM original age scale, so that a maximum correlation less than 0.8 was discarded. This procedure was repeated to make 20 sets of maximum correlation time series, and the mean ages of 20 replicate simulations were set to synchronized age scale. The uncertainty range of IPD was calculated from synchronization uncertainty and $CH_4$ data uncertainty. To estimate synchronization uncertainty, we created 20 IPDs from the 20 sets of maximum correlation time series, and the standard deviation of the 20 records was taken as synchronization uncertainty for each of the data points. The $CH_4$ data uncertainty was estimated with the stated uncertainty of each data set (4.3 ppb for NEEM discrete / 2.7 ppb for SDMA / 1.5 ppb for WAIS continuous, 1 sigma). To check the sensitivity of the uncertainties, we carried out Monte Carlo simulations. We produced 1000 different sets of IPD, which vary randomly with Gaussian propagation in their ages and $CH_4$ concentration uncertainties. Each IPD was annually interpolated and smoothed by a $1/1000$ year$^{-1}$ low-pass filter. The cutoff frequency of 1000 years was chosen to examine multi-centennial to millennial scale change, because the IPD calculation is very sensitive to high frequency variability of $CH_4$ records from both poles. To report 95% confidence interval, we multiplied the standard deviation by 1.96 and enveloped the IPD.

Figure 6 displays the IPDs calculated from various pairs of data set with 95% significant interval. The two IPD records derived from most reliable data sets are plotted in red (NEEM discrete – Siple Dome, IPD-1 hereafter) and green (NEEM discrete – WAIS continuous, IPD-2 hereafter). Both IPD-1 and IPD-2 show a long-term increase from 11.5 to 9.9 ka, which indicates that boreal source contribution enhanced. However, IPD-1 shows a sharper increase during the PBO followed by decrease until ~10.7 ka, and in the latter case both IPDs differ beyond 95% envelope (from 10.4 to 10.8 ka). Although these differences are significant, and are probably due to small errors in the time scale and absolute concentrations differences, for example, due to uncertainties in blank corrections or solubility corrections, or core quality, they do not affect our basic interpretation of the trends. Instead, we combined the two IPDs to resolve this. Given the IPD-2 is better constrained than IPD-1, we use IPD-2 curve from 9.9 to 11.5 ka interval and IPD-1 for the rest of the studied period (Fig. 6). The combined IPD shows ~13 ppb increase from 11.5 to 9.5 ka. It displays similar trend with the NH extratropical (30° - 90°N) temperature

reconstruction (Marcott et al., 2013) and the modelled $CH_4$ emission from boreal thermokarst lakes (Walter et al., 2014), indicating that NH extratropical source strength increased during this period.

To quantify the source strength of low- and high latitude sources, we employed a simple 3-box $CH_4$ source distribution model used in previous studies (Chappellaz et al., 1997; Brook et al., 2000). Briefly, the model contains 3 boxes; northern extra-tropical latitude (30°N – 90°N, N-box), tropical (30°S – 30°N, T-box), and southern extra-tropical latitude boxes (30°S – 90°S, S-box). $CH_4$ mixing ratios in 3 boxes (in Tg box$^{-1}$) were determined from $CH_4$ mixing ratio of Antarctica and Greenland. The mean $CH_4$ mole fraction of N-box (30°N – 90°N) is not identical to that of Greenland ice core record, given the latitudinal $CH_4$ distribution (e.g., Fung et al., 1991). To derive the N-box $CH_4$, we followed the assumption of Chappellaz et al. (1997), where the authors assumed that difference between Greenland and the mean N-box $CH_4$ is 7% of IPD. Hence here the N-box $CH_4$ is calculated by subtracting 7% of IPD from the Greenland mixing ratio. T-box mixing ratio is inferred by assuming that the S-box emission is constant of 15 Tg yr$^{-1}$ (Fung et al., 1991). Emission from each box (Tg yr$^{-1}$) is then estimated by using the mixing ratios of the boxes, lifetime of $CH_4$ in each box, and transport times among the boxes. Following Chappellaz et al. (1997), we assume the lifetime of 18.7, 8.1, and 26.8 years in N, T, and S-box, respectively, and transport time of 9 months. The modelled emission changes are plotted in Figure 8. The model results reveal that tropical sources decrease (accounting for the largest portion in $CH_4$ budget), while NH extratropical emissions increase. The T-box emission is reduced from ~118 Tg yr$^{-1}$ to ~109 Tg yr$^{-1}$, and the N-box source strength increases from ~60 Tg yr$^{-1}$ to ~71 Tg yr$^{-1}$ during the 11.5 – 9.5 ka interval (Fig. 8). The long-term decrease of tropical emission follows the NH summer insolation change. This covariation may reflect the insolation-driven changes in emissions on multi-millennial timescale (e.g., Loulergue et al., 2008; Guo et al., 2012). Also plotted in Figure 8 is the boreal source fraction, defined as ratio of N-box emission to total source emissions, showing 5% increase (from 31.5 to 36.5%) during the same interval. The box model results at 9.0, 9.5, and 11.5 ka time slices are summarised in Table 2.

Our results are supported by proxy-based temperature reconstructions that indicate a gradual warming in northern extratropical regions (30°N – 90°N) until ~9.6 ka, while tropical temperature remains stable (Marcott et al., 2013). The climate warming in northern high latitudes caused ice sheet retreat (e.g., Dyke, 2004) and may have enhanced $CH_4$ emission by forming new wetlands in permafrost regions (e.g., Gorham et al., 2007; Yu et al., 2013) and accelerating microbial decomposition of organic material (e.g., Christensen et al., 2004; Schuur et al., 2015). Thermokarst lakes created by thawing ice wedges and ground ice in Alaskan- and Siberian permafrost has been suggested as a source of $CH_4$ (e.g., Walter et al., 2006, 2007; Brosius et al., 2012). The modelled enhancement of NH extratropical emission of ~11 Tg yr$^{-1}$ is similar to the $CH_4$ release of 8.2 Tg yr$^{-1}$ from thermokarst lake thawing, which is estimated based on present-day observations (Walter et al., 2014). Since most thermokarst lakes are located in NH high latitude regions (e.g., Walter et al., 2006, 2014), it may support the box model results. Our results are consistent with previous findings based on $CH_4$ stable isotope analysis. Fischer et al. (2008) found that increase of boreal source contribution is required to explain the more depleted $\delta^{13}$C-$CH_4$ during Preboreal period than the Younger Dryas interval. Sowers (2010) extended the $CH_4$ isotopic ratio into the entire Holocene and showed a gradual decrease of $\delta^{13}$C-$CH_4$ by ~2‰ from 10.5 to 4 ka, which was attributed to progressive expansion of NH high latitude sources.

## 4. Conclusion and summary

We reconstructed a new high resolution $CH_4$ record during the early Holocene from Siple Dome ice core, Antarctica, to study millennial $CH_4$ variability and its natural controls under Holocene interglacial condition. The new Siple Dome record agrees well with previous records measured at OSU within analytical uncertainty, showing a mean difference of 0.1 ppb. By combining the two data sets, we present a SDMA $CH_4$ composite record covering from ~7.7 to 11.6 ka. We observed four millennial scale $CH_4$ minima having 10–20 ppb of amplitude with 300–400 years duration. It is found that these $CH_4$ minima were accompanied with Greenland cooling, changes in ITCZ position and reduced Asian and Indian monsoon intensities. The observed evidences suggest that low latitude hydro climate changes were closely related to millennial scale $CH_4$ minima. Further, this study presented the millennial scale change of IPD, which was calculated from high resolution discrete data set of NEEM and SDMA, and a continuous record of WAIS Divide. Here we reported that the IPD increased by ~13 ppb from the onset of the Holocene to ~9.5 ka following the temperature rise in NH extra-tropical regions. The three-box model demonstrates that NH extratropical emissions elevated by ~11 Tg yr$^{-1}$, while tropical emission was reduced by ~9 Tg yr$^{-1}$, resulting the increased contribution of the NH extra-tropical sources by ~5%.

*Acknowledgements.* Financial support was provided by the Basic Science Research Program through the National Research Foundation of Korea (NRF) (NRF-2015R1A2A2A01003888) and Korea Polar Research Institution (KOPRI) research grant (PD12010 and PE15010). This work was also supported by the US National Science Foundation Grant PLR 1043518. We appreciate all the efforts of sample cutting and shipping of the Siple Dome ice core by Brian Bencivengo, Richard Nunn, and Geoffrey Hargreaves of National Ice Core Laboratory, Denver, Colorado. We sincerely thank to Yoo-Hyeon Jin, Jinhwa Shin, and Hun-Gyu Lee for their laboratory assistance and helpful discussions. Thanks should go to Heejo Lee for her help in preparing English manuscript. We are grateful to Mark Twickler and the NICL Science Management Office for providing the Siple Dome ice core samples, the collection of which was supported by the US National Science Foundation.

## Data availability

The early Holocene Siple Dome $CH_4$ data will be available on NOAA Paleoclimatology database and PANGAEA data repository.

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

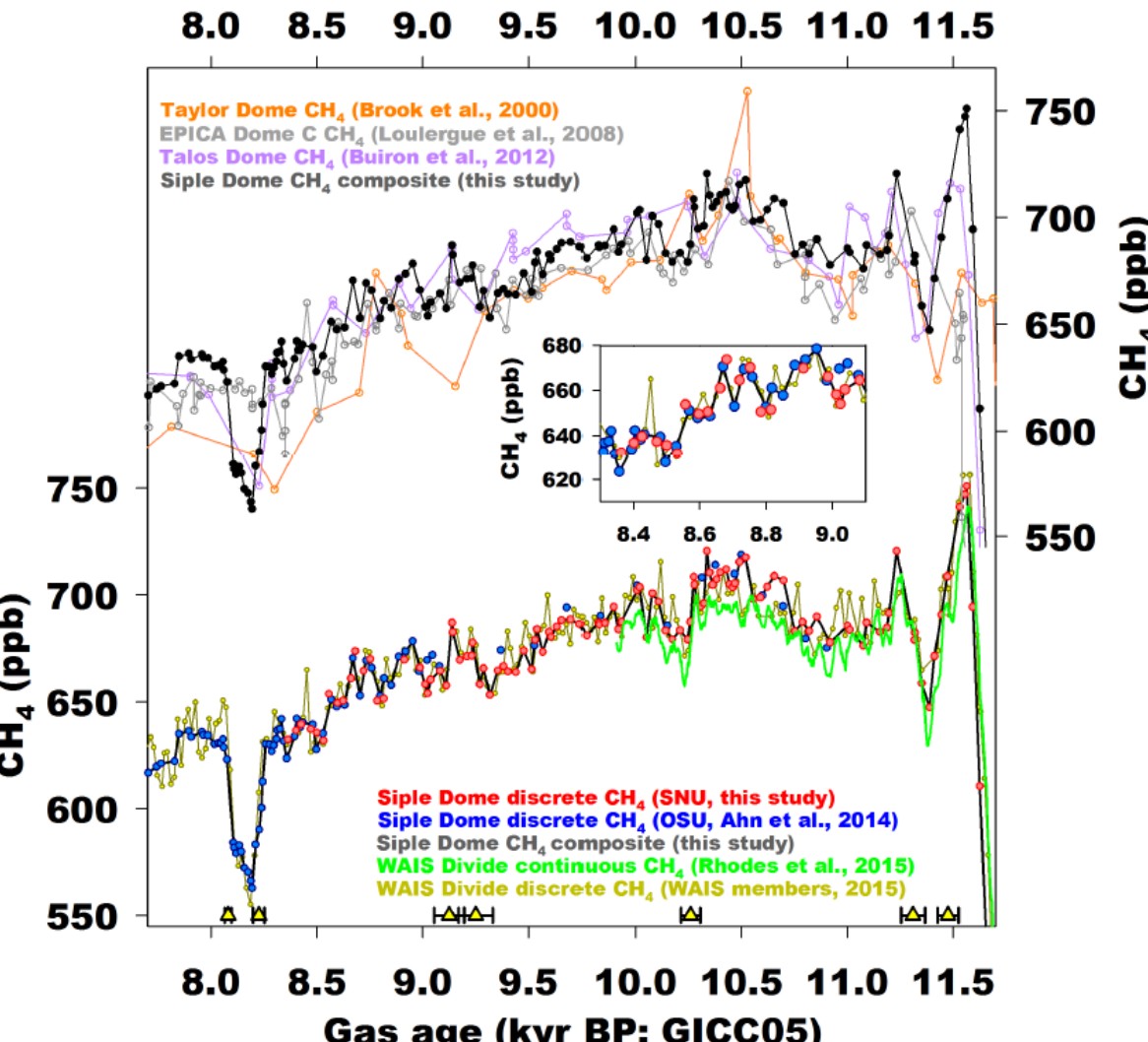

**Figure 1. Atmospheric CH₄ concentration reconstructions during the early Holocene. Top: new high-resolution Siple**
**Dome composite (black, this study and Ahn et al., 2014) compared with previous records from Taylor Dome (orange,**
**Brook et al., 2000), EPICA Dome C (grey, Loulergue et al., 2008), and Talos Dome (purple, Buiron et al., 2012). Bottom:**
**Siple Dome CH₄ records measured at OSU (blue, Ahn et al., 2014) and SNU (red, this study). Siple Dome composite**
**(black line) is plotted with WAIS Divide discrete (dark yellow, WAIS Divide project members, 2015) and continuous**
**measurement records (green, Rhodes et al., 2015). Inset: Enlarged plot showing overlapped interval between OSU and**
**SNU Siple Dome data.**

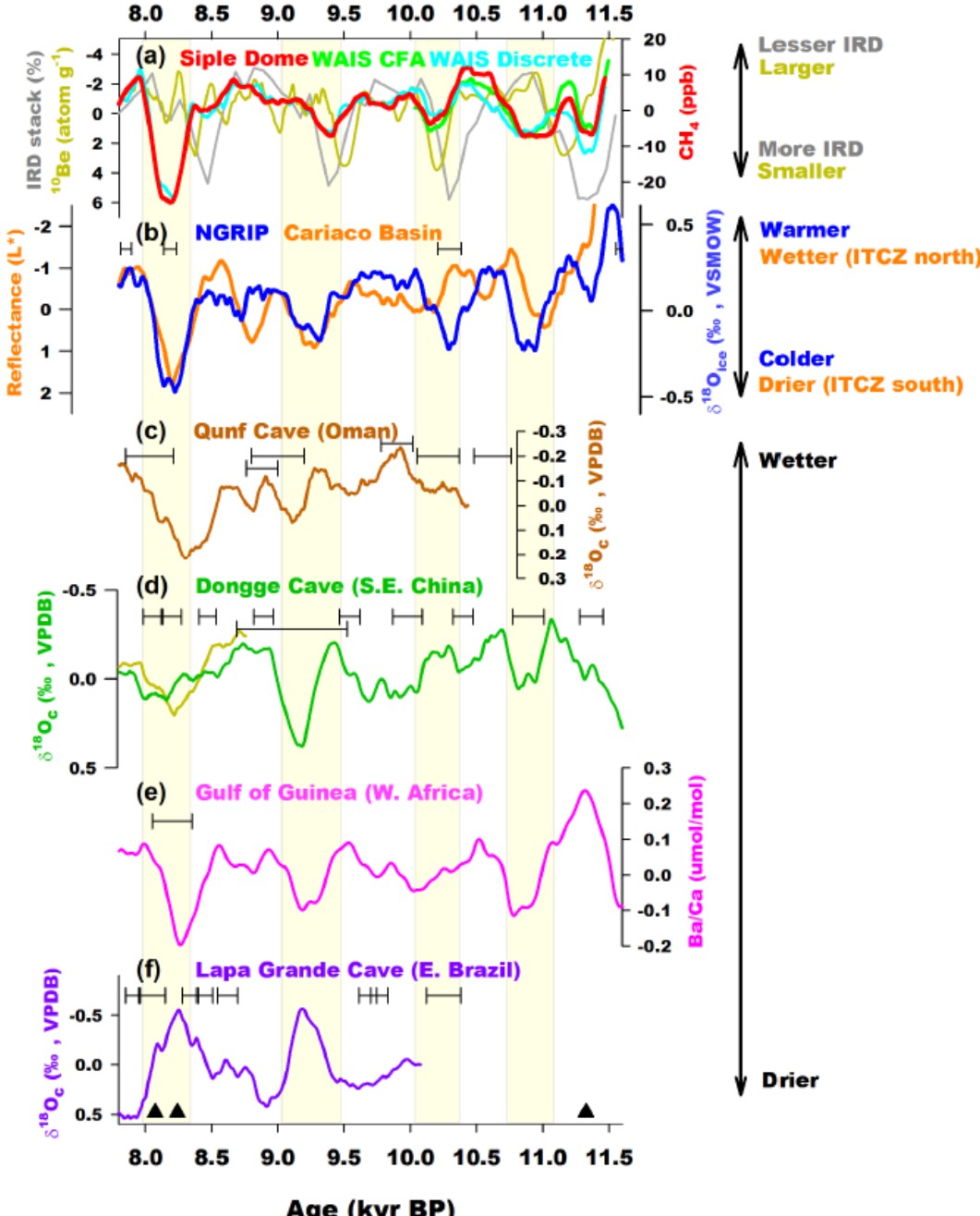

Figure 2. Millennial scale climate variability. All proxies presented here were smoothed by 250-year running average and detrended by high-pass filter with 1/1800-year window. (a) Siple Dome CH$_4$ (red, this study), Greenland $^{10}$Be (dark yellow, Finkel and Nishizumii, 1997), North Atlantic IRD stack (grey, Bond et al., 2001). Also shown are WAIS Divide CH$_4$ data by discrete (cyan, denoted "WAIS Discrete", WAIS Divide project members, 2015) and continuous (yellow green, denoted "WAIS CFA", Rhodes et al., 2015) technique. (b) NGRIP stable water isotope ratio (blue, Rasmussen et al., 2006) and Cariaco Basin reflectance (orange, Deplazes et al., 2013). (c) Qunf Cave speleothem oxygen isotope (Fleitmann et al., 2007). (d) Dongge Cave speleothem oxygen isotope (green, Dykoski et al., 2005; dark yellow, Wang et al., 2005). (e) Gulf of Guinea planktonic Ba/Ca ratio (Weldeab et al., 2007). (f) Lapa Grande Cave speleothem oxygen isotope (purple, Strikis et al., 2011). Black solid triangles are age tie-points used to adjust Siple Dome and WAIS Divide CH$_4$ data to GICC05 scale.

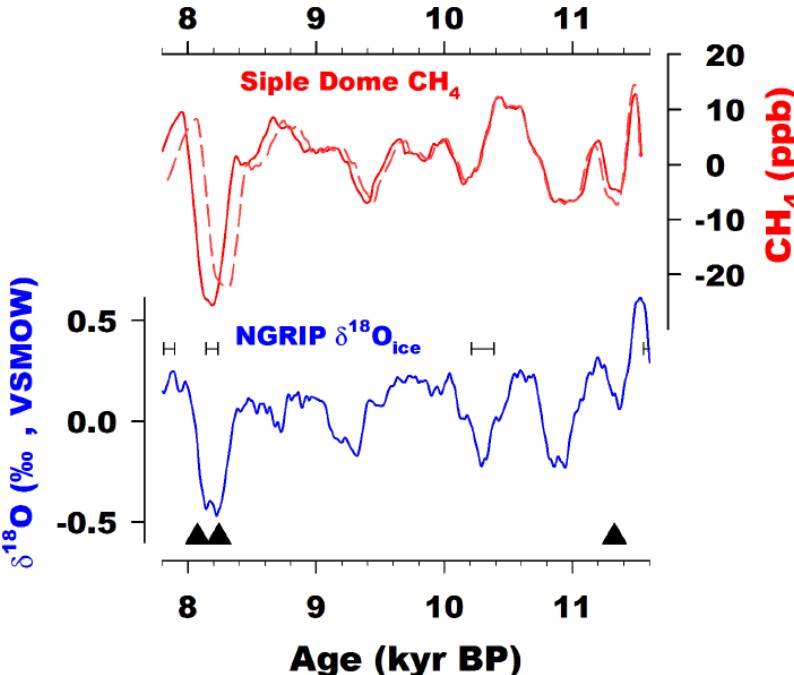

**Figure 3. Upper: Comparison between Siple Dome CH$_4$ anomalies plotted with gas age adjusted to GICC05 (red, solid)**
**and previous gas age (red, dashed; Brook et al., 2005). Lower: NGRIP $\delta^{18}O$ anomaly in GICC05 scale. The horizontal**
**error bars denote the age uncertainty of GICC05 chronology (Rasmussen et al., 2006), and the black triangles are age**
**tie points used to adjusting the Siple Dome age scale to GICC05 scale.**

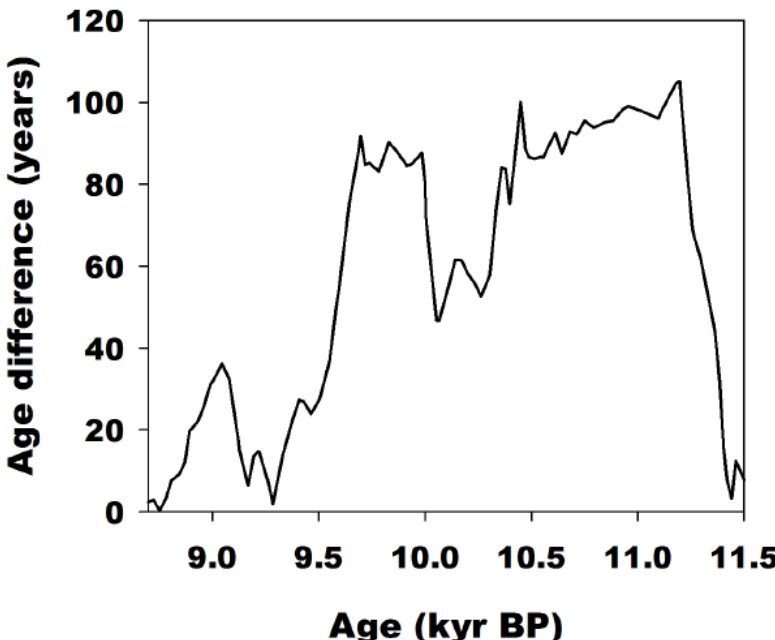

2    **Figure 4. Age difference between the new gas age scale adjusted to GICC05 by Monte Carlo matching with NEEM**

3    **discrete CH$_4$ (Chappellaz et al., 2013) and the original gas age based on CH$_4$ and $\delta^{18}O_{atm}$ correlation (Severinghaus et**

4    **al., 2009).**

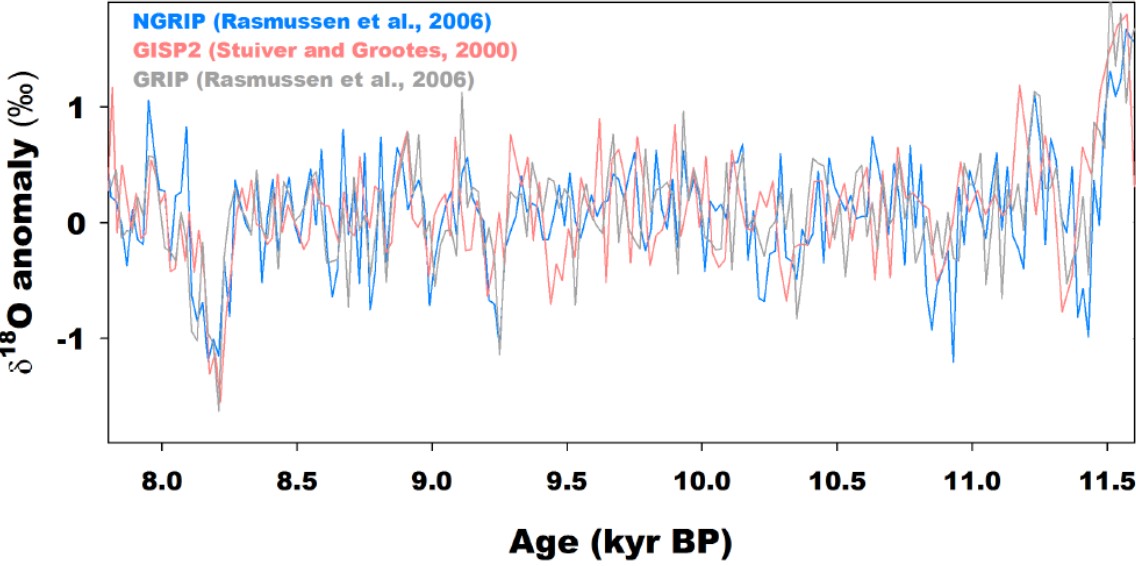

**Figure 5. Comparison of Greenland oxygen isotope ratios from NGRIP (blue, Rasmussen et al., 2006), GRIP (g**
**rey, Rasmussen et al., 2006) and GISP2 (red, Stuiver and Grootes, 2000). All time series were high-pass filtere**
**d with 1/1800-year window. Note that the cooling amplitude at 10.3 ka is smaller than 8.2 and 9.3 ka events in**
**NGRIP records, but this is not clear in GRIP and GISP2 ice cores.**

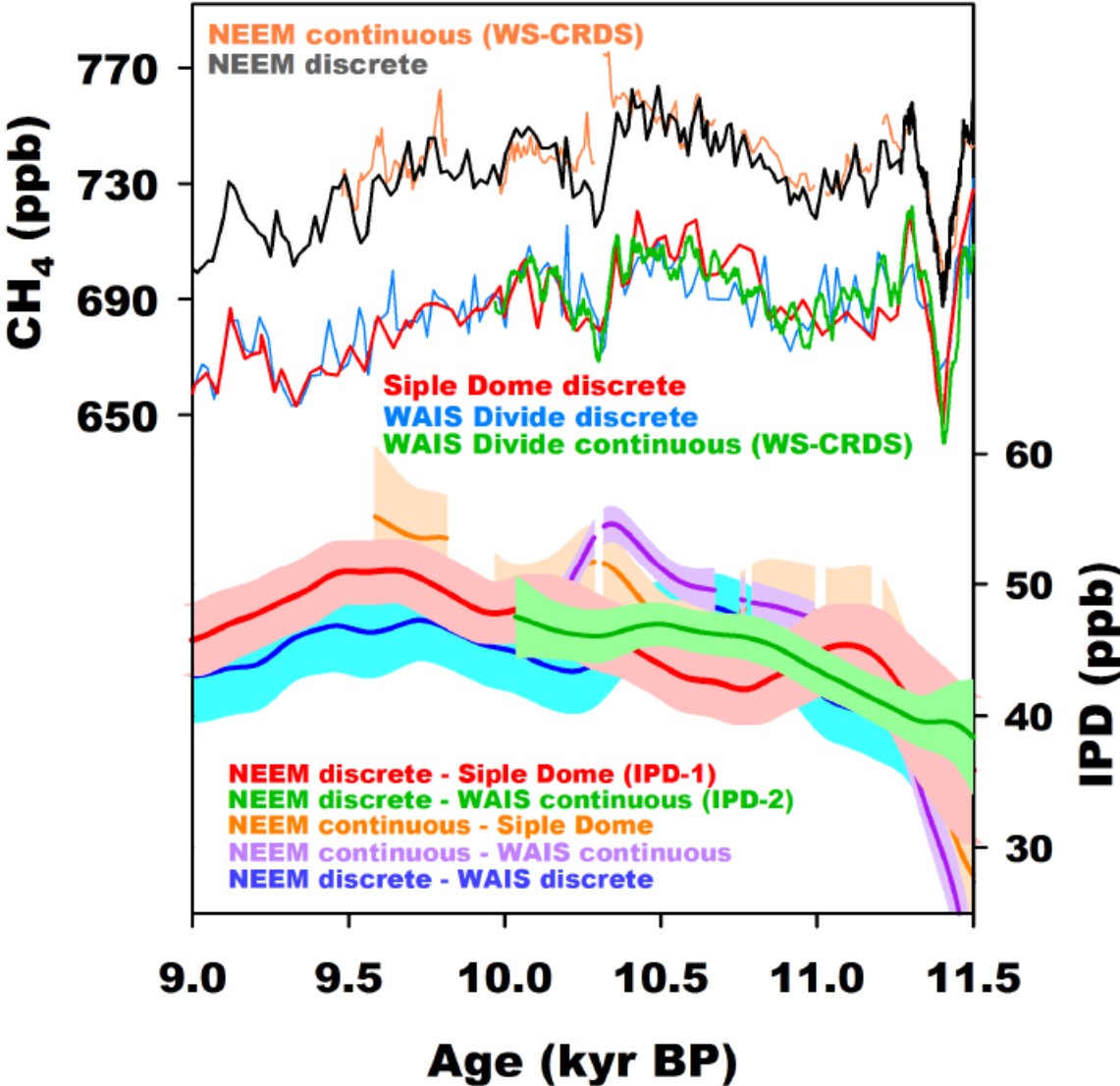

**Figure 6. CH₄ inter-polar difference (IPD) and high latitude CH₄ sources. Top: High-resolution CH₄ discrete**
**measurements from NEEM discrete (black, Chappellaz et al., 2013), NEEM continuous (orange, Chappellaz et al.,**
**2013), WAIS Divide discrete (light blue, WAIS Divide project members, 2015), WAIS Divide continuous (green, Rhodes**
**et al., 2015), and Siple Dome (red, this study) ice core records.    Bottom: 1000-year low-pass filtered IPD**
**reconstructions by using various pairs of Greenland- and Antarctic records,   in which the IPD-1 and IPD-2 are shown**
**in red and green, respectively. The shaded area indicate 95% significance interval.**

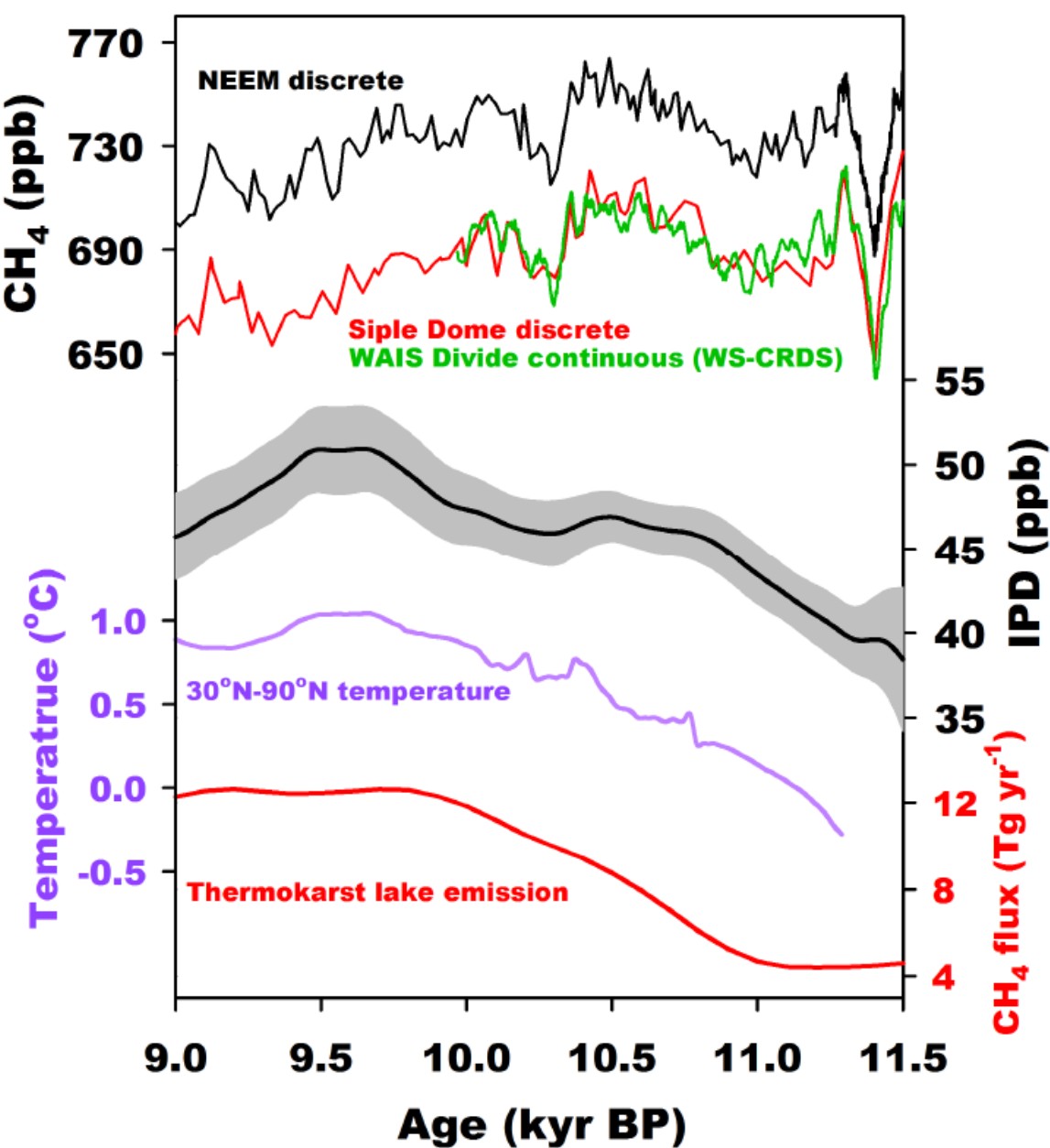

**Figure 7. CH₄ inter-polar difference (IPD) and high latitude CH₄ sources. Top: High-resolution CH₄ discrete**
**measurements from NEEM discrete (black, Chappellaz et al., 2013), WAIS Divide continuous (green, Rhodes et al.,**
**2015), and Siple Dome (red, this study) ice core records. Middle: 1000-year low-pass filtered combined IPD with 95%**
**significance interval (shaded). Bottom: Previous estimates are marked in green and orange (Brook et al., 2000;**
**Chappellaz et al., 2013). Proxy-based temperature reconstruction for 30°N-90°N (purple, Marcott et al., 2013). CH₄**
**flux estimate from Siberian- and Alaskan thermokarst lakes (red, Walter-Anthony et al., 2014).**

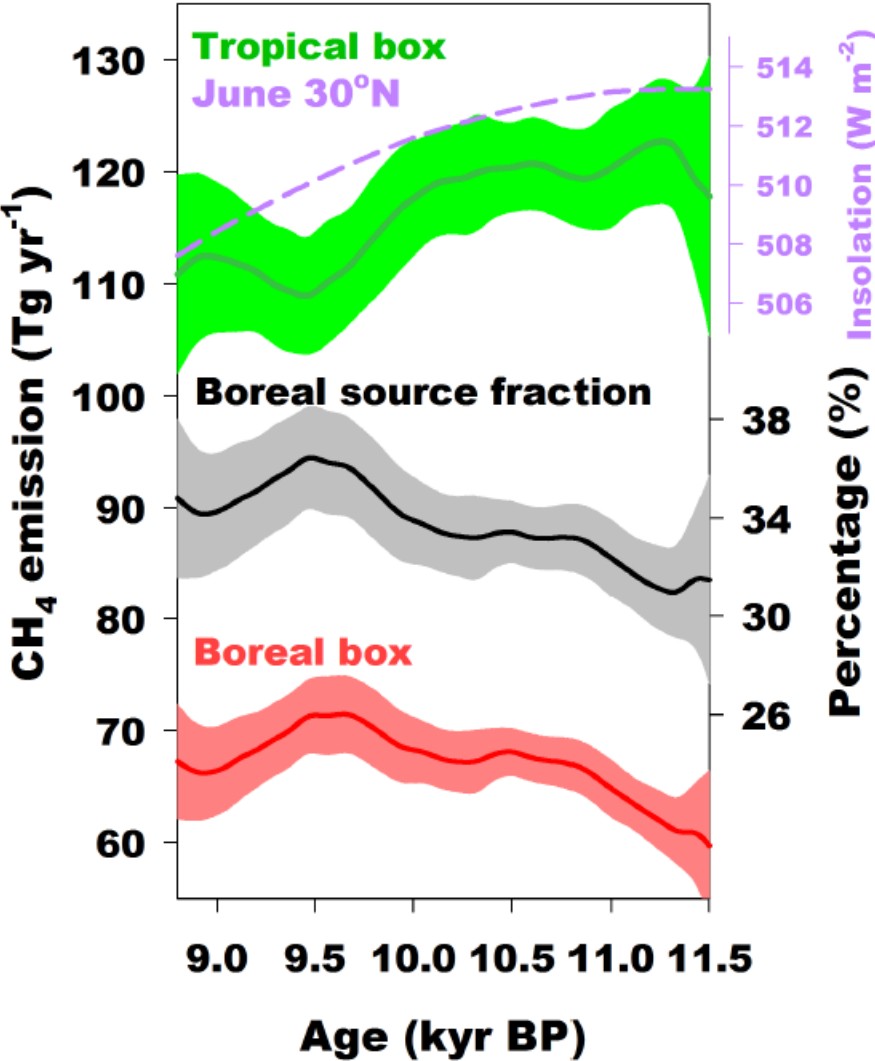

2 **Figure 8. 3-box source distribution model results of tropical (green) and boreal (red) boxes. Black line shows the boreal**

3 **to total source fraction (see text). Purple dashed line plotted with tropical emission is summer insolation in 30°N (Berger**

4 **and Loutre, 1991).**

1 **Table 1. Summary of results of replicate analysis from 8 depth intervals. Depth difference between the first- and second**

2 **replicate samples is 10 cm.**

| | 1st measurements | | | | | 2nd measurements | | | | |
|---|---|---|---|---|---|---|---|---|---|---|
| Depth | Dup.1 | Dup.2 | Mean | 1sigma | Date | Dup.1 | Dup.2 | Mean | 1sigma | Date |
| (m) | (ppb) | (ppb) | (ppb) | (ppb) | (dd/mm/yy) | (ppb) | (ppb) | (ppb) | (ppb) | (dd/mm/yy) |
| 523.150 | 634.8 | 634.7 | 634.7 | 0.1 | 27-1-14 | 637.5 | 634.3 | 635.9 | 1.6 | 24-2-14 |
| 530.950 | 669.0 | 665.8 | 667.4 | 1.6 | 03-2-14 | 669.4 | 670.7 | 670.0 | 0.7 | 24-2-14 |
| 558.295 | 682.5 | 678.2 | 680.3 | 2.2 | 14-3-14 | 687.5 | 678.3 | 682.9 | 4.6 | 02-4-14 |
| 559.850 | 689.8 | 680.3 | 685.0 | 4.7 | 03-2-14 | 683.8 | 690.0 | 686.9 | 3.1 | 26-3-14 |
| 561.150 | 687.8 | 689.2 | 688.5 | 0.7 | 14-3-14 | 684.0 | 690.4 | 687.2 | 3.2 | 02-4-14 |
| 562.407 | 687.2 | 685.5 | 686.4 | 0.8 | 26-3-14 | 689.4 | 686.4 | 687.9 | 1.5 | 02-4-14 |
| 575.913 | 679.2 | 679.2 | 679.2 | 0.0 | 07-2-14 | 686.7 | 678.9 | 682.8 | 3.9 | 28-3-14 |

**Table 2. Results of the 3-box source distribution model from the combined IPD showing emissions of tropical (green, T)**
**and boreal (red, N) boxes and boreal source fraction (N/(T+N+S)) at specific time slices. Also shown are previous**
**estimates for comparison. Errors denote 95% confidence interval. The uncertainty for 9.5 – 11.5 ka period is the average**
**of 95% confidence interval of the low-pass filtered reconstruction of each box emission.**

| Ref. | N box | T box | Boreal source fraction N/(N+T+S) |
|---|---|---|---|
| (ka) | (Tg yr$^{-1}$) | | (%) |
| Brook et al., 2000 (9.5-11.5 ka) | 64 ± 5 | 123 ± 8 | 32 ± 3 |
| Chappellaz et al., 1997 (9.5-11.5 ka) | 66 ± 8 | 120 ± 9 | 33 ±3 |
| This study (9.5 – 11.5 ka) | 67 ± 3 | 118 ± 5 | 33 ± 2 |
| This study (11.5 ka) | 60 ± 7 | 118 ± 12 | 31 ± 4 |
| This study (9.5 ka) | 71 ± 3 | 109 ± 5 | 36 ± 2 |
| This study (9.0 ka) | 66 ± 4 | 112 ± 7 | 34 ± 2 |

