# Peer review of "Atmospheric methane control mechanisms during the early"

_Climate of the Past, 2016_

## Referee Comment (RC1) · Anonymous Referee #1 · 19 Aug 2016

**Summary of manuscript**

First of all, I would like to congratulate Ji-Woong Yang et al. for the excellent work they put into producing a high-resolution record of CH$_4$ mole fractions as well as their interpretation of the data. So far, the early Holocene is underrepresented in high-resolution CH$_4$ reconstructions and this paper will be a valuable addition to the literature. I hope the following comments will be helpful and I look forward to reading the revised version of the manuscript.

Ji-Woong Yang et al. reconstruct the CH$_4$ variability of the Early Holocene, from 11.6 to 8.5 ka before 1950, using a melt-refreeze extraction system coupled to a GC-FID analyser, which was newly developed at Seoul National University (SNU). The new method is very briefly described in this paper and not yet published. The authors show that the SNU data are in good agreement with two existing benchmark records from the WAIS divide ice core, where the latter two were measured using *i*) a similar technique (WAIS members 2015) and *ii*) a sample gas stream derived from a continuously melted ice core, analysed by an optical instrument (Rosen et al., 2015).

Ji-Woong Yang et al. observe millennial CH$_4$ minima besides the 8.2 ka event, which have not been identified in previous studies. The authors relate these CH$_4$ minima to events in other geological records that indicate climate variability in the low and high latitudes of the Northern Hemisphere. These records include: $\delta^{18}$O-H$_2$O (NGRIP), ice rafted debris, $^{10}$Be, reflectance of Cariaco Basin sediments, $\delta^{18}$O-CaCO$_3$ (speleothems) and $\Delta\varepsilon$. The authors show convincingly that the millennial CH$_4$ variability correlates with millennial variations in $\delta^{18}$O-H$_2$O (NGRIP)/Greenland temperature, which is a new and interesting finding. The authors furthermore discuss the relation of CH$_4$ with the other records and suggest that Northern Hemispheric cooling and a concomitant southward shift of the ITCZ created a teleconnection pattern of reduced intensities of Asian and Indian monsoons. Thereby, the authors identified changes in CH$_4$ emissions from tropical wetlands as the most likely cause of the millennial CH$_4$ minima. The authors claim that this mechanism cannot explain the CH$_4$ minima around 10.2 ka alone.

In a next section, the authors review how the variability in external forcing during may cause an "El Nino-like" climate. They discuss some relation in the climate system but how this is hypothetically related to their CH$_4$ data remains unclear. The authors conclude this discussion cannot be developed further as there is no ENSO reconstruction for the Early Holocene. The purpose of this section is not entirely clear to me, also in the light of a range of existing publications on ENSO reconstructions (e.g. Z. Liu et al., 2014, Nature, Vol. 515, p. 550-553)

In order to investigate the CH$_4$ record further, the authors calculate the inter-polar difference in CH$_4$ (IPD) using the presented SNU and previously published NEEM data. The calculated IPD record is close to the range of previously published estimates, but exhibits interesting features on millennial time scales. The authors suggest a high IPD at the onset of the Holocene, which then decreases between 11.1 and 10.7 ka and increases again between 10.7 and 9.9 ka to previous levels. They furthermore use a previously published box model to separate hemispheric and tropical CH$_4$ emissions. The authors discuss the variability of tropical and boreal source fractions over time in the light of other studies and conclude that the IPD increase between 10.7 and 9.9 ka is a result from Northern Hemispheric warming/thawing and expansion of boreal wetlands. This is in convincingly good agreement with previous reconstructions of increased CH$_4$ emissions from boreal wetlands.

The study of Ji-Woong Yang et al. is a valuable contribution and in the scope of Climate of the Past. I would recommend the publication of this manuscript but think that major revisions are required.

**General comments:**

1) Ice core: The authors could provide more complete information on the ice core and related logistics. For example, it is not explicitly mentioned that the samples were shipped from the USA to South Korea (just names of institutions). Neither did the authors mention the year the ice core was retrieved nor how long it was stored prior to analysis. Do the authors think that storage time affects the analysis? Do the authors think that extended storage time could help with the handling of samples from the brittle ice zone? The authors did not specifically address that their samples were from the brittle ice region. However, I would strongly recommend to raise this point and how it might or might not have affected the data.

2) The analytical system: Even though the authors are preparing another manuscript on the analytical method, a more detailed description of the analytical system would be helpful. For example, the authors describe their melt-refreeze method as "traditional", though, I am in doubt that most readers have a melt-refreeze technique in their lab, if they work in a lab at all. I find the term "traditional" misleading, as it might be taken as a support for the performance of the method.

I think the presentation of the analytical performance needs to be developed further. The authors describe their system as similar to Mitchell et al., (2011). To my understanding, the system of Mitchell et al., (2011) is the benchmark GC-FID system in the community, with an estimate for measurement uncertainty of ±2.8 ppb, based on the pooled standard deviation of replicate samples that were measured with extended periods of time between the analysis of each sample pair. That is, the uncertainty estimate of Mitchell et al., (2011) can be understood as "worst case scenario". The method of Yang et al. is presented with an uncertainty of ±1.0 ppb, which is determined by the pooled standard deviation of 8 replicate measurements. While I am more than happy to be convinced that a newly developed method is superior in performance to an existing method, I feel strongly that this claim has to be proven. I think a more detailed description of the method to determine the uncertainty estimate is required, especially because the method of Yang et al. is supposed to be by far superior than that of Mitchell et al., (2011). In the light of a total number of 295 samples measured for this study, the authors need to describe why they chose these specific 8 replicate samples to determine the analytical uncertainty and why they did not chose other samples. Furthermore, the method includes several corrections, including blank (determined with bubble-free ice, 5-15 ppb), gravitational fractionation (1.97±0.15 ppb) and correction for dissolved $CH_4$ (range should be specified). These corrections include uncertainties that should be represented in the uncertainty budget of the method. Furthermore, a blank of 5-15 ppb is enormous, both in absolute values as well as in the range, especially when the total performance is stated with ±1.0 ppb. For comparison, the method of Mitchell et al., (2011) has a blank of 1.1±0.5 ppb, while a blank of 5-14 ppb was interpreted as indicator of leakage. I think this issues need to be clearly demonstrated so that superior performance can be claimed. Further issues that could be clarified include the uncertainty of the standard air (available for each NOAA04 cylinder), how the linearity of the system is controlled (additional air standards covering the analytical $CH_4$ range?) and what reason the authors base their decision on to subtract 3 ppb from the OSU data instead of adding 3 ppb to the SNU data or to take the 3 ppb as temporal signal? If the measurement uncertainty claimed by both institutes is realistic, both data should be on the NOAA04 scale and therefore be in close agreement. I think this is in the order of expected disagreement, but I feel the manuscript is stronger if these issues are clearly addressed.

3) CH$_4$ data: The authors mention 295 measurements. However, Figure 1 seems to show a much smaller number. If the displayed measurements are averages of duplicates or if other measurements are not displayed in Figure 1, the number would have to be revised. Otherwise, please clarify. How long is the overlapping period between OSU and SNU data? Maybe Figure 1 could show this in a detailed Figure? Figure 1 has a reference in the captions (Brook and Sowers, 2016) that is new to me and that I cannot find in the reference list. The authors may or may not consider to show data from previous publications (e.g. Brook et al., 2000, Flueckiger et al., 2002) to highlight the superiority of their temporal resolution.

4) IPD: IPD is a powerful concept, but very one has to be very careful in its reconstruction and interpretation. The authors mention the potential for ill-calculated IPDs based on errors in the gas age scale of the CH$_4$ records. Therefore, the authors developed a tool to synchronize the CH$_4$ records, which I think is a very good approach! However, it is not clear to me why the authors chose to calculate the IPD based on the Siple Dome data? My concerns have several reasons:

i) The authors state themselves that the histories of gas enclosure is more similar between NEEM and WAIS. The authors state that, based on just this fact, the amplitude of CH$_4$ variations is 10-20 ppb larger in the WAIS than in the Siple Dome record. Therefore, I understand that both the NEEM and the Siple Dome CH$_4$ records are altered by physical processes during gas enclosure that are different for each core. I understand that a dampened amplitude in the Siple Dome CH$_4$ record would create a IPD variation. Ideally, gas enclosure effects should be identical in both records so that they would cancel. Since the WAIS record is in that sense more similar to the NEEM record than the Siple Dome record is, I would suggest to use the WAIS record for the IPD reconstruction.

ii) The WAIS record is of even higher temporal resolution than the Siple Dome record. I would expect that the IPD reconstruction based on NEEM and WAIS would be more robust.

iii) The comparison of the CH$_4$ records from Siple Dome and WAIS in Figure 1 shows two periods (~10.5 and ~10.7 ka) where Siple Dome CH$_4$ exceeds WAIS CH$_4$ by up to ~20 ppb. During this period, the smoothed CH$_4$ variations (Figure 2) also show an ice core specific pattern of disagreement. Around 11 ka, the pattern of agreement is different. Here, the continuous CH$_4$ record from WAIS seems to agree better with CH$_4$ from Siple Dome, while the GC-FID record from WAIS contains a number of samples that exceed the former records by ~20 ppb. The difference between the records during these times exceeds the stated measurement uncertainty by far. It is also important that the difference seems to be in the order of the presented IPD variability.

All of the above mentioned reasons directly impact on the IPD reconstructions. The choice of the authors to use their Siple Dome data for the IPD reconstruction is justified and understandable. However, I fear that the interpretation is sensitive to the choice of CH$_4$ record so that this choice could impact on the IPD result for the above mentioned reasons. Therefore, I suggest to calculate the IPD also using both WAIS records as three independent sets of IPD reconstructions as a sensitivity test. This will make the interpretation of reconstructed IPDs more robust and will furthermore give valuable insights into the IPD technique on high temporal resolution records for future studies. Because this concerns one of the main outcomes of this manuscript, I would consider this essential.

5) Consideration of significant publications: *i*) The authors claim that no correlation between $CH_4$ and tropical monsoon signals has been reported on shorter time scales, however, I feel this is not accurate. Both Cruz et al., (2005, Nature, Vol, 434, p. 63-65) and Sperlich et al., (2015, Global Biogeochemical Cycles, 29) have related $CH_4$ and $\delta^{13}$C-$CH_4$ records to South American speleothem records, respectively. Both publications would principally support the interpretation of this study. *ii*) The authors discuss that their IPD reconstruction suggests increasing boreal source fractions during the early Holocene and support their finding with studies on boreal wetland dynamics. However, their finding of increased boreal source fractions is in line with the interpretation of $\delta^{13}$C-$CH_4$ data by Fischer et al., (2008) and Sowers, (2010). Again, both publications would principally support the interpretation of this study while Sowers (2010) had the same finding for the early Holocene previously. *iii*) The authors stated that there is currently no ENSO reconstruction for the early Holocene, even though a range of them exist (e.g. Z. Liu et al., 2014, Nature, Vol. 515, p. 550-553, Clement et al., 1999, Paleoceanography, Vol. 14, p. 441-456).

6) Chosen data filtering technique: I suggest to provide more information why a 250 year window width was used. What is the effect of other window widths on the data you use and on your resulting interpretation? The authors state on p5L2-5 that the 250 year window was also used in other studies. However, that is not necessarily a satisfying justification. The window width should be carefully chosen in dependence on the time scales you are investigating.

7) Sometimes, I have trouble to understand the point the authors intend to make, e.g. p1, 12–18; p5, 18–29; p6, 25–p7, 5; p10, 6–12, p22, 1-2. Please consider re-wording.

8) I have difficulties to follow the discussion and the display of $CH_4$ and other records. For example, the authors discuss why there is agreement/disagreement between some records within the uncertainty of the age model. However, I find it tough to see this in Figure 2. For example, the $CH_4$ minima are highlighted with yellow bars. During 9.4 and 10.2 ka, the yellow bars include both local minima and local maxima of the other records, e.g. of the Cariaco Basin record, of $\Delta\varepsilon$, of Dongge Cave. Other local extrema, e.g. in the Cariaco Basin record have no correspondence in $CH_4$ during 8.5-8.7 ka or 10.5 ka, which is not mentioned at all. Therefore, I feel this discussion needs to be further developed to provide more guidance to the reader. Is [10]Be really important here? Sometimes it seems to correlate, other times it is anti-correlated. Could figure clarity increase without it?

9) Figure 2: Presented is $CH_4$ anomaly. I don't see an advantage of anomaly over $CH_4$ mole fractions. Also, how is anomaly of 0 defined for each record?

10) Structure: I would suggest to avoid three levels, e.g. 3.1.1 and 3.1.2 but make 3. Millennial scale variability, 4. Latitudinal distribution…. to keep structure with max. two levels.

11) In many places of the manuscript, the author review literature, e.g. on pattern of climate teleconnections, for which they allow extensive text sections. While I think it is important to review

in such detail, I feel the authors could improve their discussion of how they think this is linked/relevant to their $CH_4$ interpretation. A good example for this is the entire section 3.1.3. I would like to encourage the authors to consider this throughout the entire manuscript, even though this probably means either adding to, or re-writing many sections of the manuscript.

12) Understanding the variability in tropical wetlands is crucial for the understanding of $CH_4$ source regions and tropical $CH_4$ fractions. (The same rule applies for boreal wetlands.) The authors fully acknowledge this throughout the manuscript. However, I note that the authors exclusively focus their interpretation on Asian/Indian monsoon systems. It has been described previously that the African monsoon system and wetland extension changed tremendously throughout the Holocene (e.g. Sahara region etc). There are also several publications on South American monsoon systems besides the Cariaco Basin reflectance, which I understand the authors only use as proxy for ITCZ migration, but not for their interpretation of hydrological changes in South American wetlands. Including further records (e.g. Cruz 2005) might allow for a more comprehensive evaluation of hydrological changes. Based on $\delta^{13}$C-$CH_4$ data, the South American monsoon system has recently been suggested to be a controlling factor in rapid $CH_4$ changes leading up to DO21 (Sperlich 2015). I would strongly recommend to either include a complete representation of tropical wetlands or to discuss why you think monsoon systems other than the Asian/Indian are not relevant.

13) Data availability: I understand that Copernicus has developed a new policy for authors to provide either descriptions on data access, or to provide the data through international data-bases or supplementary information. Copernicus requires a dedicated section that describes this in detail, which the authors might want to consider during their revisions.

**Specific comments**

The manuscript may be subject to considerable re-writing. Therefore, the specific comments will not include comments on grammar, wording or writing that might be subject to change. Since I am not a native English language speaker myself, I am not sure to what extend my comments would help to make it better or worse. Please understand suggested re-formulations as suggestions, only.

p1L1: Understanding processes controlling atmospheric methane

p2L2: reference Daniau et al., is it possible to provide a reference on palaeo-fire or a reference that is more specific on pyrogenic gas emissions?

p2L5: sink strength and light availability? e.g. polar winter

p2L16: Lisiecki and Raymo 2005, though this reference is not on $CH_4$, general I think a reference on $CH_4$ and Northern Hemisphere temperature would be useful here

p2L18: See comment 5) in general comments

p2L18-19: too weak. The correlation between $CH_4$ and NH temperature ($\delta^{18}$O-ice) is well established

p3L15: here and everywhere else: It is recommended to restrict the use of "concentrations" to public debate but to use "mole fraction" or "mixing ratio" in scientific literature (WMO, recommendations of GGMT experts)

p3L17: here and everywhere else, $\delta^n$X, the $\delta$ is supposed to be *italicised*, same for Δ➜$\Delta$ (Coplen 2011, DOI: 10.1002/rcm.5129)

p3L27: there is still some ice left in Greenland

p3L30: ensure stated, used and displayed sample numbers agree,  give age interval with depth

p3L32: (NICL, city, state, country), (SNU, city, state, country)

p4L3: "are described in…", referring to a paper that is currently in preparation as XYZ et al., (in prep.) seems strange as it is not useful to look it up. "A manuscript that describes the method in detail is currently in preparation."

p4L7: was the standard air added before or after the bubble-free ice was melted?

p4L7: "traditional" melt-refreeze seems misleading to me, traditional can be left out

p4L7-17: see point 2) in general comments

p4L17: provide reference how you calculated gravitational fractionation

p4L19: provide reference for GICC05 time scale

p4L25: "one of the high resolution data sets" sounds vague and strange to me, where do you draw the line between high and low resolution? The temporal resolution of your data is higher than some but lower than both WAIS records. "It has the currently second/third highest temporal resolution of Antarctic $CH_4$ records covering the early Holocene."

p4L26: develop a more complete representation of analytical uncertainty

p4L30: describe the overlapping interval and describe why you think the OSU record should be adjusted to match the SNU data. Why not the other way around, why not accepted as real difference? Both should be on NOAA04?

p5L3: this comparison example only makes sense if you look at variations on similar time scales. Otherwise the argument that you use the same filter as has been applied for the WAIS record is not valuable, but could be misleading.

p5L14: add references that show anthropogenic signal in LPIH $CH_4$, e.g. Ferretti 2005, Mischler 2009, Sapart 2012)

p5L16: "even though this conclusion is less robust as there are no age tie-points…"

p5L21-22: shift this sentence to after the following sentence to keep logical flow from NH to tropics

p5L25-29: describe the meaning for $CH_4$, develop the discussion towards $CH_4$, what does a ITCZ shift mean for South American $CH_4$ source regions?

p6L4: there are other monsoon systems that Asian/Indian that should be considered

p6L10: the monsoon intensity change. (delete Asian, include other monsoon systems)

p6L20: even though the Cariaco Basin record is shown, it is presented only as indicator for ITCZ migration, without direct connection to $CH_4$. I feel the assumed passiveness of South American $CH_4$ source regions during the study period might not be a natural assumption and should be explained.

p6L23-p7L5: describe relevance for $CH_4$, what is the $CH_4$ controlling process chain? A sentence that says "the proxies show this and that which could explain the increase/decrease in $CH_4$ during time period XY".

p7L8-9: you could add Cruz et al., 2005 to the reference list, as they discussed the interplay of solar radiation, monsoon intensity and $CH_4$ mole fractions

p7L16-17: see above comment regarding references on ENSO variability during Holocene period

p7L23: provide temporal resolution of NEEM record, 1 sample in how many years?

p7L24: consider IPD reconstructions with $CH_4$ records from WAIS

p7L25-30: NICE approach!

p8L2: …show an increase by XYZ ppb from…

p8L4: …in both hemispheres during…

p8L6: …from both hemispheres and…

p8L8-9: extra-tropical latitudes (30N or 30S is not high latitude, rather extra-tropical)

p8L13-14: develop description of model assumptions and impacts, e.g. what life times did you assume and why? did you tune life times to match previous flux estimates?

p8L15-16: quantify and discuss flux estimates, otherwise meaningless

p8L18: in tropical emissions by XYZ Tg. (quantify)

p8L25-30: increased $CH_4$ emissions from boreal wetlands were previously suggested by Sowers 2010, that agreement should be acknowledged

p8L29: explain "conventional" northern $CH_4$ emissions

p9L1: the isotope records are already published and need to be acknowledged (Sowers, 2010). these isotope data are available and should be added to the figures of this manuscript.

p9L10-15: therefore, IPD should be calculated with WAIS records as well

p9L30: there is also no explanation for the drop in IPD if I am not mistaken?

p10L5-6: why can the 10.2 ka event not be explained by low latitude hydrology, but the other events can? I feel this is a section where great care has to be taken to prevent from over interpretation. There is no quantitative estimate for low latitude emissions during other events, i am not convinced that the presented records allow for a partial explanation of the $CH_4$ minima and that there is only a missing bit. I would recommend to re-formulate. Even if the revised IPD reconstruction supports the current discussion, this might seem as two results are made to fit together. I feel this can be toned down and still be strong a conclusion.

p10L6-12: I am not sure I understand this section

p10L11: the quantification 20-40 ppb is mentioned for the first time here. The conclusions cannot include information that have not been presented earlier in the manuscript. I am not sure how you quantify ppb changes? Is that from the box model?

p16L19: I didn't check all references, but Sowers 2010 is not correct. This is "Atmospheric methane isotope records covering the Holocene period, Quaternary Science Reviews 29, 213-221, 2010". The title/journal name in your references refers to his 2006 paper

p18L4: add reference to the list or check reference

p18F1: show overlapping period, show minor ticks on both axes,

p19F2: define 0 ppb in anomaly or show $CH_4$ mole fractions, check width of yellow bars, can be confusingly wide

p20F3: add IPD with WAIS data

p22T1: caption is confusing to me, also what is this table supposed to add? how can this agreement be explained, life time?

---

## Referee Comment (RC2) · Anonymous Referee #2 · 23 Sep 2016

Overall assessment

The manuscript provides new high-resolution CH4 data from the Siple Dome ice core over the time interval 8.5-11.5 kyr BP, extending previous work by Ahn et al., 2014. The data quality is generally very good (see comments below) and I commend the authors for their painstaking work to provide high-resolution data sets using discrete CH4 analyses. The data is interpreted with respect to millennial climate variations during this time interval and, based on correlation with other climate proxy data, a suggestive hypothesis about the influence of changes in the ITCZ is presented to explain the millennial variability in CH4.

Finally, the interpolar CH4 difference (IPD) is calculated using Greenland data from the literature and this difference is then analyzed using a simple three-box model. As

outlined below, I have some fundamental questions about the reliability of the inferred IPD, which subsequently has also important implications for its interpretation. This prevents me from recommending the manuscript for publication in CP in its current form despite the nice data. Moreover, the quality of the manuscript in terms of the use of the English language has to be considerably improved. With a native English speaker on the author list, I see no problem that this can be achieved. In summary, after additional work I am confident that the manuscript will become suitable for publication in CP in a resubmitted version.

General comments

a) comparison of early and late Holocene CH4 variations: In the introduction the authors make the important point that only the early Holocene period allows us to study the natural CH4 variability on centennial to millennial time scales. Unfortunately, the authors do not follow up on this in the discussion. It would be interesting to compare the centennial and millennial variability in CH4 concentrations in the early Holocene (as documented in the Siple Dome, WAIS (including the continuous CH4 data by Rhodes et al., 2015) and NEEM record with the late Holocene as documented in WAIS by Mitchell et al., 2013. Are the amplitudes of this variability different and if so, is that due to an anthropogenic influence in the late Holocene or related to the changes in seasonal and geographical distribution of solar insolation (due to orbital parameter changes) between the early and late Holocene? Note that summer insolation in high northern latitudes was several tens of W/m2 higher in the early Holocene. For this analysis it may be beneficial to use the continuous WAIS data instead of the Siple Dome data to calculate a record with comparable resolution to the Mitchell data and to compare CH4 frequency spectra for the early and late Holocene.

b) Millennial CH4 variations: The authors suggest that climate cooling in the northern hemisphere has led to a southward shift in the ITCZ, which again led to a decline in CH4 low latitude emissions due to changes in monsoon systems. The first part of this hypothesis (ITCZ shift) appears to be straightforward and has been observed in

models, however, the second part (CH4 emission changes) appears not so straightforward and requires some more quantitative support. Rhodes et al. (2015) suggest a first order relationship between CH4 emissions and intense rainfall area, where from a certain point on also an increase in southern hemisphere wetland emissions is possible. Accordingly, a discussion focusing on Asian monsoon systems only, as in the manuscript by Yang, seems to be too narrow. Please explain how your hypothesis fits into this picture. Please note also the work by Bozbiyik et al., CP, 2011, performing a North Atlantic fresh water hosing experiment under interglacial conditions connected to a southward shift of the ITCZ, showing decreases in tropical precipitation and the modeling work by Zurcher et al., Biogeoscience, 2013, which shows that also boreal peatland CH4 emissions are reduced during such an experiment. Finally, the discussion of the millennial CH4 variability and the corroborating proxy evidence from other archives lacks some clarity and could be improved.

c) Interpolar CH4 difference: The IPD is a tricky business and erroneous effects can be easily introduced by comparing CH4 data from different labs, different sites, or insufficient robustness of the results. Accordingly, this needs more supporting information and detail:

In the method part it is mentioned that blank ice measurements show an offset with a very large scatter between 5 and 15 ppb. I read the text in such a way that a daily blank correction is applied based on 4 blank measurements per day. This needs more detailed discussion as a potentially erroneous correction, which varies by 10 ppb, has a huge influence on the IPD, which varies with a similar amplitude. Please add the following information/discussion:

- Can you be sure that the CH4 blank is coming from the extraction system and is not reflecting dissolved CH4 in the bubble-free ice? In the latter case you should not correct for this blank. Did you perform blank tests without bubble-free ice or did you repeat the extraction of bubble-free ice for a second or third time to see, whether the blank is constant or declining?

- Please add information on the time scale on which the blank changes. If my understanding is correct that a daily mean blank correction of 5-10 ppb is performed, it is important to know what the variability of the four blank ice measurements is within one day. If this intra-day variability is of the same size as the inter-day variability, then a daily blank correction varying between 5 and 15 ppb introduces offsets from one day to the other which only reflect the stochastic variability of the blank measurements themselves and not systematic day-to-day differences in the entire measurement system. In that case a long-term mean blank correction seems more appropriate. If the blank values are reproducible within one day, a daily correction seems justified.

- On the other hand if you have a long-term trend in this blank value, which may not reflect a trend in the extraction system but in the bubble-free ice quality, you introduce this error into the IPD. Did you use a randomized order to measure the samples to avoid such spurious trends.

- There was an average offset of 3 ppb observed between Siple Dome data measured at SNU and OSU and the OSU data have been corrected by subtracting 3 ppb, but it is not discussed what the influence of this correction may be on the IDP. Note that the NEEM discrete data used to calculate the IPD are also measured at OSU. Accordingly, to avoid any systematic errors in the IPD it is mandatory to add the 3 ppb to the Siple Dome data measured at SNU and not to subtract 3 ppb from OSU data.

Comparing the continuous WAIS data with the discrete (or continuous) CH4 data from NEEM, it becomes apparent that the relative changes in CH4 concentrations in the northern and southern hemisphere are much more similar than when comparing Siple Dome and NEEM data after the Monte Carlo synchronization in Figure 3. For example the downward trend in NEEM between 11.3 and 10.9 kyr BP is also seen in the continuous WAIS data, while in Siple this time interval shows essentially constant CH4 after a first short peak. Consequently, the constant values in the Siple data lead to an erroneous downward trend in the IPD in this time interval. Vice versa, there is an upward trend from 10.9 to 10.4 kyr BP found in NEEM and continuous WAIS data. The

same time interval in Siple looks more like a broad maximum, again with implications on the derived IPD. A similar observation holds for the maximum around 10 kyr BP. Note that on these centennial to millennial time scales, which are much longer than the atmospheric lifetime and the interhemispheric exchange time, you may have changes in the size of the IPD, however, it is extremely difficult to create a millennial trend in one hemisphere without a trend in the other. This is nicely illustrated in the high-resolution data by Mitchell et al., 2013.

Obviously, the resolution and quality of the data in Figure 3 does not suffice to gain a robust IPD and/or the Monte Carlo synchronization fails to synchronize the records sufficiently. In fact, it seems crucial that the IPD analysis is performed not only on the Siple but also on the WAIS discrete and continuous data to study the robustness of the results gained from the Siple Dome core. Note that the WAIS very high-resolution data from continuous measurements can be used to much better synchronize WAIS to the continuous records available from NEEM. This would circumvent the synchronization problems apparent between Siple and NEEM. As a final remark on this topic, I do not agree with the authors' statement that the IPD values in Siple Dome over the time interval 9.5-11.5 kyr BP are in agreement with previous results. If you calculate the mean over this time interval in the Siple IPD data and calculate the standard error, this appears to clearly higher than the literature values. In summary, the IPD discussion needs more work before the manuscript should be published in CP.

Specific comments

I started to correct for English language issues, but had to stop at some point. Please ask your English speaking co-author for a thorough language check not only for typos but also to improve the clarity of the arguments. As major textual changes are still required for this manuscript, I will not comment on language issues here.

P(age) 2 l(ine) 2: Daniau et al., 2012 is not an appropriate reference in this respect (CH4 emissions)

P2 l20: Cite recent work by Baumgartner et al. CP 2014

P2: discuss in more detail previous work on the relationship between ITCZ changes and CH4 emissions

P3: discuss the difference in orbital parameters for the early and late Holocene and the potential implications for CH4 emissions

P4 methods: Is it correct that you use only a one standard calibration? Comment on the potential systematic error introduced by this approach

P4 l25-32: This paragraph should be moved to the methods section

P5 1st paragraph. You say that you use a 250 year running average (and similar a high-pass filter with 1800 cut-off), however, your data is not equidistant. Please explain in more detail how you averaged the data

P5 l14: Is the significance level of the correlation coefficient really taking the reduced degrees of freedom into account after averaging the data? Looking at the value, I am afraid it didn't and the significance is highly overestimated.

P5: see comment on insufficient discussion of the effect of an ITCZ shift on CH4 emissions north and south of the equator. Please discuss also in more detail the dating uncertainties of the various archives and their potential impacts on the conclusions.

P5 l30: Reference Bjorck et al. is not in the reference list

P6 l5-7: There is also variability in GRIP and GISP2. Please explain in more detail what you refer to.

P7 l1-17: This paragraph is highly speculative and lacks clarity and detail.

P7 following l24: You disturbed the age of the data points by a Gaussian distribution with sigma=30 years. How did you make sure that the chronological order of all data points was ensured in your approach? How did you take the measurement uncertainty

in each data point into account? Please explain in more detail.

P8 l1: there is a significant offset between your average and previous IPD estimates

P8 l10-12: not entirely clear to the outsider what you did, please clarify

P8 l19: the boreal sources increased

P9 l10. You discuss the effect of the different age distributions in the Siple Dome and NEEM cores, but you do not follow up on this in your analysis. Either you use WAIS to compare with NEEM (as it has essentially the same enclosure characteristics) or you low-pass filter NEEM to the same enclosure characteristics as Siple. I would strongly recommend to do both to study the robustness of the results.

P9 l20-21: The results by Fischer et al. (2008) on LGM biomass burning emissions result from the use of temporally constant isotopic source signatures in the box model approach. Moller et al., Nature Geoscience, 2013 showed that also the source signatures changed significantly over time and they revised the biomass burning estimates, showing that LGM emissions were lower than Holocene emissions.

P9 l26: why do you only refer to biomass burning in the tropics?

P20 l5 Chappellaz et al., 1997 not 2013

––––––––––––––––––––––––––––––––

---

## Editor Comment (EC1) · EW Wolff (Editor) · 4 Oct 2016

Your paper has receieved two reviews. They both consider your paper could have scientific significance but they both have significant concerns particularly about the calculation of the interpolar difference (IPD), as well as other issues that could be addressed. They make some serious technical challenges that question whether you can derive a reliable IPD from your Siple Dome data. One reviewer goes as fra as to recommend rejection, while the other asks for major revision. You are now invited to post author responses to all the review comments and I urge you to do that. However I suggest you do NOT prepare a new version of the paper until after that. Once the author comments are all there I will be asked as editor to make a formal editorial decision and recommendation. While I certainly hope that parts of the paper can be improved and published, I do not want to encourage you to revise the paper itself until I have seen

your responses. This will allow me to decide how to recommend that you proceed.

---

## Author Comment (AC1) · 26 Dec 2016

*We thank the anonymous referee #1 for her/his careful review of our paper. We appreciate the useful comments and believe the input will improve the manuscript. The original referee comments are copied in black, and the author's response to the comments are given in red italics. We add paragraphs from original discussion paper in blue italics and our modifications in green italics.*

**Summary of manuscript**

First of all, I would like to congratulate Ji-Woong Yang et al. for the excellent work they put into producing a high-resolution record of $CH_4$ mole fractions as well as their interpretation of the data. So far, the early Holocene is underrepresented in high-resolution $CH_4$ reconstructions and this paper will be a valuable addition to the literature. I hope the following comments will be helpful and I look forward to reading the revised version of the manuscript.

Ji-Woong Yang et al. reconstruct the $CH_4$ variability of the Early Holocene, from 11.6 to 8.5 ka before 1950, using a melt-refreeze extraction system coupled to a GC-FID analyser, which was newly developed at Seoul National University (SNU). The new method is very briefly described in this paper and not yet published. The authors show that the SNU data are in good agreement with two existing benchmark records from the WAIS divide ice core, where the latter two were measured using *i*) a similar technique (WAIS members 2015) and *ii*) a sample gas stream derived from a continuously melted ice core, analysed by an optical instrument (Rosen et al., 2015).

Ji-Woong Yang et al. observe millennial $CH_4$ minima besides the 8.2 ka event, which have not been identified in previous studies. The authors relate these $CH_4$ minima to events in other geological records that indicate climate variability in the low and high latitudes of the Northern Hemisphere. These records include: $\delta^{18}O$-$H_2O$ (NGRIP), ice rafted debris, $^{10}Be$, reflectance of Cariaco Basin sediments, $\delta^{18}O$-$CaCO_3$ (speleothems) and $\Delta\varepsilon$. The authors show convincingly that the millennial $CH_4$ variability correlates with millennial variations in $\delta^{18}O$-$H_2O$ (NGRIP)/Greenland temperature, which is a new and interesting finding. The authors furthermore discuss the relation of $CH_4$ with the other records and suggest that Northern Hemispheric cooling and a concomitant southward shift of the ITCZ created a teleconnection pattern of reduced intensities of Asian and Indian monsoons. Thereby, the authors identified changes in $CH_4$ emissions from tropical wetlands as the most likely cause of the millennial $CH_4$ minima. The authors claim that this mechanism cannot explain the $CH_4$ minima around 10.2 ka alone.

In a next section, the authors review how the variability in external forcing during may cause an "El Nino-like" climate. They discuss some relation in the climate system but how this is hypothetically related to their $CH_4$ data remains unclear. The authors conclude this discussion cannot be developed further as there is no ENSO reconstruction for the Early Holocene. The purpose of this section is not entirely clear to me, also in the light of a range of existing publications on ENSO reconstructions (e.g. Z. Liu et al., 2014, Nature, Vol. 515, p. 550-553)

In order to investigate the $CH_4$ record further, the authors calculate the inter-polar difference in $CH_4$ (IPD) using the presented SNU and previously published NEEM data. The calculated IPD record is close to the range of previously published estimates, but exhibits interesting features on millennial time scales. The authors suggest a high IPD at the onset of the Holocene, which then decreases between 11.1 and 10.7 ka and increases again between 10.7 and 9.9 ka to previous levels. They furthermore use a previously published box model to separate hemispheric and tropical $CH_4$ emissions. The authors discuss the variability of tropical and boreal source fractions over time in the

light of other studies and conclude that the IPD increase between 10.7 and 9.9 ka is a result from Northern Hemispheric warming/thawing and expansion of boreal wetlands. This is in convincingly good agreement with previous reconstructions of increased $CH_4$ emissions from boreal wetlands.

The study of Ji-Woong Yang et al. is a valuable contribution and in the scope of Climate of the Past. I would recommend the publication of this manuscript but think that major revisions are required.

**General comments:**

1)      Ice core: The authors could provide more complete information on the ice core and related logistics. For example, it is not explicitly mentioned that the samples were shipped from the USA to South Korea (just names of institutions). Neither did the authors mention the year the ice core was retrieved nor how long it was stored prior to analysis. Do the authors think that storage time affects the analysis? Do the authors think that extended storage time could help with the handling of samples from the brittle ice zone? The authors did not specifically address that their samples were from the brittle ice region. However, I would strongly recommend to raise this point and how it might or might not have affected the data.

*More paragraphs dedicated to sample logistics and ice core will be added based on details below:*

- *Ice core and sample logistics: Siple Dome A (SDMA) deep ice core was drilled from 1997 to 1999 on the Siple Coast, West Antarctica (81.65S, 148.81W; 621m elevation) (Taylor et al., 2004). The Siple Dome ice core samples were selected and cut at National Ice Core Lab (NICL), Denver in January to February 2013. The samples were packed in isothermal foam boxes with numerous eutectic gels and shipped to South Korea via expedited air freight. A temperature logger was enclosed within the isothermal box to track the temperature change during the logistics, and it showed that the temperature was maintained below -25°C during the transit. Then the Siple Dome ice samples were stored in a walk-in freezer of Seoul National University (SNU) that maintained below -20°C, and the $CH_4$ analysis was carried out from autumn of 2013 to spring of 2014.*

- *Brittle zone: Siple Dome ice was reported to be severely fractured below 400 m depth (e.g., Gow and Meese, 2007). Hence, during the ice sample preparation at NICL, the samples were carefully selected from unbroken part of the ice cores. Replicate measurements demonstrate a good integrity and reproducibility of the adjacent Siple Dome ice samples. In addition, comparison between the Siple Dome $CH_4$ records from OSU and SNU shows a good agreement within analytical uncertainty and no systematic drift, which ensures again lack of contamination in ice from the brittle zone. Further, comparison of $CH_4$ data from various ice cores with the Siple Dome records (from both OSU and SNU) show good agreement (below Figure R1). Therefore, it seems unlikely that the brittle zone ice affected our results.*

2)      The analytical system: Even though the authors are preparing another manuscript on the analytical method, a more detailed description of the analytical system would be helpful. For example, the authors describe their melt-refreeze method as "traditional", though, I am in doubt that most readers have a melt-refreeze technique in their lab, if they work in a lab at all. I find the term "traditional" misleading, as it might be taken as a support for the performance of the method.

*We agree with the suggestion. We will delete the expression "traditional" and add a more detailed description of the analytical method as follows:*

*The air occluded in ice was extracted by melting and refreeze process under vacuum. Gas extraction line was evacuated to under detection limit of 0.1 mTorr, and our system leak check was done daily before analysis. Ice samples were prepared in a walk-in freezer in the morning of each experiment day, and trimmed the outermost >2 mm to eliminate possible contamination by ambient air during the storage. Then the samples were moved to the laboratory and placed in glass sample containers. The sample containers (sample flask hereafter) were custom-made glass flasks welded to stainless steel flange, and attached to the vacuum line with a copper gasket. The sample flasks were partially submerged into a chilled ethanol bath during ice insert and attaching to the line for preventing temperature increase by laboratory air. After that all flasks were evacuated at least 40 minutes, the ice samples were melted by submerging the sample flasks in a warm water bath. Melting process was usually completed within 30 minutes. The sample flasks were then submerged in the cold ethanol bath chilled to ~-82℃ more than 1 hour to refreeze. During the refreezing, we carried out GC pre-running (20 injections) and daily calibration that normally took 90 minutes. The ethanol temperature rises up to ~-55°C just after submerging the flasks, and it was chilled to below -65°C before expansion of the air in the flasks. The extracted air in the head space was expanded into a gas chromatograph (GC) equipped with a flame ionization detector (FID) to measure $CH_4$ mixing ratio. The detailed configuration of the vacuum line and GC is described in another paper that is currently in preparation (Yang et al., manuscript in preparation).*

*The inter-tank calibration using four working standard air cylinders (395.50, 721.31, 895.03, and 1384.91 ppb $CH_4$) that were calibrated and manufactured by NOAA GMD CCG in NOAA04 scale (Dlugokencky et al., 2005) shows good linearity of our GC system. The daily GC calibration curve was determined by measurements of a working standard having the closest $CH_4$ mixing ratio of expected value from the samples; in this study we used 721.31 ppb $CH_4$ standard for samples of the early Holocene. To account for system condition change throughout experiments (i.e., influence by water vapor), we calibrated with a standard air 6 times before and after sample measurements.*

I think the presentation of the analytical performance needs to be developed further. The authors describe their system as similar to Mitchell et al., (2011). To my understanding, the system of Mitchell et al., (2011) is the benchmark GC-FID system in the community, with an estimate for measurement uncertainty of ±2.8 ppb, based on the pooled standard deviation of replicate samples that were measured with extended periods of time between the analysis of each sample pair. That is, the uncertainty estimate of Mitchell et al., (2011) can be understood as "worst case scenario". The method of Yang et al. is presented with an uncertainty of ±1.0 ppb, which is determined by the pooled standard deviation of 8 replicate measurements. While I am more than happy to be convinced that a newly developed method is superior in performance to an existing method, I feel strongly that this claim has to be proven. I think a more detailed description of the method to determine the uncertainty estimate is required, especially because the method of Yang et al. is supposed to be by far superior than that of Mitchell et al., (2011). In the light of a total number of 295 samples measured for this study, the authors need to describe why they chose these specific 8 replicate samples to determine the analytical uncertainty and why they did not chose other samples.

*We will add more words to better clarify the methods as we described below.*

*"Duplicate measurements for 8 depths show ±1.0 ppb precision (1 sigma; pooled standard deviation)" will be more explained. Our $CH_4$ data are presented by averaging the results of duplicate sample analysis from the same depths. In order to estimate our data precision, we reanalyzed duplicate samples from the adjacent ices (~10 cm depth difference) at 8 depth intervals 8-80 days after the first analysis (see Table R1). The depth intervals were randomly chosen. The pooled standard deviation of the average of duplicates from first and second measurements was ±1.0 ppb (mean absolute difference of 1.9 ppb).*

**Table R1.** Summary of the first (original) and second (replicate) measurements from the depths used for system reproducibility test.

| Depth (m) | 1st measurements | | 2nd measurements | | Difference | |
|---|---|---|---|---|---|---|
| | $CH_4$ (ppb) | Date (dd/mm/yy) | $CH_4$ (ppb) | Date (dd/mm/yy) | $1^{st} - 2^{nd}$ (ppb) | Time (days) |
| 523.15 | 631.8±0.1 | 27/01/14 | 632.7±1.6 | 24/02/14 | -0.9 | 29 |
| 530.95 | 663.0±2.2 | 03/02/14 | 664.7±1.6 | 24/02/14 | -1.7 | 22 |
| 558.295 | 676.7±2.1 | 14/03/14 | 679.3±4.5 | 02/04/14 | -2.6 | 20 |
| 559.85 | 682.0±5.8 | 03/02/14 | 684.7±1.9 | 26/03/14 | -2.7 | 52 |
| 561.15 | 685.0±0.8 | 14/03/14 | 683.7±3.1 | 02/04/14 | 1.3 | 20 |
| 562.407 | 682.8±0.8 | 26/03/14 | 684.3±1.5 | 02/04/14 | -1.5 | 8 |
| 575.913 | 676.5±0.2 | 07/02/14 | 679.3±3.8 | 28/03/14 | -2.8 | 50 |
| 578.15 | 676.2±4.3 | 04/02/14 | 674.9±2.3 | 24/04/14 | 1.3 | 80 |

*We note that Mitchell et al. (2011) used different solubility correction methods. Applying the solubility correction method described in Mitchell et al. (2011) yields pooled standard deviation of the average of duplicates at adjusted samples of 1.4 ppb. One of the main differences of our analytical system compared to the OSU one is that, while in OSU the blank test was done once in several days and the systematic offset was interpolated between the blank days (Mitchell et al., 2011), we measured at least 3-4 blank ices every day. By doing this, the systematic offset can be quantified more precisely that accounts for daily changing conditions of instrument.*

Furthermore, the method includes several corrections, including blank (determined with bubble-free ice, 5-15 ppb), gravitational fractionation (1.97±0.15 ppb) and correction for dissolved $CH_4$ (range should be specified). These corrections include uncertainties that should be represented in the uncertainty budget of the method. Furthermore, a blank of 5-15 ppb is enormous, both in absolute values as well as in the range, especially when the total performance is stated with ±1.0 ppb. For comparison, the method of Mitchell et al., (2011) has a blank of 1.1±0.5 ppb, while a blank of 5-14 ppb was interpreted as indicator of leakage. I think this issues need to be clearly demonstrated so that superior performance can be claimed.

*The blank offset, which is calculated from the mean of the 3-4 blank results, reflects any errors by contaminants, leaks, or different GC conditions. The exact cause of the blank offset is not clear, but*

*the important thing is that the 4 blank results agree well each other, yielding the intra-day standard error of the mean of 2.0±1.0 ppb. This implies that the daily offsets of the 10 sample flasks are rather systematic although the inter-day blank correction varies 5-14 ppb. The robustness of our final results was proven by reanalysis of duplicates at adjacent depths (< 10 cm) 8-80 days after the first analysis. The absolute difference of mean of duplicates between the time intervals was 1.9 ppb on average (see the table above).*

*System leakage is unlikely. We carried out system leak check as a daily routine before starting gas extraction. If any pressure increase in $10^{-4}$ Torr scale for 30 seconds is detected, the experiment was stopped to figure out the leakage.*

Further issues that could be clarified include the uncertainty of the standard air (available for each NOAA04 cylinder), how the linearity of the system is controlled (additional air standards covering the analytical $CH_4$ range?) and what reason the authors base their decision on to subtract 3 ppb from the OSU data instead of adding 3 ppb to the SNU data or to take the 3 ppb as temporal signal? If the measurement uncertainty claimed by both institutes is realistic, both data should be on the NOAA04 scale and therefore be in close agreement. I think this is in the order of expected disagreement, but I feel the manuscript is stronger if these issues are clearly addressed.

*We appreciate Referee #1 for his pin-point review. We compared the Siple Dome $CH_4$ data measured at OSU (Ahn et al., 2014) to the WAIS Divide $CH_4$ records to see which one is more reasonable between adding 3 ppb to SNU data and subtracting from OSU data. To do this, the WAIS Divide $CH_4$ data points were linearly interpolated to the ages of the Siple Dome $CH_4$ records. The comparison of the mean $CH_4$ difference between the two data set shows that the OSU Siple Dome data itself fit better than the 3 ppb subtracted data, which were previously used for the early Holocene $CH_4$ composite in our discussion paper.*

*It seems that the 3 ppb offset is due to different correction for gas solubility effect. Unlikely to Mitchell et al. (2011), we calculated the expected gas enrichment by assuming that equilibrium state is achieved between air-water during the melting process. In OSU, the gas solubility effect is empirically estimated by measuring $CH_4$ mixing ratio from the air that is re-dissolved after melt-refreeze process (Mitchell et al., 2011). The directly-measured OSU correction method seems more realistic, but the uncertainty is large (~10%) due to small amount of air. After applying the OSU solubility correction scheme, the average difference between SNU and OSU reduces to less than 1 ppb. Therefore, for the better fitted composite record, we applied the OSU correction method to SNU data set.*

3)      $CH_4$ data: The authors mention 295 measurements. However, Figure 1 seems to show a much smaller number. If the displayed measurements are averages of duplicates or if other measurements are not displayed in Figure 1, the number would have to be revised.

*The number will be corrected as below. Basically we made duplicates for all depths, and the average of the replicates was used for the data.*

*"We measured 295 samples from 156 depth intervals ranging from 518.87 to 718.83 m of Siple Dome ice core (including 8 replicates and 13 rejections, all duplicated), covering from 8.36 to 20.25 kyr BP after synchronizing to GICC05 scale. Analytical uncertainty of each point is deduced from standard error of the mean of duplicate samples. Figure 1 presents only the data points during the early Holocene period (119 depths, 518.87 – 623.38 m, 8.36 – 11.71 kyr BP) that we discuss in this study*

*Data older than 11.71 kyr BP have lower time resolution and are unevenly spaced, thus those records have not been used in this paper."*

Otherwise, please clarify. How long is the overlapping period between OSU and SNU data? Maybe Figure 1 could show this in a detailed Figure?

*We will add an enlarged figure in Figure 1 (See Figure R1). The overlapping period is from ~8.4 to ~9.1 ka. To make the SDMA CH₄ composite, we used the OSU data from 7.7 to 8.5 ka, due to higher mean temporal resolution of the OSU data.*

Figure 1 has a reference in the captions (Brook and Sowers, 2016) that is new to me and that I cannot find in the reference list.

*Reference will be modified.*

The authors may or may not consider to show data from previous publications (e.g. Brook et al., 2000, Flueckiger et al., 2002) to highlight the superiority of their temporal resolution.

*Figure 1 will be modified as Figure R1 that includes the data from Brook et al. (2000) and Flueckiger et al. (2002) (see below).*

[Figure]

*Figure R1. CH₄ reconstructions during the early Holocene. Top: new high-resolution Siple Dome CH₄ data (black, this study) compared with previous records from Taylor Dome (orange, Brook et al., 2000), EPICA Dome C (grey, Loulergue et al., 2008), and Talos Dome (purple, Buiron et al., 2012).*

*Bottom: Siple Dome $CH_4$ records measured in OSU (blue, Ahn et al., 2014) and SNU (red, this study). Siple Dome composite (black line) is plotted with WAIS Divide discrete (dark yellow, WAIS members, 2015) and continuous technique (green, Rhodes et al., 2015). Inset: Enlarged plot showing overlapped interval between OSU and SNU Siple Dome data. Note that this figure may be subject to change.*

4)      IPD: IPD is a powerful concept, but very one has to be very careful in its reconstruction and interpretation. The authors mention the potential for ill-calculated IPDs based on errors in the gas age scale of the $CH_4$ records. Therefore, the authors developed a tool to synchronize the $CH_4$ records, which I think is a very good approach! However, it is not clear to me why the authors chose to calculate the IPD based on the Siple Dome data? My concerns have several reasons:

i)      The authors state themselves that the histories of gas enclosure is more similar between NEEM and WAIS. The authors state that, based on just this fact, the amplitude of $CH_4$ variations is 10-20 ppb larger in the WAIS than in the Siple Dome record. Therefore, I understand that both the NEEM and the Siple Dome $CH_4$ records are altered by physical processes during gas enclosure that are different for each core. I understand that a dampened amplitude in the Siple Dome $CH_4$ record would create a IPD variation. Ideally, gas enclosure effects should be identical in both records so that they would cancel. Since the WAIS record is in that sense more similar to the NEEM record than the Siple Dome record is, I would suggest to use the WAIS record for the IPD reconstruction.

ii)      The WAIS record is of even higher temporal resolution than the Siple Dome record. I would expect that the IPD reconstruction based on NEEM and WAIS would be more robust.

iii)      The comparison of the $CH_4$ records from Siple Dome and WAIS in Figure 1 shows two periods (~10.5 and ~10.7 ka) where Siple Dome $CH_4$ exceeds WAIS $CH_4$ by up to ~20 ppb. During this period, the smoothed $CH_4$ variations (Figure 2) also show an ice core specific pattern of disagreement. Around 11 ka, the pattern of agreement is different. Here, the continuous $CH_4$ record from WAIS seems to agree better with $CH_4$ from Siple Dome, while the GC-FID record from WAIS contains a number of samples that exceed the former records by ~20 ppb. The difference between the records during these times exceeds the stated measurement uncertainty by far. It is also important that the difference seems to be in the order of the presented IPD variability.

All of the above mentioned reasons directly impact on the IPD reconstructions. The choice of the authors to use their Siple Dome data for the IPD reconstruction is justified and understandable. However, I fear that the interpretation is sensitive to the choice of $CH_4$ record so that this choice could impact on the IPD result for the above mentioned reasons. Therefore, I suggest to calculate the IPD also using both WAIS records as three independent sets of IPD reconstructions as a sensitivity test. This will make the interpretation of reconstructed IPDs more robust and will furthermore give valuable insights into the IPD technique on high temporal resolution records for future studies. Because this concerns one of the main outcomes of this manuscript, I would consider this essential.

*We appreciate the Referee #1 for her/his useful and reasonable comment on IPD reconstruction. Calculating IPD from different ice core data would draw more objective conclusion. We will provide alternative IPD reconstructions from various records (NEEM discrete, NEEM continuous, WAIS discrete, WAIS continuous, and Siple Dome discrete data) to test robustness of early Holocene IPD trend.*

*As mentioned in comment iii) above, the Siple- and WAIS ice core records do not agree at some period, and this offset could lead erroneous IPD change. However, here we consider Siple data show more reliable result by following reasons. First, it was shown that SNU Siple Dome discrete data (this study) have a good agreement with OSU Siple Dome discrete data (Ahn et al., 2014) during the early Holocene interval, while PSU (Penn State University) WAIS discrete data that covers most of the early Holocene period show offset up to ~9.9 ppb to OSU data (Rhodes et al., 2015). Second, PSU WAIS discrete data show a pooled standard deviation (for depth-adjacent samples) of 7.3 ppb (1sigma), which is larger than SNU Siple data. Thus we will weigh our interpretation on IPDs calculated with Siple data than those from WAIS discrete record. In addition, NEEM continuous- and discrete data do not agree well in some intervals (Figure R2), even though NEEM continuous data were calibrated against to discrete measurements carried out at OSU (Chappellaz et al., 2013). Further, the NEEM continuous record is not exactly "continuous", that may introduce uncertainty into synchronization. Hence, here we will regard the NEEM discrete, Siple Dome discrete, and WAIS Divide continuous records as more reliable ones than the others during the early Holocene period (Black and Green curves in Figure R2).*

*Resulting IPD curve from NEEM discrete and WAIS continuous (Green) shows long-term increase from the onset of Holocene to ~9.9 kyr BP. This indicates that contribution from boreal sources increased during ~11.5 to 9.9 kyr BP, which is consistent with increase of northern extratropical temperatures and thermokarst lake $CH_4$ emissions. The revised 3-box model results (Table S1) show elevated emission from N-box and slight decline in T-box. However, since a small IPD increase during ~10.8-11.2 kyr BP observed in Siple Dome IPD is not supported by alternative IPDs, it remains unclear the short-term IPD change during this time period. Therefore, we will modify descriptions and findings on early Holocene IPD trend. Also we address the inconsistency of ice core records for the period older than ~10.3 kyr BP, which makes the alternative IPDs different from each other.*

*Table S1. 3-box source distribution model results of tropical (green, T) and boreal (red, N) boxes and boreal emission fraction (N/(T+N+S)) compared with previous results. Errors denote 95% confidence interval.*

| Ref. | N box | T box | N/(N+T+S) ratio |
|---|---|---|---|
| (ka) | (Tg yr$^{-1}$) | | (%) |
| Brook et al., 2000 (9.5-11.5 ka) | 64 ± 5 | 123 ± 8 | 32 ± 3 |
| Chappellaz et al., 1997 (9.5-11.5 ka) | 66 ± 8 | 120 ± 9 | 33 ± 3 |
| This study (9.5-11.5  ka) | 66 ± 4  | 120 ± 4  | 33 ± 2  |
| This study (11.5 ka) | 57 ± 6 | 119 ± 11 | 30 ± 4 |
| This study (9.9 ka) | 70 ± 4  | 115 ± 4  | 35 ± 2  |

[Figure]

*Figure R2. Inter-polar difference (IPD) reconstructions. Top: high resolution $CH_4$ records from Greenland and Antarctica, synchronized to NEEM gas age scale by Monte Carlo procedure. Middle: Millennial-scale IPD trends derived from various pairs of data set. Shaded area indicates 95% significance interval. Bottom: Proxy-based temperature reconstruction for northern mid to high latitude and boreal $CH_4$ emission from northern thermokarst lakes. Note that this figure may be subject to change.*

[Figure]

*Figure R3. Same as Figure R2, but shown are IPDs using NEEM discrete, Siple discrete, and WAIS Divide continuous data. Note that this figure may be subject to change.*

5)        Consideration of significant publications: *i*) The authors claim that no correlation between CH₄ and tropical monsoon signals has been reported on shorter time scales, however, I feel this is not accurate. Both Cruz et al., (2005, Nature, Vol, 434, p. 63-65) and Sperlich et al., (2015, Global Biogeochemical Cycles, 29) have related CH₄ and $\delta^{13}$C-CH₄ records to South American speleothem records, respectively. Both publications would principally support the interpretation of this study. *ii*) The authors discuss that their IPD reconstruction suggests increasing boreal source fractions during the early Holocene and support their finding with studies on boreal wetland dynamics. However, their finding of increased boreal source fractions is in line with the interpretation of $\delta^{13}$C-CH₄ data by Fischer et al., (2008) and Sowers, (2010). Again, both publications would principally support the interpretation of this study while Sowers (2010) had the same finding for the early Holocene previously. *iii*) The authors stated that there is currently no ENSO reconstruction for the early Holocene, even though a range of them exist (e.g. Z. Liu et al., 2014, Nature, Vol. 515, p. 550-553, Clement et al., 1999, Paleoceanography, Vol. 14, p. 441-456).

*We checked the suggested publications and take them into consideration for further developing discussions.*

*i)        We will modify the statement as below:*

*"However, no direct correlation between $CH_4$ and tropical monsoon signals has been reported for the early Holocene. Abrupt increases in stalagmite $\delta^{18}O$ record from subtropical Brazil (Botuvera Cave) are found to be coincided with rapid increase in $CH_4$ during the last glacial period and Younger Dryas events (Cruz et al., 2005). This indicates that the monsoon precipitation in subtropical South America was reduced when atmospheric $CH_4$ increased, which implies northward migration of monsoon rain belt and increasing $CH_4$ emissions in tropics (Cruz et al., 2005). A recent study using $\delta^{13}C$-$CH_4$ analysis from Greenland ice cores reported that the sharp $CH_4$ increase at Greenland Interstadial 21.2 (~85 kyr BP, ~150-year duration) is concurrent with intensifying precipitation in Amazonian and Asian wetlands (Sperlich et al., 2015)."*

ii)    *We will cite the suggested publications and modified the relevant paragraph as below:*

*"The IPD estimates from Siple Dome and NEEM discrete $CH_4$ records are generally consistent with the previous results from $CH_4$ isotopic ratio analysis. Fischer et al. (2008) found that increasing boreal source contribution is required to explain the decreasing $\delta^{13}C$-$CH_4$ during the Younger Dryas - Preboreal Holocene transition. Sowers (2010) disclosed the $CH_4$ isotopic ratio covering the entire Holocene that displays a gradual decrease of $\delta^{13}C$-$CH_4$ by ~2 ‰ from 10.5 to 4 kyr BP, which they attributed it to progressive expansion of NH high latitude sources. Using an isotopic mass balance model, they proposed an additional emission of 19.6 Tg yr-1 from boreal thermokarst lakes to explain such amount of $d^{13}C$-$CH_4$ depletion. According to $\delta^{13}C$-$CH_4$ data in Sowers (2010), we observe ~0.8 ‰ decrease in $\delta^{13}C$-$CH_4$ from 11.3 to 9.0 kyr BP. It corresponds to ~8.3 Tg yr$^{-1}$ increase of the Arctic lake emissions, which agrees with our three-box model results and thermokarst lake emission model result by Water Anthony et al. (2014) (see below). The boreal source expansion during the early Holocene is further supported by proxy-based temperature reconstructions that indicate a gradual warming in northern high latitude region (30N-90N) until ~9.6 ka, while tropical temperature remains stable (Marcott et al., 2013). The climate warming in northern high latitudes caused ice sheet retreat (e.g., Dyke, 2004) and enhanced $CH_4$ emission from boreal permafrost by forming new wetlands in mid- to high latitudes (e.g., Gorham et al., 2007; Yu et al., 2013) and accelerating microbial decomposition of organic materials (e.g., Christensen et al., 2004; Schuur et al., 2015). Thermokarst lakes created by thawing ice wedges and ground ices in Alaskan- and Siberian permafrost are suggested as a source of $CH_4$ (e.g., Walter et al., 2006, 2007; Brosius et al., 2012). Indeed, the increased boreal $CH_4$ emission of $9.7 \pm 3.5$ ($1\sigma$) Tg yr$^{-1}$ during 11.5 – 9.0 kyr BP is in similar order of the $CH_4$ release of 8.2 Tg yr$^{-1}$ from thermokarst lake reported by Walter Anthony et al. (2014). However, it should be noted that the $CH_4$ release estimates from the thermokarst lakes are based on present-day $CH_4$ flux measurements in Siberian- and Alaskan lakes and that 9 Tg yr$^{-1}$ is a small change in the budget that could be driven by conventional northern $CH_4$ emission. A recent study also argued a possibility of underestimation of such $CH_4$ emission measurements (Wik et al., 2016)."*

iii)   *We appreciated the suggestion, but unfortunately, the mentioned publications (Liu et al., 2014; Clement et al., 1999) are dealing with climate modelling result, not proxy-based reconstructions. Instead, we will cite Kirby et al. (2010) for PDO, and Moy et al. (2002) and Rodbell et al. (1999) for Holocene ENSO reconstructions. According to Holocene ENSO activity reconstructions by Moy et al. (2002), weak- or no ENSO event was recorded during the early Holocene until around 7 kyr BP. Nevertheless, it shows a slight (up to 3 events per 100 years)*

*increase in warm ENSO event numbers during 10.4-10.1 kyr BP, where abrupt CH₄ drop is observed without significant changes in ITCZ and NH monsoon intensities (Figure R4). Recent studies have shown that warm ENSO event leads drying in low latitude wetlands (e.g., Dai and Wigley, 2000; Lyon and Barnston, 2005), however it is currently uncertain if this magnitude of change in ENSO frequency caused significant drying in wetlands and CH₄ emissions. In the other hand, by using sediment grain size analysis, Kirby et al. (2010) found PDO-related drying intervals in North America during 9.5 – 9.1, 8.9 – 8.6, and 8.3 – 7.8 kyr BP. The overlap of CH₄ minima at around 8.2 and 9.3 kyr BP is compelling because Mitchell et al. (2011) have reported a significant positive correlation between CH₄ and PDO variability for the late Holocene.*

6)      Chosen data filtering technique: I suggest to provide more information why a 250-year window width was used. What is the effect of other window widths on the data you use and on your resulting interpretation? The authors state on p5L2-5 that the 250 year window was also used in other studies. However, that is not necessarily a satisfying justification. The window width should be carefully chosen in dependence on the time scales you are investigating.

*Power spectrum (REDFIT, Schulze and Mudelsee, 2002) of Siple Dome CH₄ data indicates moderate (over 90% significance level) powers at ~1340, 401, 309 and 96-year period. Considering ~42 years of gas age distribution of Siple Dome (Ahn et al., 2014), it is not reliable to study centennial scale variability. Thus we applied 250-year window to smooth out any high frequency component having shorter period than 309 years. We will add above details to that paragraph.*

7)      Sometimes, I have trouble to understand the point the authors intend to make, e.g. p1, 12–18; p5, 18–29; p6, 25–p7, 5; p10, 6–12, p22, 1-2. Please consider re-wording.

*We will thoroughly revise the mentioned sentences.*

8)      I have difficulties to follow the discussion and the display of CH₄ and other records. For example, the authors discuss why there is agreement/disagreement between some records within the uncertainty of the age model. However, I find it tough to see this in Figure 2. For example, the CH₄ minima are highlighted with yellow bars. During 9.4 and 10.2 ka, the yellow bars include both local minima and local maxima of the other records, e.g. of the Cariaco Basin record, of Δε, of Dongge Cave. Other local extrema, e.g. in the Cariaco Basin record have no correspondence in CH₄ during 8.5-8.7 ka or 10.5 ka, which is not mentioned at all. Therefore, I feel this discussion needs to be further developed to provide more guidance to the reader. Is [10]Be really important here? Sometimes it seems to correlate, other times it is anti-correlated. Could figure clarity increase without it?

*Not every single variation of the Cariaco Basin reflectance record corresponds to abrupt CH₄ change, because the Cariaco Basin record could also have local signal. What we wanted to show in this Figure is that abrupt cooling occurred around Greenland changes tropical rain belts and hence CH₄ emission. [10]Be and IRD proxies were included to discuss trigger of abrupt Greenland cooling, but data resolution and dating uncertainty (±100 to 150 years, Bond et al., 2001) prevent us from drawing rigorous conclusion. By taking into consideration with the comment #12 (Figure R4), we will modify the Figure 2 and relevant discussions.*

[Figure]

*Figure R4. Millennial scale climate variability. All proxies present here were smoothed by 250-year running average and detrended by high-pass filter with 1/1800-year window. (a) Siple Dome CH₄ (red, this study), Greenland ¹⁰Be (dark yellow, Finkel and Nishizumii, 1997), North Atlantic IRD stack (grey, Bond et al., 2001). Also shown are WAIS Divide CH₄ data by discrete (cyan, Buizert et al., 2015) and continuous (yellow green, Rhodes et al., 2015) technique. (b) NGRIP stable water isotope ratio (blue, Rasmussen et al., 2006) and Cariaco Basin reflectance (orange, Deplazes et al., 2013). (c) Qunf Cave speleothem oxygen isotope (Fleitmann et al., 2007). (d) Dongge Cave speleothem oxygen isotope (green, Dykoski et al., 2005; dark yellow, Wang et al., 2005). (e) Gulf of Guinea planktonic Ba/Ca ratio (Weldeab et al., 2007). (f) Lapa Grande Cave speleothem oxygen isotope (purple, Strikis et al., 2011). Age tie-points used to adjust Siple Dome and WAIS Divide CH₄ data to GICC05 scale are marked in black triangles. This figure may be subject to change.*

9)      Figure 2: Presented is CH₄ anomaly. I don't see an advantage of anomaly over CH₄ mole fractions. Also, how is anomaly of 0 defined for each record?

*We refer "anomaly" in Figure 2 as detrended time series after filtered with highpass window. We will use "detrended data" instead of "anomaly" to clarify it.*

10)     Structure: I would suggest to avoid three levels, e.g. 3.1.1 and 3.1.2 but make 3. Millennial scale variability, 4. Latitudinal distribution…. to keep structure with max. two levels.

*We will simplify the structure in two levels.*

11)     In many places of the manuscript, the author review literature, e.g. on pattern of climate teleconnections, for which they allow extensive text sections. While I think it is important to review in such detail, I feel the authors could improve their discussion of how they think this is linked/relevant to their $CH_4$ interpretation. A good example for this is the entire section 3.1.3.  I would like to encourage the authors to consider this throughout the entire manuscript, even though this probably means either adding to, or re-writing many sections of the manuscript.

*We appreciate Referee #2 for pointing out this aspect and we agree with it. We will insert below paragraph and re-structure the 3.1.2. section. Moreover, other sections will be thoroughly revised and re-written.*

*"From the previous publications it can be inferred that low solar activity might be a cause of tropical rainfall change via abrupt cooling in North Atlantic and southward migration of ITCZ (e.g., Broccoli et al., 2006; Renssen et al., 2006; Jiang et al., 2015). Meanwhile, shifting to El Nino-like SST condition was suggested as another mechanism that changes tropical rainfall pattern (Marchitto et al., 2010). According to modern atmospheric observation, El Niño condition leads drying in low latitude wetlands in Africa, Asia, and America (e.g., Dai and Wigley, 2000; Lyon and Barnston, 2005), which reduces tropical $CH_4$ emissions. Thus we can hypothesize that both the ITCZ migration and El Niño-like SST change affected the tropical surface hydrology and $CH_4$ emission."*

12)     Understanding the variability in tropical wetlands is crucial for the understanding of $CH_4$ source regions and tropical $CH_4$ fractions. (The same rule applies for boreal wetlands.) The authors fully acknowledge this throughout the manuscript. However, I note that the authors exclusively focus their interpretation on Asian/Indian monsoon systems. It has been described previously that the African monsoon system and wetland extension changed tremendously throughout the Holocene (e.g. Sahara region etc). There are also several publications on South American monsoon systems besides the Cariaco Basin reflectance, which I understand the authors only use as proxy for ITCZ migration, but not for their interpretation of hydrological changes in South American wetlands. Including further records (e.g. Cruz 2005) might allow for a more comprehensive evaluation of hydrological changes. Based on $\delta^{13}C\text{-}CH_4$ data, the South American monsoon system has recently been suggested to be a controlling factor in rapid $CH_4$ changes leading up to DO21 (Sperlich 2015). I would strongly recommend to either include a complete representation of tropical wetlands or to discuss why you think monsoon systems other than the Asian/Indian are not relevant.

*As mentioned by the Referee #1, it is possible that the monsoon system of other regions could play a role in $CH_4$ change. However, considering that mean position of ITCZ was moved to northward than glacial condition (e.g., Deplazes et al., 2013), the rainfall and $CH_4$ emission of Asian/Indian monsoon regions should be stronger than southern hemisphere monsoon regions, for example, South America and Africa. This makes boundary condition different from those arguing that Southern Hemisphere emission leads abrupt $CH_4$ increase during Heinrich Stadial 1, 2, 4, and 5 (Rhodes et al., 2015) or DO21 (Sperlich et al., 2015). Both studies are dealing with abrupt $CH_4$ change under glacial condition. In the meanwhile, one of main conclusion of our paper is that abrupt cooling in Greenland lead ITCZ mean position change and tropical $CH_4$ emission. Therefore, it seems sufficient to support the idea with the Asian/Indian monsoon records and Cariaco Basin reflectance data.*

*However, for the interested readers it would be good to insert additional discussions and/or supplement figure on changing other monsoon systems at the same time. Lapa Grande Cave (Eastern*

*Brazil, 14°25'22"S) records demonstrate clear δ¹⁸O depletion at 8.2 and 9.2 kyr BP (Strikis et al., 2011), indicating that ITCZ rain belts temporarily migrated southward and induced wet condition over eastern Amazonia region. This evidence agrees well with Cariaco Basin rainfall reconstructions and does not go back beyond ~10.2 kyr BP, therefore we thought that adding the South American monsoon proxy would not improve our conclusions. The monsoon record of northern Peru (El Condor and Cueva del diamante, Cheng et al., 2013) and southern Brazil (Botuvera, 27°13'24"S, Cruz et al., 2005) are not highly resolved enough to see abrupt changes. Otherwise, the reconstructions of West African monsoon (Gulf of Guinea, Ba/Ca ratio of planktonic foraminifera, Weldeab et al., 2007) and Indian monsoon entire early Holocene (Qunf Cave, Fleitmann et al., 2007) track well the millennial scale $CH_4$ minima. Furthermore, Australian-Indonesian monsoon rainfall records (not shown) from Borneo (Partin et al., 2007) and Liang Luar (Griffiths et al., 2009) do not show clear evidence of abrupt change that coincides with Greenland cooling and $CH_4$ drop. This may reflect that the rainfall in tropical western Pacific region was affected by both northern- and southern hemispheric climate change (Griffiths et al., 2009; Partin et al., 2007).*

13)     Data availability: I understand that Copernicus has developed a new policy for authors to provide either descriptions on data access, or to provide the data through international data-bases or supplementary information. Copernicus requires a dedicated section that describes this in detail, which the authors might want to consider during their revisions.

*Thanks for this information. We will describe how to access our new data in a dedicated section of revised manuscript.*

**Specific comments**

The manuscript may be subject to considerable re-writing. Therefore, the specific comments will not include comments on grammar, wording or writing that might be subject to change. Since I am not a native English language speaker myself, I am not sure to what extend my comments would help to make it better or worse. Please understand suggested re-formulations as suggestions, only.

p1L1: Understanding processes controlling atmospheric methane

*The sentence will be changed as below:*

*"Understanding  processes controlling atmospheric methane ($CH_4$) mixing ratio is crucial to predict and mitigate the future climate change.*

p2L2: reference Daniau et al., is it possible to provide a reference on palaeo-fire or a reference that is more specific on pyrogenic gas emissions?

*Carbon monoxide (CO) and ethane ($C_2H_6$) can be used as paleo-fire indicator, but currently there is no available CO and $C_2H_6$ reconstruction covering the early Holocene.*

p2L5: sink strength and light availability? e.g. polar winter

*We will add more details on methane sink process, especially on photochemical oxidation of $CH_4$.*

p2L16: Lisiecki and Raymo 2005, though this reference is not on $CH_4$, general I think a reference on $CH_4$ and Northern Hemisphere temperature would be useful here

*We will change the citation as below and added the reference:*

*(e.g., Brook et al., 2000; Chappellaz et al., 1993; Huber et al., 2006; Loulergue et al., 2008).*

*Huber, C., Leuenberger, M., Spahni, R., Fluckiger, J., Schwander, J., Stocker, T. F., Johnsen, S., Landais, A., and Jouzel, J.: Isotope calibrated Greenland temperature record over Marine Isotope Stage 3 and its relationship to $CH_4$, Earth Planet. Sci. Lett., 243, 504-519, 2006.*

p2L18: See comment 5) in general comments

*See Author Comment to general comment #5.*

p2L18-19: too weak. The correlation between $CH_4$ and NH temperature ($\delta^{18}$O-ice) is well established

*It is true for longer time scales. The $CH_4 - \delta^{18}O_{ice}$ of Greenland has yet been revealed in shorter time scales during the early Holocene period.*

p3L15: here and everywhere else: It is recommended to restrict the use of "concentrations" to public debate but to use "mole fraction" or "mixing ratio" in scientific literature (WMO, recommendations of GGMT experts)

*This will be updated as suggested.*

p3L17: here and everywhere else, $\delta^n$X, the $\delta$ is supposed to be *italicised*, same for $\Delta$ $\Delta$ (Coplen 2011, DOI: 10.1002/rcm.5129)

*This will be updated as suggested.*

p3L27: there is still some ice left in Greenland

*We will change the sentence as below:*

*" It should be noted that environmental boundary conditions of the early Holocene were not identical to those of the late Holocene. The global sea level was rising throughout the early Holocene and there is still some ice left in Greenland."*

p3L30: ensure stated, used and displayed sample numbers agree, give age interval with depth

*Please refer to our response to General Comment #3.*

p3L32: (NICL, city, state, country), (SNU, city, state, country)

*This will be updated as below:*

*"… National Ice Core Laboratory (NICL, Denver, Colorado, USA), and shipped to Seoul National University (SNU, Seoul, South Korea) …"*

p4L3: "are described in…", referring to a paper that is currently in preparation as XYZ et al., (in prep.) seems strange as it is not useful to look it up. "A manuscript that describes the method in detail is currently in preparation."

*We will change the sentence as below:*

*" Further details of analytical method is described in another manuscript that is currently in preparation."*

p4L7: was the standard air added before or after the bubble-free ice was melted?

*The standard air was injected before melting the bubble-free ice. We revised the wording for clarity.*

p4L7: "traditional" melt-refreeze seems misleading to me, traditional can be left out

*We will delete the word "traditional".*

p4L7-17: see point 2) in general comments

*See Author Comment to general comment #2.*

p4L17: provide reference how you calculated gravitational fractionation

*Relevant citation and reference will be added: Craig et al., 1988*

*Craig, H., Horibe, Y., and Sowers, T.: Gravitational separation of gases and isotopes in polar ice caps, Science, 242, 1675-1678, 1988.*

p4L19: provide reference for GICC05 time scale

*Reference will be added: Rasmussen et al. (2006)*

p4L25: "one of the high resolution data sets" sounds vague and strange to me, where do you draw the line between high and low resolution? The temporal resolution of your data is higher than some but lower than both WAIS records. "It has the currently second/third highest temporal resolution of Antarctic CH₄ records covering the early Holocene."

*The sentence will be revised as below:*

*"Our new Siple Dome CH₄  data has the currently third highest temporal resolution of Antarctic CH₄ records covering  the early Holocene after the WAIS Divide continuous (~2 years, Rhodes et al., 2015) and discrete (~20 years, WAIS Divide members, 2015) records."*

p4L26: develop a more complete representation of analytical uncertainty

*Please refer to our response to general comment #2, as well as our response to the Referee #2.*

p4L30: describe the overlapping interval and describe why you think the OSU record should be adjusted to match the SNU data. Why not the other way around, why not accepted as real difference? Both should be on NOAA04?

*We attribute the SNU-OSU offset to different correction method for solubility effect. After applying revised solubility correction used in OSU (Mitchell et al., 2011), the mean offset reduces 0.6 ppb which lies well within analytical uncertainty of both institutes.*

p5L3: this comparison example only makes sense if you look at variations on similar time scales. Otherwise the argument that you use the same filter as has been applied for the WAIS record is not valuable, but could be misleading.

*Here we hesitate to remove the comparison. As we describe in this sentence and Figure 2, to ensure robustness of millennial scale variability from other Antarctic ice core record is important. We used filtered data because both data set have different time resolution, and they show different fluctuations in short time scales as shown in Figure 1. We will revise the wording not to mislead.*

p5L14: add references that show anthropogenic signal in LPIH $CH_4$, e.g. Ferretti 2005, Mischler 2009, Sapart 2012)

*We will change the sentence with relevant citations as below:*

*"Mitchell et al. (2011) found no significant correlation with Greenland climate in multi-decadal time scale during the late pre-industrial Holocene (LPIH), possibly due to growing anthropogenic emissions (e.g., Ferretti et al., 2005; Mischler et al., 2009; Mitchell et al., 2013; Sapart et al., 2012)."*

p5L16: "even though this conclusion is less robust as there are no age tie-points…"

*The sentence will be modified as suggested:*

*P5L14-18: "In contrast, we observe a moderate positive correlation between the Siple Dome $CH_4$ and NGRIP $\delta^{18}O_{ice}$ during the early Holocene, which implies that natural $CH_4$ budget is closely connected with Greenland climate in millennial timescales, even though this conclusion is less robust as there is no age tie-point between the 8.2 ka episode and the Preboreal oscillation (Fig. S1)."*

p5L21-22: shift this sentence to after the following sentence to keep logical flow from NH to tropics

*We agree with the suggestion. The paragraph will be changes as suggested.*

p5L25-29: describe the meaning for $CH_4$, develop the discussion towards $CH_4$, what does a ITCZ shift mean for South American $CH_4$ source regions?

*South American monsoon proxy shows concurrent intensification at similar timings of ~8.2 and 9.3 ka $CH_4$ drop (Strikis et al., 2011), where precipitations in Cariaco Basin and other NH monsoon regions decreased. For older period it is difficult to draw robust conclusion due to lack of high resolution data at the timing of preparing this paper. The southward ITCZ migration may lead reduction in NH wetland emission and enhanced in SH. However, given the orbital parameters that show maximum summer insolation in NH while minimum in SH during the early Holocene, it can be inferred that contribution of SH wetland emission was relatively low and cancelled by reduction of NH emission. We will update the paragraph as below:*

*" The $^{18}O$ enrichments of Asian (Dongge) and Indian (Hoti and Qunf) cave stalagmites occurred at similar timing with abrupt cooling in Greenland, which indicate the reduction of monsoonal rainfall at northern tropical wetlands. The speleothem records from Chinese and Oman caves seem to lag by ~100-200 years after the $CH_4$ change at ~9.3 ka, but this lies within chronological uncertainties of ~200-400 years at around 9.0 ka (Dykoski et al., 2005; Fleitmann et al., 2007). Moreover, sediment Ba/Ca ratio from Gulf of Guinea demonstrates concurrent decrease of west African monsoon (Weldeab et al., 2007). These evidences indicate that precipitation over the major wetland area was reduced and in turn it would lower the wetland $CH_4$ emissions in NH. In the*

*meanwhile, an inverse relationship is observed from the Eastern Brazilian speleothem data (Lapa Grande Cave, Strikis et al., 2011) that demonstrate the increasing of precipitation at the time of abrupt $CH_4$ drop occurred as a result of southward migration of ITCZ. Considering the orbital parameter that shows maximum summer insolation in NH while minimum in SH during the early Holocene, it can be inferred that contribution of SH wetland emission was relatively low and cancelled by reduction of NH emission."*

p6L4: there are other monsoon systems that Asian/Indian that should be considered

*See our response to general comment #12.*

p6L10: the monsoon intensity change. (delete Asian, include other monsoon systems)

*We will change the sentence as below:*

*"Given the weak reduction of precipitation over the Asian, Indian, and African monsoon regions (Figure R4),  it may imply $CH_4$  drop was controlled by other processes than the  monsoon intensity change."*

p6L20: even though the Cariaco Basin record is shown, it is presented only as indicator for ITCZ migration, without direct connection to $CH_4$. I feel the assumed passiveness of South American $CH_4$ source regions during the study period might not be a natural assumption and should be explained.

*Considering the orbital configurations that show maximum summer insolation in NH while minimum in SH, contribution of SH wetland emission had to be reduced during the early Holocene.*

p6L23-p7L5: describe relevance for $CH_4$, what is the $CH_4$ controlling process chain? A sentence that says "the proxies show this and that which could explain the increase/decrease in $CH_4$ during time period XY".

*We will explain more details on proxies and its relevance to $CH_4$ change. Adding following sentence at the end of the paragraph will be more explained: "Above evidences indicate that the early Holocene $CH_4$ minima were triggered by anomalous low solar activity, but future study is warranted to draw more conclusive result."*

*The entire paragraph will be revised as below:*

*"Bond et al. (1997) reported four large ice-rafted debris (IRD) drifts occurred at ~8.1, 9.4, 10.3 and 11.1 ka caused by surface cooling of North Atlantic Ocean. They found that the ocean surface cooling and the IRD events are closely related to cooling over the Greenland. Figure 2 shows that each IRD event (maxima in hematite stained grain) occurred concurrently with minima of NGRIP $\delta^{18}O_{ice}$ record within age uncertainty. Then the Greenland cooling leads southward shift of ITCZ and in turn it changes wetland $CH_4$ emission. Bond et al. (2001) found that IRD maxima during the Holocene coincide with solar activity minima. The authors suggested that solar forcing could affect the climate change around the North Atlantic Ocean (and Greenland), through amplification by changes in sea ice and/or deep water formation. However, the forcing mechanism of solar activity on the North Atlantic and global climate is not well understood during the early Holocene. Renssen et al. (2006) suggested that low solar activity (in terms of total solar irradiance) can induce sea-ice expansion*

*around the Nordic Seas and weakening of deep water formation and cooling in North Atlantic region. Nevertheless, the anti-correlation between solar forcing and sea-ice expansion (and hence deep water formation weakening) is not strong during the early Holocene due to relatively warm climate conditions. Jiang et al. (2015) also found a negative correlation between North Atlantic SST and solar forcing proxies ($^{14}C$ and $^{10}Be$), which is statistically significant for the last 4000 years, while the correlation disappeared during the mid- and early Holocene. They hypothesized that climate sensitivity to solar forcing is high for cooler climate. Above evidences indicate that the early Holocene $CH_4$ minima were triggered by anomalous low solar activity, but future study is warranted to draw more conclusive result."*

p7L8-9: you could add Cruz et al., 2005 to the reference list, as they discussed the interplay of solar radiation, monsoon intensity and $CH_4$ mole fractions

*Here we hesitate to cite Cruz et al. (2005) here because the time scale they are dealing with is quite different.*

p7L16-17: see above comment regarding references on ENSO variability during Holocene period

*Please refer to our response to general comment #5. We will add further discussions on relationship with PDO- and ENSO proxies for the early Holocene interval.*

p7L23: provide temporal resolution of NEEM record, 1 sample in how many years?

*We will provide information on temporal resolution of each record as below:*

*"The NEEM $CH_4$ record is chosen as a reference because the mean time resolution is ~11 years, which is higher than Siple Dome data (~26 years) during the early Holocene."*

p7L24: consider IPD reconstructions with $CH_4$ records from WAIS

*We will provide alternative IPD reconstructions from more reliable records; NEEM discrete, WAIS continuous, and Siple Dome discrete data. Also refer to our response to Referee #2.*

p7L25-30: NICE approach!

p8L2: …show an increase by XYZ ppb from…

*We will add the magnitude of increment as follows:*

*"Our results show an increase by about 8 ppb from ~10.7 ka to ~9.9 ka, which was not previously reported."*

p8L4: …in both hemispheres during…

*We will change the sentence as suggested:*

*"Considering the long-term decreasing trend of $CH_4$ mixing ratio in both  hemispheres during the early Holocene, the increasing IPD implies that the amount of boreal emission reduction should have been less than that of low latitude emissions."*

p8L6: …from both hemispheres and…

*This sentence will be modified as below:*

*"Given the new high resolution $CH_4$ records from both  hemispheres and IPD, we ran a simple 3-box $CH_4$ source distribution model to quantify how much the boreal and tropical source strengths were changed."*

p8L8-9: extra-tropical latitudes (30N or 30S is not high latitude, rather extra-tropical)

*This will be modified.*

*"Briefly, the model contains 3 boxes; northern  extra-tropical latitude (30-90°N, N-box), tropical (30°S-30°N, T-box), and southern  extra-tropical latitude boxes (30-90°S, S-box). $CH_4$*

*concentrations in 3 boxes (in Tg box$^{-1}$) were determined from CH$_4$ mixing ratio of Antarctica and Greenland."*

p8L13-14: develop description of model assumptions and impacts, e.g. what life times did you assume and why? did you tune life times to match previous flux estimates?

*We used the identical parameters described in Chappellaz et al. (1997) and Brook et al. (2000). Previous estimates are averaged values throughout the early Holocene, so that it is difficult to compare and match our higher resolution IPD to the previous results.*

p8L15-16: quantify and discuss flux estimates, otherwise meaningless
*The flux estimation will be given.*
p8L18: in tropical emissions by XYZ Tg. (quantify)
*Amount of tropical emission change will be given with its proper uncertainty.*
p8L25-30: increased CH$_4$ emissions from boreal wetlands were previously suggested by Sowers 2010, that agreement should be acknowledged

*Please refer to our response to General Comment #5.*

p8L29: explain "conventional" northern CH$_4$ emissions

*We will change the phrase to "…small compared to current understanding of boreal emission…".*

p9L1: the isotope records are already published and need to be acknowledged (Sowers, 2010). these isotope data are available and should be added to the figures of this manuscript.

*We will add "high resolution" into the sentence as "…as well as high resolution CH$_4$ isotope ratio data for the younger time period". Also we will add the citation.*

p9L10-15: therefore, IPD should be calculated with WAIS records as well
*See our response above and Figure R2 and R3.*
p9L30: there is also no explanation for the drop in IPD if I am not mistaken?
*The slight decrease in IPD between ~10.3 and 10.8 ka is not observed in the alternative IPDs. Although it is argued above that Siple Dome CH$_4$ is more reliable data than others, we cannot rule out the possibility that the Siple data deteriorate during this period. At present it is difficult to say the drop in our IPD represents the real change or not.*

p10L5-6: why can the 10.2 ka event not be explained by low latitude hydrology, but the other events can? I feel this is a section where great care has to be taken to prevent from over interpretation. There is no quantitative estimate for low latitude emissions during other events, i am not convinced that the presented records allow for a partial explanation of the CH$_4$ minima and that there is only a missing bit. I would recommend to re-formulate. Even if the revised IPD reconstruction supports the current discussion, this might seem as two results are made to fit together. I feel this can be toned down and still be strong a conclusion.

*The 10.2 ka event does not bring corresponding abrupt change in Cariaco Basin record, Dongge Cave speleothem data, and Siple Dome Δε$_{LAND}$. Thus we speculated that CH$_4$ drop at 10.2 ka was caused by source reduction in boreal regions and/or Southern Hemisphere. We will add further discussions on this and rephrase the sentence.*

*Assessing the latitudinal source change from IPD is not reliable, given each ice core gas record has had own characteristic smoothing process at firn, because it could lead erroneous result for abrupt changing intervals such as 10.2 ka event.*

p10L6-12: I am not sure I understand this section

*We will re-formulate the sentences to better clarity.*

p10L11: the quantification 20-40 ppb is mentioned for the first time here. The conclusions cannot include information that have not been presented earlier in the manuscript. I am not sure how you quantify ppb changes? Is that from the box model?

*The amplitude of each millennial scale $CH_4$ drop is quantified by $CH_4$ anomaly curve shown in Figure 2. We will add this into earlier part of manuscript.*

p16L19: I didn't check all references, but Sowers 2010 is not correct. This is "Atmospheric methane isotope records covering the Holocene period, Quaternary Science Reviews 29, 213-221, 2010". The title/journal name in your references refers to his 2006 paper

*The reference will be modified.*

p18L4: add reference to the list or check reference

*We will check the reference list.*

p18F1: show overlapping period, show minor ticks on both axes

*Please see our response to the General Comment #1.*

p19F2: define 0 ppb in anomaly or show $CH_4$ mole fractions, check width of yellow bars, can be confusingly wide

*The Figure will be updated.*

p20F3: add IPD with WAIS data

*Please see our response above, and Figure R2 and R3.*

p22T1: caption is confusing to me, also what is this table supposed to add? how can this agreement be explained, life time?

*The table and caption will be updated with revised IPD reconstructions.*

**References that are not cited in discussion paper**

Brook, E. J., Harder, S., Severinghaus, J., Steig, E. J., and Sucher, C. M.: On the origin and timing of rapid changes in atmospheric methane during the last glacial period, Global Biogeochem. Cycles, 14, 559-572, 2000.

Buiron, D., Chappellaz, J., Stenni, B., Frezzotti, M., Baumgartner, M., Capron, E., Landais, A., Lemieux-Dudon, B., Masson-Delmotte, V., Montagnat, M., Parrenin, F., and Schilt, A.: TALDICE-1 age scale of the Talos Dome deep ice core, East Antarctica, Clim. Past, 7, 1-16, 2011.

Cheng, H., Sinha, A., Cruz, F. W., Wang, X., Edwards, R. L., d'Horta, F. M., Ribas, C. C., Vuille, M., Stott, L. D., and Auler, A. S.: Climate change patterns in Amazonia and biodiversity, Nat. Commun. 4, 1411, 2013.

Craig, H., Horibe, Y., and Sowers, T.: Gravitational separation of gases and isotopes in polar ice caps, Science, 242, 1675-1678, 1988.

Cruz, F. W., Burns, S. J., Karmann, I., Sharp, W. D., Vuille, M., Cardoso, A. O., Ferrari, J. A., Silva Dias, P. L., and Vianna, O.: Insolation-driven changes in atmospheric circulation over the past 116,000 years in subtropical Brazil, Nature, 434, 63-66, 2005.

Dai, A., and Wigley, T. M. L.: Global patterns of ENSO-induced precipitation, Geophys. Res. Lett., 27, 1283-1286, 2000.

Griffiths, M. L., Drysdale, R. N., Gagan, M. K., Zhao, J. –X., Ayliffe, L. K., Hellstrom, J. C., Hantoro, W. S., Frisia, S., Feng, Y. –X., Cartwright, I., Pierre, E. St., Fischer, M. J., and Suwargadi, B. W.: Increasing Australian-Indonesian monsoon rainfall linked to early Holocene sea-level rise, Nature Geosci., 2, 636-639, 2009.

Huber, C., Leuenberger, M., Spahni, R., Fluckiger, J., Schwander, J., Stocker, T. F., Johnsen, S., Landais, A., and Jouzel, J.: Isotope calibrated Greenland temperature record over Marine Isotope Stage 3 and its relationship to $CH_4$, Earth Planet. Sci. Lett., 243, 504-519, 2006.

Kirby, M. E., Lund, S. P., Patterson, P., Anderson, M. A., Bird, B. W., Ivanovici, L., Monarrez, P., and Nielsen, S.: A Holocene record of Pacific Decadal Oscillation (PDO) – related hydrologic variability in Southern California Lake Elsinore, CA, J. Paleolim., 44, 819-839, 2010.

Loulergue, L., Schilt, A., Spahni, R., Masson-Delmotte, V., Blunier, T., Lemieux, B., Barnola, J.-M., Raynaud, D., Stocker, T. F., and Chappellaz, J.: Orbital and millennial-scale features of atmospheric $CH_4$ over the past 800,000 years, Nature, 453, 383-386, 2008.

Lyon, B., and Barnston, A. G.: ENSO and the spatial extent of interannual precipitation extremes in tropical land areas, J. Clim., 18, 5095-5109, 2005.

Moy, C. M., Seltzer, G. O., Rodbell, D. T.: Variability of El Niño/Southern Oscillation activity at millennial timescales during the Holocene epoch, Nature, 420, 162-165, 2002.

Rodbell, D. T., Seltzer, G. O., Anderson, D. M., Abbott, M. B., Enfield, D. B., and Newman, J. H.: An ~15,000-year record of El Niño-driven alleviation in southwestern Ecuador, Science, 22, 516-520, 1999.

Partin, J. W., Cobb, K. M., Adkins, J. F., Clark, B., and Fernandez, D. P.: Millennial-scale trends in west Pacific warm pool hydrology since the Last Glacial Maximum, Nature, 449, 452-456, 2007.

Schulz, M., and Mudelsee, M.: REDFIT: estimating red-noise spectra directly from unevenly spaced paleoclimatic time series, Computers & Geosciences, 28, 421-426, 2002.

Sperlich, P., Schaefer, H., Mikaloff Fletcher, S. E., Guillevic, M., Lassey, K., Sapart, C. J., Röckmann, T., and Blunier, T.: Carbon isotope ratios suggest no additional methane from boreal wetlands during the rapid Greenland Interstadial 21.2, Global Biogeochem. Cycles, 29, 1962-1976, 2015.

Strikis, N. M., Cruz, F. W., Cheng, H., Karmann, I., Edwards, R. L., Vuille, M., Wang, X., Paula, M. S., Novello, V. F., and Auler, A. S.: Abrupt variations in South American monsoon rainfall during the Holocene based on a speleothem record from central-eastern Brazil, Geology, 39, 2011.

Taylor, K. C., Alley, R. B., Meese, D. A., Spencer, M. K., Brook, E. J., Dunbar, N. W., Finkel, R. C., Gow, A. J., Kurbatov, A. V., Lamorey, G. W., Mayewski, P. A., Meyerson, E. A, Nishiizumi, K., Zielinski, G. A.: Dating the Siple Dome (Antarctica) ice core by manual and computer interpretation of annual layering, J. Glaciol., 50, 453-461, 2004.

Weldeab, S., Lea, D. W., Schneider, R. R., and Andersen, N.: Centennial scale climate instabilities in a wet early Holocene West African monsoon, Geophys. Res. Lett., 34, L24702, 2007.

---

## Author Comment (AC2) · 26 Dec 2016

*We thank the Referee #2 for her/his pinpoint comments and important suggestions. Below we copied the Referee comments in black, and add our responses to each comment in red italics. Further we attached the paragraphs of original discussion paper in blue, and our modifications in green.*

Overall assessment

The manuscript provides new high-resolution CH4 data from the Siple Dome ice core over the time interval 8.5-11.5 kyr BP, extending previous work by Ahn et al., 2014. The data quality is generally very good (see comments below) and I commend the authors for their painstaking work to provide high-resolution data sets using discrete CH4 analyses. The data is interpreted with respect to millennial climate variations during this time interval and, based on correlation with other climate proxy data, a suggestive hypothesis about the influence of changes in the ITCZ is presented to explain the millennial variability in CH4.

Finally, the interpolar CH4 difference (IPD) is calculated using Greenland data from the literature and this difference is then analyzed using a simple three-box model. As outlined below, I have some fundamental questions about the reliability of the inferred IPD, which subsequently has also important implications for its interpretation. This prevents me from recommending the manuscript for publication in CP in its current form despite the nice data. Moreover, the quality of the manuscript in terms of the use of the English language has to be considerably improved. With a native English speaker on the author list, I see no problem that this can be achieved. In summary, after additional work I am confident that the manuscript will become suitable for publication in CP in a resubmitted version.

General comments

a)          comparison of early and late Holocene CH4 variations: In the introduction the authors make the important point that only the early Holocene period allows us to study the natural CH4 variability on centennial to millennial time scales. Unfortunately, the authors do not follow up on this in the discussion. It would be interesting to compare the centennial and millennial variability in CH4 concentrations in the early Holocene (as documented in the Siple Dome, WAIS (including the continuous CH4 data by Rhodes et al., 2015) and NEEM record with the late Holocene as documented in WAIS by Mitchell et al., 2013. Are the amplitudes of this variability different and if so, is that due to an anthropogenic influence in the late Holocene or related to the changes in seasonal and geographical distribution of solar insolation (due to orbital parameter changes) between the early and late Holocene? Note that summer insolation in high northern latitudes was several tens of W/m2 higher in the early Holocene. For this analysis it may be beneficial to use the continuous WAIS data instead of the Siple Dome data to calculate a record with comparable resolution to the Mitchell data and to compare CH4 frequency spectra for the early and late Holocene.

*We appreciate to Referee #2 for this important comment. We will add a dedicated chapter based on discussion below:*

*We compared amplitude of methane variability between the early- and late Holocene in multi centennial to millennial time scales. Figure R1 shows amplitude spectrum and root mean square (RMS) amplitude for the early Holocene and late Holocene, respectively. The amplitude of the early Holocene CH$_4$ change is ~10 ppb and does not change a lot except for PBO and the 8.2k event, while the late Holocene spectrum shows smaller amplitude than early Holocene for shorter-term change and larger for longer-term fluctuation. LPIH (Late Pre-Industrial Holocene) CH$_4$ amplitude is elevated to early Holocene level from ~0 C.E., and increases up to higher from ~1450 C.E.*

*The reason of low amplitude variability during 3.5 to 2.0 kyr BP, or inversely, why the early Holocene CH$_4$ variability is larger than this period, is probably related to different orbital configuration in both time periods. Previous studies found covariation between CH$_4$ amplitude and NH summer insolation change, reflecting that mean temperature of the warmest seasons is an important factor of CH$_4$ emission, during the interstadial conditions (Flückiger et al., 2004; Baumgartner et al., 2014). Accordingly, lower summer insolation during the late Holocene might*

*induce diminished CH₄ amplitude, and vice versa during the early Holocene. This evidence indicates the natural forcing in centennial- to millennial time scales is reduced in the late Holocene, given that the atmospheric CH₄ budget is similar between 3.5-2.0 kyr BP (604.9 ppb) and 9.0-7.6 kyr BP (628.6 ppb) interval, and that anthropogenic emission is larger in later Holocene than early Holocene.*

*Abrupt increase of CH₄ amplitude since ~800 C.E. is likely driven by increasing anthropogenic contribution, which is consistent with anthropogenic emission scenario based on past population and agricultural activity (Mitchell et al., 2013). Also superimposed are short-term cooling events during Little Ice Age, making CH₄ variability larger.*

[Figure]

*Figure R1. Upper: Detrended (75 to 1800-year band-pass filtered) CH₄ for the early (a) and late (b) Holocene from Siple Dome (red, this study), WAIS divide continuous (purple, Rhodes et al., 2015), and WAIS divide discrete (blue, Mitchell et al., 2013) data. Dashed lines are root mean square (RMS) amplitude running averaged by 75-year window. Lower: Amplitude spectrum of Early (c) and Late (d) Holocene CH₄ records. Note that CH₄ data before 1750 C.E. are used for the preindustrial late Holocene.*

b) Millennial CH4 variations: The authors suggest that climate cooling in the northern hemisphere has led to a southward shift in the ITCZ, which again led to a decline in CH4 low latitude emissions due to changes in monsoon systems. The first part of this hypothesis (ITCZ shift) appears to be straightforward and has been observed in models, however, the second part (CH4 emission changes) appears not so straightforward and requires some more quantitative support. Rhodes et al. (2015) suggest a first order relationship between CH4 emissions and intense rainfall area, where from a certain point on also an increase in southern hemisphere wetland emissions is possible. Accordingly, a discussion focusing on Asian monsoon systems only, as in the manuscript by Yang, seems to be too narrow. Please explain how your hypothesis fits into this picture. Please note also the work by Bozbiyik et al., CP, 2011, performing a North Atlantic fresh water hosing experiment under interglacial conditions connected to a southward shift of the ITCZ, showing decreases in tropical precipitation and the modeling work by Zurcher et al., Biogeoscience, 2013, which shows that also boreal peatland CH4 emissions are reduced during such an experiment. Finally, the discussion of the millennial CH4 variability and the corroborating proxy evidence from other archives lacks some clarity and could be improved.

*We thank again for useful comment and paper suggestion. We will consider those papers and update the sections based on the discussion and figure below:*

*One of main point of Rhodes et al. (2015) is that abrupt CH₄ increase occurred during Heinrich Stadial 1, 2, 4, and 5 events could be induced by the southern hemisphere emission as a result of strong southward migration of ITCZ at that time. However, mean latitudinal position of ITCZ moved northward during the Holocene climate conditions, therefore monsoon intensity in northern tropics should be strengthened while tropical rainfall in southern hemisphere decreased. This is identified from Cariaco Basin reflectance record which shows increase in rainfall during the Holocene compared to glacial period (e.g., Deplazes et al., 2013), as well as from anti-correlation*

*between Chinese- and Brazilian cave speleothem record (e.g., Wang et al., 2006). Further, Asian monsoon intensity during the Heinrich Stadials was weaker than during the early Holocene (e.g., Wang et al., 2008). For these reasons, we focused more on Asian/Indian monsoon variability than others, but we agree that our manuscript lacks detailed explanation on other monsoon systems. We will add more discussions and change Figure 2.*

*The new Figure 2 (Figure R2 below) now shows Asian, Indian, African and Brazilian monsoon proxies, with age uncertainties indicated as black horizontal error bars. Gulf of Guinea (western tropical Africa) planktonic Ba/Ca ratio, a proxy of riverine runoff, shows decreased rainfalls at similar timings of local $CH_4$ minima at 8.2, 9.3 and 10.9 kyr BP, indicating that the abrupt Greenland cooling leads to hemispheric-wide hydroclimatic changes. Inverse pattern of South American rainfall (Lapa Grande Cave, eastern Brazil) supports that ITCZ was temporarily migrated southward at that time. High-resolution sediment reflectance records from Cariaco Basin and Arabian Sea clearly show that the strength of southward migration of ITCZ and its effect of precipitation change are smaller during the early Holocene than during the Heinrich Stadials (Deplazes et al., 2013; 2014).*

*We thank the reviewer for suggesting appropriate paper. Zurcher et al. (2013) found that abrupt cooling in Greenland and northern high latitudes by large freshwater input causes boreal peatland $CH_4$ emission to decrease substantially, which explains ~23% of abrupt $CH_4$ drop (~80 ppb) during the 8.2k event. If we assume linear scaling of model response, it implies that boreal peatland source change only accounts for ~23% of total $CH_4$ change during the rest of $CH_4$ decrease events. Given the meltwater pulses during the early Holocene before the 8.2 k event are more than 10 times smaller (Teller and Leverington, 2004)than that corresponds to the 8.2 k event, we consider the boreal emission change is not the major cause of $CH_4$ local minima.*

[Figure]

*Figure R2. Millennial scale climate variability. All proxies present here were smoothed by 250-year running average and detrended by high-pass filter with 1/1800-year window. (a) Siple Dome $CH_4$ (red, this study), Greenland $^{10}Be$ (dark yellow, Finkel and Nishizumii, 1997), North*

*Atlantic IRD stack (grey, Bond et al., 2001). Also shown are WAIS Divide CH₄ data by discrete (cyan, WAIS Divide members, 2015) and continuous (yellow green, Rhodes et al., 2015) technique. (b) NGRIP stable water isotope ratio (blue, Rasmussen et al., 2006) and Cariaco Basin reflectance (orange, Deplazes et al., 2013). (c) Qunf Cave speleothem oxygen isotope (Fleitmann et al., 2007). (d) Dongge Cave speleothem oxygen isotope (green, Dykoski et al., 2005; dark yellow, Wang et al., 2005). (e) Gulf of Guinea planktonic Ba/Ca ratio (Weldeab et al., 2007). (f) Lapa Grande Cave speleothem oxygen isotope (purple, Strikis et al., 2011). Age tie-points are used to synchronize Siple Dome and WAIS Divide CH₄ data with the GICC05 time scale (black triangles).*

c)        Interpolar CH4 difference: The IPD is a tricky business and erroneous effects can be easily introduced by comparing CH4 data from different labs, different sites, or insufficient robustness of the results. Accordingly, this needs more supporting information and detail: In the method part it is mentioned that blank ice measurements show an offset with a very large scatter between 5 and 15 ppb. I read the text in such a way that a daily blank correction is applied based on 4 blank measurements per day. This needs more detailed discussion as a potentially erroneous correction, which varies by 10 ppb, has a huge influence on the IPD, which varies with a similar amplitude. Please add the following information/discussion:

- Can you be sure that the CH4 blank is coming from the extraction system and is not reflecting dissolved CH4 in the bubble-free ice? In the latter case you should not correct for this blank. Did you perform blank tests without bubble-free ice or did you repeat the extraction of bubble-free ice for a second or third time to see, whether the blank is constant or declining?

*We've performed the gas extraction test using bubble-free ices to estimate how much air dissolves in ice melt. We found less than 5 mTorr of gas extracted after second melting-refreeze procedure. Considering the typical amount of standard air injected (30-40 Torr), the air extracted from the second melting-refreezing process should have CH₄ mixing ratio of >~100000 ppb to cause 20 ppb of blank offset. This is unlikely because (1) such high concentration cannot be explained by gas solubility effect, and (2) the bubble-free ices were trimmed outermost layer sufficiently before cutting the artificial ice samples to prevent causal contamination by ambient air dissolution. The microbial activity within bubble-free ice during storage is unrealistic, given that we use water distiller (Barnstead) and anti-bacterial membrane filter (Millipak Express) to produce deionized water, and the deionized water is boiled within a stainless steel chamber for degassing. Further, we found the blank offset is similar order from the blank test without bubble-free ices (9.4 ~ 21.6 ppb, n = 36) using various working standards (721.31, 895.03, and 1384.91 ppb CH₄ in NOAA04 scale). Therefore, we conclude that the CH₄ blank offset caused by dissolved CH₄ in bubble-free ice is not the case.*

-        Please add information on the time scale on which the blank changes. If my understanding is correct that a daily mean blank correction of 5-10 ppb is performed, it is important to know what the variability of the four blank ice measurements is within one day. If this intra-day variability is of the same size as the inter-day variability, then a daily blank correction varying between 5 and 15 ppb introduces offsets from one day to the other which only reflect the stochastic variability of the blank measurements themselves and not systematic day-to-day differences in the entire measurement system. In that case a long-term mean blank correction seems more appropriate. If the blank values are reproducible within one day, a daily correction seems justified.

*The daily systematic offset was determined using a standard air injected into the sample flasks which have bubble-free blank ices. Even though the systematic offset varied daily, the 4 blank results were rather in a small range, yielding the intra-day standard error of the mean of the 4 blanks of 2.0±1.0 ppb on average. Our final CH₄ data were presented by averaging the results of duplicate sample analysis from the same depths. To test our correction method, we reanalyzed duplicate samples from the adjacent ices (~10 cm depth difference) at 8 depth intervals 8-80 days after the first analysis (see the table in the response to reviewer #1). The pooled standard deviation of the average of duplicates from first and second measurements was ±1.0 ppb. Given that we found a good reproducibility from the reanalysis, our blank correction method is robust.*

-        On the other hand if you have a long-term trend in this blank value, which may not reflect a trend in the extraction system but in the bubble-free ice quality, you introduce this error into the IPD. Did you use a randomized order to measure the samples to avoid such spurious trends.

*Siple Dome ice samples were measured in a randomized order.  As we described above, contamination by blank ice quality and/or by air occluded in blank ice is unlikely. Therefore, we decide not to apply this correction.*

- There was an average offset of 3 ppb observed between Siple Dome data measured at SNU and OSU and the OSU data have been corrected by subtracting 3 ppb, but it is not discussed what the influence of this correction may be on the IDP. Note that the NEEM discrete data used to calculate the IPD are also measured at OSU. Accordingly, to avoid any systematic errors in the IPD it is mandatory to add the 3 ppb to the Siple Dome data measured at SNU and not to subtract 3 ppb from OSU data.

*The offset comes from different correction methods between the two laboratories. As described in our response to the Reviewer #1, instead of simply adding 3 ppb to SNU data, we applied the similar solubility correction used in Mitchell et al. (2011) at OSU, and found that the average offset between SNU-OSU reduces to ~0.6 ppb, which lies within analytical uncertainty range of both institutes. Instead of adding 3 ppb, now we apply the OSU solubility correction methods to our data for IPD calculation.*

- Comparing the continuous WAIS data with the discrete (or continuous) CH4 data from NEEM, it becomes apparent that the relative changes in CH4 concentrations in the northern and southern hemisphere are much more similar than when comparing Siple Dome and NEEM data after the Monte Carlo synchronization in Figure 3. For example the downward trend in NEEM between 11.3 and 10.9 kyr BP is also seen in the continuous WAIS data, while in Siple this time interval shows essentially constant CH4 after a first short peak. Consequently, the constant values in the Siple data lead to an erroneous downward trend in the IPD in this time interval. Vice versa, there is an upward trend from 10.9 to 10.4 kyr BP found in NEEM and continuous WAIS data. The same time interval in Siple looks more like a broad maximum, again with implications on the derived IPD. A similar observation holds for the maximum around 10 kyr BP. Note that on these centennial to millennial time scales, which are much longer than the atmospheric lifetime and the interhemispheric exchange time, you may have changes in the size of the IPD, however, it is extremely difficult to create a millennial trend in one hemisphere without a trend in the other. This is nicely illustrated in the high-resolution data by Mitchell et al., 2013. Obviously, the resolution and quality of the data in Figure 3 does not suffice to gain a robust IPD and/or the Monte Carlo synchronization fails to synchronize the records sufficiently. In fact, it seems crucial that the IPD analysis is performed not only on the Siple but also on the WAIS discrete and continuous data to study the robustness of the results gained from the Siple Dome core. Note that the WAIS very high-resolution data from continuous measurements can be used to much better synchronize WAIS to the continuous records available from NEEM. This would circumvent the synchronization problems apparent between Siple and NEEM. As a final remark on this topic, I do not agree with the authors' statement that the IPD values in Siple Dome over the time interval 9.5-11.5 kyr BP are in agreement with previous results. If you calculate the mean over this time interval in the Siple IPD data and calculate the standard error, this appears to clearly higher than the literature values. In summary, the IPD discussion needs more work before the manuscript should be published in CP.

*We thank to the Reviewer #2 for pinpoint comment and useful suggestions. We will revise the IPD section of our manuscript based on the following discussions:*

*To test robustness of the early Holocene IPD change, we revised the previous IPD and calculated various IPD curves by using different data sets, including NEEM continuous, NEEM discrete, Siple Dome discrete, WAIS continuous, and WAIS discrete data (Figure R3). We calibrated NEEM and WAIS continuous data against to discrete measurements, as discrete measurements are more accurate than continuous ones in absolute values although they are worse in precision. However, we hesitate to equally consider all the IPDs because some CH$_4$ data sets are not sufficient for centennial to millennial IPD estimates (see below for detailed discussion on data reliability). Here we consider NEEM discrete, WAIS Divide continuous, and Siple Dome discrete records are reliable to draw IPDs. Also we address the inconsistency among the different ice core records for interval older than ~10.3 kyr BP, which make IPDs different each other.*

*Resulting IPD curve from NEEM discrete and WAIS continuous (green) shows long-term increase from the onset of Holocene to ~9.9 kyr BP, and it supports the NEEM-Siple IPD reconstruction (black). The good agreement implies that the millennial scale IPD increase trend during the early Holocene is robust. However, the IPD fluctuation during 10.8 – 11.2 kyr BP is not reproduced in the alternative IPD, hence we'll modify our original argument that IPD increase from ~10.7 to 9.8 kyr BP. Instead, we'll more focus on long term IPD increase. Except for during 10.8 – 11.2 kyr BP, both IPDs show concomitant increase with NH extratropical temperature and thermokarst lake CH$_4$ emission increase. These evidences show that boreal emission increased while tropical emission decreased (Table S1).*

*Finally, we'd like to note that mean value of the Siple Dome IPD is 41 ± 6 ppb over the 9.5-11.5 kyr BP period, which is consistent with previous results within uncertainty range.*

[Figure]

*Figure R3. Inter-polar difference (IPD) reconstructions. Top: high resolution CH$_4$ records from Greenland and Antarctica, synchronized to NEEM gas age scale by Monte Carlo procedure. Middle: Millennial-scale IPD trends derived from Siple Dome (black) and WAIS Divide continuous (green) data. Shaded area indicates 95% significance interval. Bottom: Proxy-based temperature reconstruction for northern mid to high latitude and boreal CH$_4$ emission from northern thermokarst lakes. Note that this figure may be subject to change.*

*We excluded some dataset from our discussion for the following reasons. Rhodes et al. (2015) reported that WAIS continuous data are lower than OSU discrete measurements by 1.5-2.5% for 1804-2420m (9.8 – 17.1 kyr BP in WD2014 scale) interval. WAIS continuous data were calibrated against to Siple discrete data instead of WAIS data, because we consider that Siple data are more reliable during the early Holocene period. Even though the analytical method of Mitchell et al. (2011) has been regarded as a "benchmark" of discrete wet-extraction technique, but unfortunately, none of existing Antarctic CH$_4$ data for the early Holocene was measured by Mitchell et al. (2011) method. Most of WAIS discrete data covering the early Holocene were measured in a different institute (Penn State University, PSU) showing a noisy trend with pooled standard deviation of ~7.3 ppb (1σ), and there is an unexplained offset of 9.9 ppb between WAIS discrete data measured in OSU and PSU lab (Rhodes et al., 2015). As it lacks rigorous comparison between the two data sets during the early Holocene, there is no evidence to show that WAIS discrete data are more reliable. Meanwhile, it has been revealed that during the early Holocene interval, SNU Siple data (this study) agree well with OSU Siple data (Ahn et al., 2014) by comparison of the nearest data point from both labs. Furthermore, it should be noted that we used NEEM continuous data obtained by WS-CRDS (Wavelength Scanned Cavity Ring Down Spectroscopy, CFADS36, Picarro Inc.)) because the OF-CEAS (Optical Feedback Cavity Enhanced Absorption Spectroscopy, SARA, Laboratoire Interdisciplinaire de Physique, Universite Joseph Fourier, Grenoble, France) instrument was calibrated against different standard scale (CSIRO standard, Chappellaz et al., 2013), and WAIS continuous data were measured by the same instrument (WS-CRDS, 1804-2621m depth, 2012 campaign, Rhodes et al., 2015). In summary, we regard the NEEM discrete, Siple Dome discrete, and WAIS Divide continuous records as more reliable than the others during the early Holocene period.*

[Figure]

*Figure R4. Same as Figure R3, but including various IPDs calculated from different dataset.*

*Table S1. 3-box source distribution model results of tropical (green, T) and boreal (red, N) boxes and boreal emission fraction (N/(T+N+S)) compared with previous results. Errors denote 95% confidence interval.*

| Ref. | N box | T box | N/(N+T+S) ratio |
|---|---|---|---|
| (ka) | (Tg yr⁻¹) | | (%) |
| Brook et al., 2000 (9.5-11.5 ka) | 64 ± 5 | 123 ± 8 | 32 ± 3 |
| Chappellaz et al., 1997 (9.5-11.5 ka) | 66 ± 8 | 120 ± 9 | 33 ± 3 |
| This study (9.5-11.5 ka) | 66 ± 4  | 120 ± 4  | 33 ± 2  |
| This study (11.5 ka) | 57 ± 6 | 119 ± 11 | 30 ± 4 |
| This study (9.9 ka) | 70 ± 4  | 115 ± 4  | 35 ± 2  |

Specific comments

I started to correct for English language issues, but had to stop at some point. Please ask your English speaking co-author for a thorough language check not only for typos but also to improve the clarity of the arguments. As major textual changes are still required for this manuscript, I will not comment on language issues here.

P(age) 2 l(ine) 2: Daniau et al., 2012 is not an appropriate reference in this respect (CH4 emissions)
*We will replace the reference with citations below to deal with pyrogenic CH₄ emissions:*

*Ferretti, D. F., Miller, J. B., White, J. W. C., Etheridge, D. M., Lassey, K. R., Lowe, D. C., MacFarling Meure, C. M., Dreier, M. F., Trudinger, C. M., van Ommen, T. D., and Langenfelds, R. L.: Unexpected changes to the global methane budget over the past 2000 years, Science, 309, 1714-1717, 2005.*

*Andreae, M. O., and Merlet, P.: Emission of trace gases and aerosols from biomass burning, Global Biogeochem. Cycles, 15, 955-966, 2001.*

*Hao, W. M., and Ward, D. E.: Methane production from global biomass burning, J. Geophys. Res., 98, 20657-20661, 1993.*

P2 l20: Cite recent work by Baumgartner et al. CP 2014

*We will add the citation.*

P2: discuss in more detail previous work on the relationship between ITCZ changes and CH4 emissions

*We will add a paragraph such as the following one in the introduction section:*

*"Relationship between the latitudinal shift of ITCZ and $CH_4$ emission varies with time scales. Landais et al. (2010) and Guo et al. (2012) suggested that ITCZ migration is not a dominant control of glacial-interglacial $CH_4$ cycle because long-term $CH_4$ trend does not follow well the precession change. Modelling studies found the southward shift of ITCZ coincides with reduced $CH_4$ in LGM and HS events, but changes in wetland area and surface hydrology were small (Weber et al., 2010; Hopcroft et al., 2011). They instead suggested that changes in temperature and/or plant productivity affected $CH_4$ production during those events.*

*Rather, ITCZ migration seems to be related with millennial- or submillennial scale $CH_4$ change. Brook et al. (2000) found that submillennial-scale $CH_4$ minima during the last deglacial period correspond with reduced precipitation recorded in Cariaco Basin sediment data, which indicates southward displacement of ITCZ (Hughen et al., 1996). It is supported by spectral analysis of $CH_4$ during the past 800 kyr record that found that ITCZ change becomes an important driver of millennial scale $CH_4$ change (Tzedakis et al., 2009; Guo et al., 2012)."*

P3: discuss the difference in orbital parameters for the early and late Holocene and the potential implications for CH4 emissions

*We will add a dedicated paragraph based on our comment to General comment a).*

P4 methods: Is it correct that you use only a one standard calibration? Comment on the potential systematic error introduced by this approach

*The GC linearity was tested by using working standards of 395.50, 721.31, 895.03, and 1384.91 ppb (in NOAA04 scale). We will add more details in the method section.*

P4 l25-32: This paragraph should be moved to the methods section

*We will move that paragraph to method section.*

P5 1st paragraph. You say that you use a 250 year running average (and similar a high-pass filter with 1800 cut-off), however, your data is not equidistant. Please explain in more detail how you averaged the data

*We interpolated the data annually and then averaged each 250-year interval. We will explain more about data filtering and averaging process.*

P5 l14: Is the significance level of the correlation coefficient really taking the reduced degrees of freedom into account after averaging the data? Looking at the value, I am afraid it didn't and the significance is highly overestimated.

*The p-value was estimated by a reduced degree of freedom. As described in Discussion Paper, Siple Dome $CH_4$ gas age was adjusted to GICC05 scale by matching to NGRIP $\delta^{18}O$ at 8.2 ka and PBO (e.g., Kobashi et al., 2007). We will note that it might overestimate the correlation coefficient.*

P5: see comment on insufficient discussion of the effect of an ITCZ shift on CH4 emissions north and south of the equator. Please discuss also in more detail the dating uncertainties of the various archives and their potential impacts on the conclusions.

*We will add age uncertainty ($1\sigma$) of each proxy used (Figure R2).*

P5 l30: Reference Bjorck et al. is not in the reference list

*The reference will be added.*

*Björck, S., Muscheler, R., Kromer, B., Andresen, C. S., Heinemeier, J., Johnsen, S. J., Conley, D., Koç, N., Spurk, M., and Veski, S.: High-resolution analyses of an early Holocene climate event may imply decreased solar forcing as an important climate trigger, Geology, 29, 1107-1110, 2001.*

P6 l5-7: There is also variability in GRIP and GISP2. Please explain in more detail what you refer to.

*We will delete the sentence.*

P7 l1-17: This paragraph is highly speculative and lacks clarity and detail.

*In this paragraph we intended to discuss possibility of solar forcing to observed $CH_4$ change. Several previous studies found evidences solar-induced climate change, but we observed that the timings of solar activity minima differ by 195 (8.2 ka), 278 (9.3 ka), 110 (10.3 ka), and 250 (11.0 ka) years to $CH_4$ minima and Greenland cooling. The maximum layer counting error of GRIP age scale (GICC05) is less than 100 years (Rasmussen et al., 2006), and the maximum gas age uncertainty of Siple Dome is ~150 years (This study). Therefore, age difference larger than ~180 years is not explained by chronological uncertainty.*

P7 following l24: You disturbed the age of the data points by a Gaussian distribution with sigma=30 years. How did you make sure that the chronological order of all data points was ensured in your approach? How did you take the measurement uncertainty in each data point into account? Please explain in more detail.

*We chose the sigma of 30 years given the mean temporal resolution of Siple data is ~27 years. We will describe the synchronizing process more detail.*

P8 l1: there is a significant offset between your average and previous IPD estimates

*Figure R3 (above) shows the IPD calculated from NEEM discrete and SDMA discrete data together with alternative IPDs from different data set. Previous IPD estimates lie within range of the new IPDs. The IPDs calculated from NEEM continuous data show higher values, which reflects the offset between NEEM continuous and discrete record.*

P8 l10-12: not entirely clear to the outsider what you did, please clarify

*We will modify the paragraph as below:*

*"~~To calculate the N-box $CH_4$, we subtracted the 7 % of IPD from Greenland $CH_4$ concentration, assuming the difference between Greenland and the mean latitude of N-box is ~7 % of IPD (Chappellaz et al., 1997).~~"*

*"The mean $CH_4$ concentration of N-box (30-90N) is not identical to that of Greenland ice core record, given the latitudinal $CH_4$ distribution (e.g., Fung et al., 1991). To derive the N-box $CH_4$, we followed the assumption of Chappellaz et al. (1997), where the authors assumed that difference between Greenland and the mean N-box $CH_4$ is 7% of IPD. Hence here the N-box $CH_4$ is calculated by subtracting 7% of IPD from the Greenland concentration."*

P8 l19: the boreal sources increased

*This phrase will be changed as below:*

*"This result supports our interpretation that the boreal sources  increased during the early Holocene."*

P9 l10. You discuss the effect of the different age distributions in the Siple Dome and NEEM cores, but you do not follow up on this in your analysis. Either you use WAIS to compare with NEEM (as it has essentially the same enclosure characteristics) or you low-pass filter NEEM to the same enclosure characteristics as Siple. I would strongly recommend to do both to study the robustness of the results.

*See our response to general comment above. We present alternative IPD reconstructions including WAIS and NEEM data.*

P9 l20-21: The results by Fischer et al. (2008) on LGM biomass burning emissions result from the use of temporally constant isotopic source signatures in the box model approach. Moller et al., Nature Geoscience, 2013 showed that also the source signatures changed significantly over time and they revised the biomass burning estimates, showing that LGM emissions were lower than Holocene emissions.

*We will insert below paragraphs (P9 l20-31):*

*"On the other hand, Fischer et al. (2008) argued nearly constant biomass burning emission of ~45 Tg yr$^{-1}$ throughout the last glacial termination with a slight increase in PB, and also showed that the boreal sources were expanded during the YD-PB transition. However, Moller et al. (2013) pointed out the possibility of changing isotopic signature of each sources itself, and they found that less pyrogenic emission is required for LGM condition if they increased the $\delta^{13}$C-CH$_4$ signatures of tropical wetland and biomass burning. The triple isotopic … old carbon (e.g., Petrenko et al., 2009).  The IPDs at the very start of the Holocene and during the PBO show large offset among each other that prevents us from drawing a reliable trend because IPD calculation could be sensitive to sample resolution and calibration."*

P9 l26: why do you only refer to biomass burning in the tropics?

*According to model estimation by Walter-Anthony et al. (2014), CH$_4$ emission from the thermokarst lakes started to increase more later than PBO. We will describe this in that sentence.*

P20 l5 Chappellaz et al., 1997 not 2013

*The citation will be corrected.*

**References that are not cited in discussion paper**

Andreae, M. O., and Merlet, P.: Emission of trace gases and aerosols from biomass burning, Global Biogeochem. Cycles, 15, 955-966, 2001.

Baumgartner, M., Kindler, P., Eicher, O., Floch, G., Schilt, A., Schwander, J., Spahni, R., Capron, E., Chappellaz, J., Leuenberger, M., Fischer, H., and Stocker, T. F., NGRIP $CH_4$ concentration from 120 to 10 kyr before present and its relation to an $\delta^{15}N$ temperature reconstruction from the same ice core, Clim. Past, 10, 2014.

Björck, S., Muscheler, R., Kromer, B., Andresen, C. S., Heinemeier, J., Johnsen, S. J., Conley, D., Koç, N., Spurk, M., and Veski, S.: High-resolution analyses of an early Holocene climate event may imply decreased solar forcing as an important climate trigger, Geology, 29, 1107-1110, 2001.

Dai, A., and Wigley, T. M. L.: Global patterns of ENSO-induced precipitation, Geophys. Res. Lett., 27, 1283-1286, 2000.

Ferretti, D. F., Miller, J. B., White, J. W. C., Etheridge, D. M., Lassey, K. R., Lowe, D. C., MacFarling Meure, C. M., Dreier, M. F., Trudinger, C. M., van Ommen, T. D., and Langenfelds, R. L.: Unexpected changes to the global methane budget over the past 2000 years, Science, 309, 1714-1717, 2005.

Flückiger, J., Blunier, T., Stauffer, B., Chappellaz, J., Spahni R., Kawamura, K., Schwander, J., Stocker, T. F., and Dahl-Jensen, D., $N_2O$ and $CH_4$ variations during the last glacial epoch: Insight into global processes, Global Biogeochem. Cycles, 18, GB1020, 2004.

Guo, Z., Zhou, X., and Wu, H.: Glacial-interglacial water cycle, global monsoon and atmospheric methane changes, Glim. Dyn., 39, 1073-1092, 2012.

Hao, W. M., and Ward, D. E.: Methane production from global biomass burning, J. Geophys. Res., 98, 20657-20661, 1993.

Hopcroft, P. O., Valdes, P. J., and Beerling, D. J.: Simulationg idealized Dansgaard-Oeschger events and their potential impacts on the global methane cycle, Quat. Sci. Rev., 30, 3258-3268, 2011.

Hughen, K. A., Overpeck, J. T., Peterson, L. C., and Anderson, R. F.: The nature of varved sedimentation in the Cariaco Basin, Venezuela, and its palaeoclimatic significance, Geol. Soc. London Spec. Publ., 116, 171-183, 1996.

Landais, A., Dreyfus, G., Capron, E., Masson-Delmotte, V., Sanchez-Goñi, M. F., Desprat, S., Hoffman, G., Jouzel, J., Leuenberger, M., and Johnsen, S.: What drives the millennial and orbital variations of $\delta^{18}O_{atm}$?, Quat. Sci. Rev., 29, 235-246, 2010.

Lyon, B., and Barnston, A. G.: ENSO and the spatial extent of interannual precipitation extremes in tropical land areas, J. Clim., 18, 5095-5108, 2005.

Strikis, N. M., Cruz, F. W., Cheng, H., Karmann, I., Edwards, R. L., Vuille, M., Wang, X., Paula, M. S., Novello, V. F., and Auler, A. S., Abrupt variations in South American monsoon rainfall during the Holocene based on a speleothem record from central-eastern Brazil, Geology, 39, 2011.

Tzedakis, P. C., Palike, H., Roucoux, K. H., and de Abreu, L.: Atmospheric methane, southern European vegetation and low-mid latitude links on orbital and millennial timescales, Earth Planet. Sci. Lett., 277, 307-317, 2009.

Wang, X., Auler, A. S., Edwards, R. L., Cheng, H., Ito, E., and Solheid, M., Interhemispheric anti-phasing of rainfall during the last glacial period, Quat. Sci. Rev., 25, 3391-3403, 2006.

Wang, Y., Cheng, H., Lawrence Edwards, R., Kong, X., Shao, X., Chen, S., Wu, J., Jiang, X., Wang, X., and An, Z.: Millennial- and orbital-scale changes in the East Asian monsoon over the past 224,000 years, Science, 451, 1090-1093, 2008.

Weldeab, S., Lea, D. W., Schneider, R. R., and Andersen, N., Centennial scale climate instabilities in a wet early Holocene West African monsoon, Geophys. Res. Lett., 34, L24702, 2007.

Weber, S. L., Drury, A. J., Toonen, W. H. J., and van Weele, M.: Wetland methane emissions during the Last Glacial Maximum estimated from PMIP2 simulations: Climate, vegetation, and geographic controls, J. Geophys. Res., 115, D06111, 2010.

Zürcher, S., Spahni, R., Joos, F., Steinacher, M., and Fischer, H.: Impact of an abrupt cooling event on interglacial methane emissions in northern peatlands, Biogeosciences, 10, 1963-1981, 2013.

---

## Author Response (AR1)

*We thank the anonymous referee #1 and #2, and the Editor for her/his careful reviews of our paper.*

*We appreciate the useful comments and believe the input improved greatly the manuscript. Below*

*we attach our point-by-point response to all the comments. The original referee comments are copied*

*in black, and the author's response to the comments are given in red italics. We add paragraphs from*

*original discussion paper in green italics and our modifications in blue italics.*

***Editor Report*-----------------------------------------------------------------------------------------------------------------**

Comments to the Author:

Thank you for the very detailed responses to the comments from the two reviewers. Particularly through your more detailed treatment of the IPD it seems that you should be able to make a new version that will satisfy most of the concerns of the reviewers. I am therefore now happy to encourage you to prepare the new version, which I expect to send back for re-review

*We sincerely appreciate the Editor for giving us chance to revise and update our manuscript.*

On the analytical isde, I still don't feel that I understand how you can end up at a precision of 1 ppb when you have blanks varying by 10 ppb. Nor do I really understand how you arrive at a headline figure of 1 ppb when the mean absolute difference between replicate measurements was about 2

ppb. I think this might require more thought as my feeling is that the reviewers will share my concerns.

*We revised the gas solubility correction scheme following the empirical method of Mitchell et al.*

*(2011) for better fit to OSU-measured SDMA data (0.6 ppb difference instead of 3 ppb before). With*

*new correction method, the pooled standard deviation between replicates increased slightly to 1.4*

*ppb, but it does not change the main plot of this manuscript. The good agreement between 1st and*

*$2^{nd}$ replicate measurements reveals that our correction method for daily blank offset is valid. The*

*"intra-day" offset among the 4 blank is about 2 ppb, but given that all the samples were measured in*

*duplicate, this figure should be reduced by a factor of $\sqrt{2}$, which yields similar uncertainty to the*

*stated precision.*

For the IPD, I do appreciate the multiple parallel calculations, but again have two concerns. The first is simply presentational. In Fig R2 the Siple based reconstruction, central to your paper is almost completely invisible. But now I look at it in Fig R3 and it is completely different from what you showed in the original paper (Fig 3). The absolute values of IPD (NEEM/SD) there ranged from ~42-52

ppb, now you show 30-48. The shape is different as well. This is not a question that WAISDivide differs from SD, but your SD/NEEM profile has changed considerably, and yet I can't find any discussion of this in the response. The IPD at 11.5 ka, using the same (NEEM/SD) data, has changed from 52 to around 35 ppb, a very considerable change. Have you resynchronised or changed some values? Please address this in the material you submit with the paper that addresses the review comments.

*Since the initial manuscript submission, we extended IPD synchronization interval back to ~12 ka for*

*better matching during the YD termination. This has not changed the essential findings of our*

*manuscript, and the section 3.2.2. has been removed because it is out of main focus of our paper.*

*Furthermore, the Supplementary Figure 3 (Figure S3) has been removed and replaced by a plot that*

*shows the alternative IPDs not being discussed in the main text.*

Some smaller points from me:

You say Fig R1 includes Fluckiger but i don't see it.

*We included EPICA Dome C data set by Loulergue et al. (2008) instead of Flückiger et al. (2002),*
*which is more recently published data from EDC core.*

For the second rev, Fig R1, you should use the same y-scale for panels a and b so readers can
compare the magnitude of variations directly.

*We adjusted both y-axes to have the same scale and same size as suggested.*

***Referee Comment #1***--------------------------------------------------------------------------------------------------------

**Summary of manuscript**

First of all, I would like to congratulate Ji-Woong Yang et al. for the excellent work they put into
producing a high-resolution record of $CH_4$ mole fractions as well as their interpretation of the data.
So far, the early Holocene is underrepresented in high-resolution $CH_4$ reconstructions and this paper
will be a valuable addition to the literature. I hope the following comments will be helpful and I look
forward to reading the revised version of the manuscript.

Ji-Woong Yang et al. reconstruct the $CH_4$ variability of the Early Holocene, from 11.6 to 8.5 ka before
1950, using a melt-refreeze extraction system coupled to a GC-FID analyser, which was newly
developed at Seoul National University (SNU). The new method is very briefly described in this paper
and not yet published. The authors show that the SNU data are in good agreement with two existing
benchmark records from the WAIS divide ice core, where the latter two were measured using *i*) a
similar technique (WAIS members 2015) and *ii*) a sample gas stream derived from a continuously
melted ice core, analysed by an optical instrument (Rosen et al., 2015).

Ji-Woong Yang et al. observe millennial $CH_4$ minima besides the 8.2 ka event, which have not been
identified in previous studies. The authors relate these $CH_4$ minima to events in other geological
records that indicate climate variability in the low and high latitudes of the Northern Hemisphere.
These records include: $\delta^{18}O$-$H_2O$ (NGRIP), ice rafted debris, $^{10}Be$, reflectance of Cariaco Basin
sediments, $\delta^{18}O$-$CaCO_3$ (speleothems) and $\Delta\varepsilon$. The authors show convincingly that the millennial $CH_4$
variability correlates with millennial variations in $\delta^{18}O$-$H_2O$ (NGRIP)/Greenland temperature, which is
a new and interesting finding. The authors furthermore discuss the relation of $CH_4$ with the other
records and suggest that Northern Hemispheric cooling and a concomitant southward shift of the
ITCZ created a teleconnection pattern of reduced intensities of Asian and Indian monsoons. Thereby,
the authors identified changes in $CH_4$ emissions from tropical wetlands as the most likely cause of
the millennial $CH_4$ minima. The authors claim that this mechanism cannot explain the $CH_4$ minima
around 10.2 ka alone.

In a next section, the authors review how the variability in external forcing during may cause an "El
Nino-like" climate. They discuss some relation in the climate system but how this is hypothetically
related to their $CH_4$ data remains unclear. The authors conclude this discussion cannot be developed
further as there is no ENSO reconstruction for the Early Holocene. The purpose of this section is not
entirely clear to me, also in the light of a range of existing publications on ENSO reconstructions (e.g.

Z. Liu et al., 2014, Nature, Vol. 515, p. 550-553)

In order to investigate the CH$_4$ record further, the authors calculate the inter-polar difference in CH$_4$
(IPD) using the presented SNU and previously published NEEM data. The calculated IPD record is
close to the range of previously published estimates, but exhibits interesting features on millennial
time scales. The authors suggest a high IPD at the onset of the Holocene, which then decreases
between 11.1 and 10.7 ka and increases again between 10.7 and 9.9 ka to previous levels. They
furthermore use a previously published box model to separate hemispheric and tropical CH$_4$
emissions. The authors discuss the variability of tropical and boreal source fractions over time in the
light of other studies and conclude that the IPD increase between 10.7 and 9.9 ka is a result from
Northern Hemispheric warming/thawing and expansion of boreal wetlands. This is in convincingly
good agreement with previous reconstructions of increased CH$_4$ emissions from boreal wetlands.

The study of Ji-Woong Yang et al. is a valuable contribution and in the scope of Climate of the Past. I
would recommend the publication of this manuscript but think that major revisions are required.

**General comments:**

1)      Ice core: The authors could provide more complete information on the ice core and related
logistics. For example, it is not explicitly mentioned that the samples were shipped from the USA to
South Korea (just names of institutions). Neither did the authors mention the year the ice core was
retrieved nor how long it was stored prior to analysis. Do the authors think that storage time affects
the analysis? Do the authors think that extended storage time could help with the handling of
samples from the brittle ice zone? The authors did not specifically address that their samples were
from the brittle ice region. However, I would strongly recommend to raise this point and how it
might or might not have affected the data.

*Now we added more details on samples and ice logistics as below:*

*"In this study we used ice samples from Siple Dome A (SDMA) deep ice core drilled from 1997 to 1999*
*on the Siple Coast, West Antarctica (81.65°S, 148.81°W; 621m elevation) (Taylor et al., 2004). The*
*SDMA samples were selected and cut at National Ice Core Lab (NICL), Denver from January to*
*February 2013. Since brittle zone of SDMA ice starts below 400 m depth (Gow and Meese, 2007) that*
*makes some part of ices fractured and/or cracked internally. Hence the samples were carefully*
*collected from unbroken ices during the sample preparation at NICL. The samples were packed in*
*isothermal foam boxes with numerous eutectic gels, and shipped to South Korea via expedited air*
*freight. Temperature loggers were enclosed within the isothermal boxes to record the temperature*
*change inside during the logistics, and it showed the temperatures were maintained below -25°C for*
*each box. The boxes were picked up directly just after custom clearance at the airport and then the*
*ice samples were stored in a walk-in freezer of Seoul National University (SNU) that maintained*
*below -20°C."*

2)      The analytical system: Even though the authors are preparing another manuscript on the
analytical method, a more detailed description of the analytical system would be helpful. For
example, the authors describe their melt-refreeze method as "traditional", though, I am in doubt
that most readers have a melt-refreeze technique in their lab, if they work in a lab at all. I find the
term "traditional" misleading, as it might be taken as a support for the performance of the method.

*We agree with the suggestion. We deleted the expression "traditional" and added more detailed description of the analytical method as follows:*

*"The air occluded in ice was extracted by melting and refreeze process under vacuum. Ice samples were prepared in a walk-in freezer in the morning of each experiment day, and trimmed the outermost >2 mm to eliminate possible contamination by ambient air during the storage. Then the samples were moved to the laboratory and placed in glass sample containers. The sample containers (sample flask hereafter) were custom-made glass flasks welded to stainless steel flange, and attached to the vacuum line with a copper gasket.  The sample flasks were partially submerged into a chilled ethanol bath during ice insert and attaching to the line for preventing temperature increase by laboratory air. After that all flasks were evacuated at least 40 minutes, the ice samples were melted by submerging the sample flasks in a warm water bath. Melting process was usually completed within 30 minutes. The sample flasks were then submerged in the cold ethanol bath chilled to around -82 ℃ more than 1 hour to refreeze. During the refreezing, we carried out GC pre-running (20 injections) and daily calibration that normally took 90 minutes. The ethanol temperature rises up to -55°C just after submerging the flasks, and it was chilled to below -65°C before expansion of the air in the flasks. The extracted air in the head space was expanded into a gas chromatograph (GC) equipped with a flame ionization detector (FID) to measure $CH_4$ mixing ratio. The GC linearity was tested by a series of inter-tank calibration using four working standard air cylinders (395.5, 721.3, 895.0, and 1384.9 ppb $CH_4$ in NOAA04 scale, Dlugokencky et al., 2005). Daily GC calibration curve was determined by measurements of a working standard having the closest $CH_4$ mixing ratio of expected value from the samples; in this study we used 721.3 ppb $CH_4$ standard for samples of the early Holocene. To account for system condition change throughout experiments (i.e., influence by water vapor), we calibrated with a standard air 6 times before and after sample measurements. The detailed configuration of the vacuum line and GC is described in another paper that is currently in preparation (Yang et al., in preparation). Gas extraction line was evacuated to under detection limit of 0.1 mTorr, and system leak check was done as a daily routine before sample preparation. If any pressure increase in $10^{-4}$ Torr scale for 30 seconds is detected, the experiment was stopped to figure out the leakage."*

I think the presentation of the analytical performance needs to be developed further. The authors describe their system as similar to Mitchell et al., (2011). To my understanding, the system of Mitchell et al., (2011) is the benchmark GC-FID system in the community, with an estimate for measurement uncertainty of ±2.8 ppb, based on the pooled standard deviation of replicate samples that were measured with extended periods of time between the analysis of each sample pair. That is, the uncertainty estimate of Mitchell et al., (2011) can be understood as "worst case scenario". The method of Yang et al. is presented with an uncertainty of ±1.0 ppb, which is determined by the pooled standard deviation of 8 replicate measurements. While I am more than happy to be convinced that a newly developed method is superior in performance to an existing method, I feel strongly that this claim has to be proven. I think a more detailed description of the method to determine the uncertainty estimate is required, especially because the method of Yang et al. is supposed to be by far superior than that of Mitchell et al., (2011). In the light of a total number of 295 samples measured for this study, the authors need to describe why they chose these specific 8 replicate samples to determine the analytical uncertainty and why they did not chose other samples.

*Our $CH_4$ data are presented by averaging the results of duplicate sample analysis from the same*

*depths. In order to estimate our data precision, we reanalyzed duplicate samples from the adjacent*
*ices (~10 cm depth difference) at 8 depth intervals 8-80 days after the first analysis (see Table R1).*
*The depth intervals were randomly chosen. The pooled standard deviation of the average of*
*duplicates from first and second measurements was 1.5 ppb (mean difference of 1.9 ppb).*
*We note that Mitchell et al. (2011) used different solubility correction methods. Applying the*
*solubility correction method described in Mitchell et al. (2011) yields pooled standard deviation of the*
*average of duplicates at adjusted samples of 1.4 ppb. One of the main differences of our analytical*
*system compared to the OSU one is that, while in OSU the blank test was done once in several days*
*and the systematic offset was interpolated between the blank days (Mitchell et al., 2011), we*
*measured at least 3-4 blank ices every day. By doing this, the systematic offset can be quantified*
*more precisely that accounts for daily changing conditions of instrument.*

**Table R1.** Summary of the first (original) and second (replicate) measurements from the depths used for system
reproducibility test.

| Depth (m) | 1st measurements | | 2nd measurements | | Difference | |
|---|---|---|---|---|---|---|
| | $CH_4$ (ppb) | Date (dd/mm/yy) | $CH_4$ (ppb) | Date (dd/mm/yy) | 1st – 2nd (ppb) | Time (days) |
| 523.15 | 631.8±0.1 | 27/01/14 | 632.7±1.6 | 24/02/14 | -0.9 | 29 |
| 530.95 | 663.0±2.2 | 03/02/14 | 664.7±1.6 | 24/02/14 | -1.7 | 22 |
| 558.295 | 676.7±2.1 | 14/03/14 | 679.3±4.5 | 02/04/14 | -2.6 | 20 |
| 559.85 | 682.0±5.8 | 03/02/14 | 684.7±1.9 | 26/03/14 | -2.7 | 52 |
| 561.15 | 685.0±0.8 | 14/03/14 | 683.7±3.1 | 02/04/14 | 1.3 | 20 |
| 562.407 | 682.8±0.8 | 26/03/14 | 684.3±1.5 | 02/04/14 | -1.5 | 8 |
| 575.913 | 676.5±0.2 | 07/02/14 | 679.3±3.8 | 28/03/14 | -2.8 | 50 |
| 578.15 | 676.2±4.3 | 04/02/14 | 674.9±2.3 | 24/04/14 | 1.3 | 80 |

Furthermore, the method includes several corrections, including blank (determined with bubble-free
ice, 5-15 ppb), gravitational fractionation (1.97±0.15 ppb) and correction for dissolved $CH_4$ (range
should be specified). These corrections include uncertainties that should be represented in the
uncertainty budget of the method. Furthermore, a blank of 5-15 ppb is enormous, both in absolute
values as well as in the range, especially when the total performance is stated with ±1.0 ppb. For
comparison, the method of Mitchell et al., (2011) has a blank of 1.1±0.5 ppb, while a blank of 5-14
ppb was interpreted as indicator of leakage. I think this issues need to be clearly demonstrated so
that superior performance can be claimed.

*The blank offset, which is calculated from the mean of the 3-4 blank results, reflects any errors by*
*contaminants, leaks, or different GC conditions. The exact cause of the blank offset is not clear, but*
*the important thing is that the 4 blank results agree well each other, yielding the intra-day standard*
*error of the mean of 2.0±1.0 ppb. Since every single data point is obtained by analysis of at least in*

*duplicate, the intra-day blank offset for one depth is reduced by a factor of √2. Comparede to inter-day blank offset of 5-14 ppb, the small intra-day blank offset implies that the daily offsets of the 10 sample flasks are rather systematic. The robustness of our final results was proven by reanalysis of duplicates at adjacent depths (< 10 cm) 8-80 days after the first analysis. The difference of mean of duplicates between the time intervals was 1.9 ppb on average (see the table above).*

*System leakage is unlikely. We carried out system leak check as a daily routine before starting gas extraction. If any pressure increase in $10^{-4}$ Torr scale for 30 seconds is detected, the experiment was stopped to figure out the leakage.*

Further issues that could be clarified include the uncertainty of the standard air (available for each NOAA04 cylinder), how the linearity of the system is controlled (additional air standards covering the analytical $CH_4$ range?) and what reason the authors base their decision on to subtract 3 ppb from the OSU data instead of adding 3 ppb to the SNU data or to take the 3 ppb as temporal signal? If the measurement uncertainty claimed by both institutes is realistic, both data should be on the NOAA04 scale and therefore be in close agreement. I think this is in the order of expected disagreement, but I feel the manuscript is stronger if these issues are clearly addressed.

*It seems that the 3 ppb offset is due to different correction for gas solubility effect. Unlike to Mitchell et al. (2011), we calculated the expected gas enrichment by assuming that equilibrium stateis achieved between air-water during the melting process. In OSU, the gas solubility effect is empirically estimated by measuring $CH_4$ mixing ratio from the air that is re-dissolved after melt-refreeze process (Mitchell et al., 2011). The directly-measured OSU correction method seems more realistic, but the uncertainty is large (~10%) due to small amount of air. After applying the OSU solubility correction scheme, the average difference between SNU and OSU reduces to less than 1 ppb. Therefore, for the better fitted composite record, we applied the OSU correction method to SNU data set.*

3)      $CH_4$ data: The authors mention 295 measurements. However, Figure 1 seems to show a much smaller number. If the displayed measurements are averages of duplicates or if other measurements are not displayed in Figure 1, the number would have to be revised.

*The number was corrected as below. Basically we made duplicates for all depths, and the average of the replicates was used for the data.*

*"We measured 295 samples from 156 depth intervals from 518.87 to 718.83 m, covering from 8.36 to 20.25 kyr BP after synchronizing to the Greenland Ice Core Chronology 2005 (GICC05). All samples were duplicated, so that our final $CH_4$ data were presented by averaging the results of duplicate sample analysis from the same depths and analytical uncertainty is deduced from standard error of the mean of duplicate pairs. "*

Otherwise, please clarify. How long is the overlapping period between OSU and SNU data? Maybe Figure 1 could show this in a detailed Figure?

*We added an enlarged figure in Figure 1 (See Figure R1). The overlapping period is from ~8.4 to ~9.1 ka. To make the SDMA $CH_4$ composite, we used the OSU data from 7.7 to 8.5 ka, due to higher mean temporal resolution of the OSU data.*

Figure 1 has a reference in the captions (Brook and Sowers, 2016) that is new to me and that I cannot find in the reference list.

*Reference was modified.*

The authors may or may not consider to show data from previous publications (e.g. Brook et al.,
2000, Flueckiger et al., 2002) to highlight the superiority of their temporal resolution.

*Figure 1 was modified as Figure R1 that includes the data from Tayler Dome, EPICA Dome C, and*
*Talos Dome (see below).*

[Figure]

*Figure R1. CH$_4$ reconstructions during the early Holocene. Top: new high-resolution Siple Dome CH$_4$*
*data (black, this study) compared with previous records from Taylor Dome (orange, Brook et al., 2000),*
*EPICA Dome C (grey, Loulergue et al., 2008), and Talos Dome (purple, Buiron et al., 2012). Bottom: Siple*
*Dome CH$_4$ records measured in OSU (blue, Ahn et al., 2014) and SNU (red, this study). Siple Dome*
*composite (black line) is plotted with WAIS Divide discrete (dark yellow, WAIS Divide project members,*
*2015) and continuous technique (green, Rhodes et al., 2015). Inset: Enlarged plot showing overlapped*
*interval between OSU and SNU Siple Dome data.*

4)      IPD: IPD is a powerful concept, but very one has to be very careful in its reconstruction and
interpretation. The authors mention the potential for ill-calculated IPDs based on errors in the gas
age scale of the CH$_4$ records. Therefore, the authors developed a tool to synchronize the CH$_4$ records,
which I think is a very good approach! However, it is not clear to me why the authors chose to
calculate the IPD based on the Siple Dome data? My concerns have several reasons:

i)      The authors state themselves that the histories of gas enclosure is more similar between
NEEM and WAIS. The authors state that, based on just this fact, the amplitude of CH$_4$ variations is
10-20 ppb larger in the WAIS than in the Siple Dome record. Therefore, I understand that both the
NEEM and the Siple Dome CH$_4$ records are altered by physical processes during gas enclosure that
are different for each core. I understand that a dampened amplitude in the Siple Dome CH$_4$ record would create a IPD variation. Ideally, gas enclosure effects should be identical in both records so that they would cancel. Since the WAIS record is in that sense more similar to the NEEM record than the

Siple Dome record is, I would suggest to use the WAIS record for the IPD reconstruction.

ii)         The WAIS record is of even higher temporal resolution than the Siple Dome record. I would expect that the IPD reconstruction based on NEEM and WAIS would be more robust.

iii)        The comparison of the $CH_4$ records from Siple Dome and WAIS in Figure 1 shows two periods (~10.5 and ~10.7 ka) where Siple Dome $CH_4$ exceeds WAIS $CH_4$ by up to ~20 ppb. During this period, the smoothed $CH_4$ variations (Figure 2) also show an ice core specific pattern of disagreement. Around 11 ka, the pattern of agreement is different. Here, the continuous $CH_4$ record from WAIS seems to agree better with $CH_4$ from Siple Dome, while the GC-FID record from WAIS

contains a number of samples that exceed the former records by ~20 ppb. The difference between the records during these times exceeds the stated measurement uncertainty by far. It is also important that the difference seems to be in the order of the presented IPD variability.

All of the above mentioned reasons directly impact on the IPD reconstructions. The choice of the authors to use their Siple Dome data for the IPD reconstruction is justified and understandable.

However, I fear that the interpretation is sensitive to the choice of $CH_4$ record so that this choice could impact on the IPD result for the above mentioned reasons. Therefore, I suggest to calculate the

IPD also using both WAIS records as three independent sets of IPD reconstructions as a sensitivity test. This will make the interpretation of reconstructed IPDs more robust and will furthermore give valuable insights into the IPD technique on high temporal resolution records for future studies.

Because this concerns one of the main outcomes of this manuscript, I would consider this essential.

*We appreciate the Referee #1 for her/his useful and reasonable comment on IPD reconstruction.*

*Calculating IPD from different ice core data sets would draw more objective conclusion. We provided*

*alternative IPD reconstructions from various records (NEEM discrete, NEEM continuous, WAIS*

*discrete, WAIS continuous, and Siple Dome discrete data) to test robustness of early Holocene IPD*

*trend (We denote the original IPD between NEEM discrete and SDMA discrete data as "IPD-1"*

*hereafter).*

*As mentioned in comment iii) above, the Siple- and WAIS ice core records do not agree at some*

*period, and this offset could lead erroneous IPD change. However, here we consider Siple data show*

*more reliable result by following reasons. First, it was shown that SNU Siple Dome discrete data (this*

*study) have a good agreement with OSU Siple Dome discrete data (Ahn et al., 2014) during the early*

*Holocene interval, while PSU (Penn State University) WAIS discrete data that covers most of the early*

*Holocene period show offset up to ~9.9 ppb to OSU data (Rhodes et al., 2015). Second, PSU WAIS*

*discrete data show a pooled standard deviation (for depth-adjacent samples) of 7.3 ppb (1sigma),*

*which is larger than SNU Siple data. In addition, NEEM continuous- and discrete data do not agree*

*well in some intervals (Figure R2), even though NEEM continuous data were calibrated against to*

*discrete measurements carried out at OSU (Chappellaz et al., 2013). Further, the NEEM continuous*

*record is not exactly "continuous", that may introduce uncertainty into synchronization. Hence, here*

*we regard the NEEM discrete, Siple Dome discrete, and WAIS Divide continuous records as more*

*reliable ones than the others during the early Holocene period (black and green curves in Figure R2).*

*Resulting IPD curve from NEEM discrete and WAIS continuous (green, IPD-2 hereafter) shows long-*

*term increase from the onset of Holocene to ~10.0 kyr BP. This indicates that contribution from*

*boreal sources increased during ~11.5 to 9.9 kyr BP, which is consistent with increase of northern extratropical temperatures and thermokarst lake CH$_4$ emissions. The revised 3-box model results obtained from IPD-1 (Table R1) show elevated emission from N-box and slight decline in T-box. However, since a small IPD increase during ~10.8-11.2 kyr BP observed in Siple Dome IPD is not supported by alternative IPDs, it remains unclear the short-term IPD change during this time period. Therefore, we modified descriptions and findings on early Holocene IPD trend.*

*Table R1. 3-box source distribution model results from IPD-1. Tropical (green, T) and boreal (red, N) boxes and boreal emission fraction (N/(T+N+S)) are compared with previous results. Errors denote 95% confidence interval.*

| Ref. | N box | T box | N/(N+T+S) ratio |
|---|---|---|---|
| (ka) | (Tg yr$^{-1}$) | | (%) |
| Brook et al., 2000 (9.5-11.5 ka) | 64 ± 5 | 123 ± 8 | 32 ± 3 |
| Chappellaz et al., 1997 (9.5-11.5 ka) | 66 ± 8 | 120 ± 9 | 33 ± 3 |
| This study (9.5-11.5  ka) | 66 ± 4  | 120 ± 4  | 33 ± 2  |
| This study (11.5 ka) | 57 ± 6 | 119 ± 11 | 30 ± 4 |
| This study (9.9 ka) | 70 ± 4  | 115 ± 4  | 35 ± 2  |

*Table R2. 3-box source distribution model results from IPD-2.*

| Ref. | N box | T box | Boreal source fraction |
|---|---|---|---|
| (ka) | (Tg yr$^{-1}$) | | (%) |
| This study (11.5 ka) | 60 ± 7 | 134 ± 16 | 29 ± 4 |
| This study (10.0 ka) | 71 ± 7 | 115 ± 11 | 35 ± 4 |

[Figure]

*Figure R2. Inter-polar difference (IPD) reconstructions. Top: high resolution $CH_4$ records from*
*Greenland and Antarctica, synchronized to NEEM gas age scale by Monte Carlo procedure. Middle:*
*Millennial-scale IPD trends derived from various pairs of data set. Shaded area indicates 95%*
*significance interval. Bottom: Proxy-based temperature reconstruction for northern mid to high*
*latitude and boreal $CH_4$ emission from northern thermokarst lakes.*

[Figure]

*Figure R3. Same as Figure R2, but shown are IPD-1 (black, NEEM discrete – SDMA discrete) and IPD-2 (green, NEEM discrete – WAIS continuous) only.*

5)       Consideration of significant publications: *i*) The authors claim that no correlation between CH$_4$ and tropical monsoon signals has been reported on shorter time scales, however, I feel this is not accurate. Both Cruz et al., (2005, Nature, Vol, 434, p. 63-65) and Sperlich et al., (2015, Global Biogeochemical Cycles, 29) have related CH$_4$ and $\delta^{13}$C-CH$_4$ records to South American speleothem records, respectively. Both publications would principally support the interpretation of this study. *ii*) The authors discuss that their IPD reconstruction suggests increasing boreal source fractions during the early Holocene and support their finding with studies on boreal wetland dynamics. However, their finding of increased boreal source fractions is in line with the interpretation of $\delta^{13}$C-CH$_4$ data by Fischer et al., (2008) and Sowers, (2010). Again, both publications would principally support the interpretation of this study while Sowers (2010) had the same finding for the early Holocene previously. *iii*) The authors stated that there is currently no ENSO reconstruction for the early Holocene, even though a range of them exist (e.g. Z. Liu et al., 2014, Nature, Vol. 515, p. 550-553, Clement et al., 1999, Paleoceanography, Vol. 14, p. 441-456).

*We checked the suggested publications and take them into consideration for further developing discussions.*

*i)     The suggested citations were added:*

*P37 L28-31: "It has been found that tropical monsoon activities were closely related to orbital-scale $CH_4$ change (e.g., Brook et al., 1996; Chappellaz et al., 1990), especially reported are Asian monsoon (e.g., Loulergue et al., 2008) and South American monsoon (e.g., Cruz et al., 2005)."*

*P44 L14-15: "Sperlich et al. (2015) also found that a sharp $CH_4$ peak at Greenland Interstadial 21.2 (~85 ka) was occurred by emission from Asian and South American wetlands."*

*ii)     We removed the discussion on YD-Holocene transition and isotopic mass balance because it seems out of main focus of this paper.*

*iii)     We appreciated the suggestion, but unfortunately, the mentioned publications (Liu et al., 2014; Clement et al., 1999) are dealing with climate modelling result, not proxy-based reconstructions. Instead, we cited Kirby et al. (2010) for PDO, and Moy et al. (2002) and Rodbell et al. (1999) for Holocene ENSO reconstructions. According to Holocene ENSO activity reconstructions by Moy et al. (2002), no ENSO event was recorded during the early Holocene until around 7 ka, except weak ENSO events during 10.4-10.1 ka, where abrupt $CH_4$ decrease is observed without significant changes in ITCZ and NH monsoon intensities. Mitchell et al. (2011) observed a significant positive correlation between $CH_4$ and PDO variability during the late Holocene. It has been reported that Pacific Decadal Oscillation (PDO) modulates the wet/dry impact of ENSO depending on phase relationship between ENSO and PDO (e.g., Wang et al., 2014 and references therein). The Holocene PDO reconstruction from sediment grain size analysis by Kirby et al. (2010) shows PDO-related drying intervals in North America during 9.5-9.1, 8.9-8.6, and 8.3-7.8 ka, which overlap the $CH_4$ minima at 8.2 and 9.3 ka present in this study.*

6)     Chosen data filtering technique: I suggest to provide more information why a 250-year window width was used. What is the effect of other window widths on the data you use and on your resulting interpretation? The authors state on p5L2-5 that the 250 year window was also used in other studies. However, that is not necessarily a satisfying justification. The window width should be carefully chosen in dependence on the time scales you are investigating.

*Power spectrum (REDFIT, Schulze and Mudelsee, 2002) of Siple Dome $CH_4$ data indicates moderate (over 90% significance level) powers at ~1340, 401, 309 and 96-year period. Considering ~42 years of gas age distribution of Siple Dome (Ahn et al., 2014), it is not reliable to study centennial scale variability. Thus we applied 250-year window to smooth out any high frequency component having shorter period than 309 years. We added below paragraph:*

*"To extract millennial-scale variability, we carried out spectral analysis using REDFIT program (Schulze and Mudelsee, 2002) and moderate (over 90% significance level) powers were found at ~1340, 401, 309, and 96-year period. Given the ~42 years of gas age distribution of SDMA (Ahn et al., 2014), it would not reliable to study centennial scale variability. Thus we produced annual data by interpolation and then calculated 250-year running means to smooth high frequency components having shorter period than 309-year. Then the smoothed time series was filtered with a high-pass*

*window (cut off period of 1800 years) to study millennial scale variability throughout the early Holocene."*

7)      Sometimes, I have trouble to understand the point the authors intend to make, e.g. p1, 12–18; p5, 18–29; p6, 25–p7, 5; p10, 6–12, p22, 1-2. Please consider re-wording.

*We thoroughly revised the mentioned sentences.*

8)      I have difficulties to follow the discussion and the display of $CH_4$ and other records. For example, the authors discuss why there is agreement/disagreement between some records within the uncertainty of the age model. However, I find it tough to see this in Figure 2. For example, the $CH_4$ minima are highlighted with yellow bars. During 9.4 and 10.2 ka, the yellow bars include both local minima and local maxima of the other records, e.g. of the Cariaco Basin record, of $\Delta \varepsilon$, of Dongge Cave. Other local extrema, e.g. in the Cariaco Basin record have no correspondence in $CH_4$ during 8.5-8.7 ka or 10.5 ka, which is not mentioned at all. Therefore, I feel this discussion needs to be further developed to provide more guidance to the reader. Is [10]Be really important here? Sometimes it seems to correlate, other times it is anti-correlated. Could figure clarity increase without it?

*Not every single variation of the Cariaco Basin reflectance record corresponds to abrupt $CH_4$ change, because the Cariaco Basin record could also have local signal. What we wanted to show in this Figure is that abrupt cooling occurred around Greenland changes tropical rain belts and hence $CH_4$ emission. [10]Be and IRD proxies were included to discuss trigger of abrupt Greenland cooling, but data resolution and dating uncertainty (±100 to 150 years, Bond et al., 2001) prevent us from drawing rigorous conclusion. By taking into consideration with the comment #12 (Figure R4), we modifiedthe Figure 2 and relevant discussions.*

[Figure]

*Figure R4. Millennial scale climate variability. All proxies present here were smoothed by 250-year running average and detrended by high-pass filter with 1/1800-year window. (a) Siple Dome CH$_4$ (red, this study), Greenland $^{10}$Be (dark yellow, Finkel and Nishizumii, 1997), North Atlantic IRD stack (grey, Bond et al., 2001). Also shown are WAIS Divide CH$_4$ data by discrete (cyan, Buizert et al., 2015) and continuous (yellow green, Rhodes et al., 2015) technique. (b) NGRIP stable water isotope ratio (blue, Rasmussen et al., 2006) and Cariaco Basin reflectance (orange, Deplazes et al., 2013). (c) Qunf Cave speleothem oxygen isotope (Fleitmann et al., 2007). (d) Dongge Cave speleothem oxygen isotope (green, Dykoski et al., 2005; dark yellow, Wang et al., 2005). (e) Gulf of Guinea planktonic Ba/Ca ratio (Weldeab et al., 2007). (f) Lapa Grande Cave speleothem oxygen isotope (purple, Strikis et al., 2011). Age tie-points used to adjust Siple Dome and WAIS Divide CH$_4$ data to GICC05 scale are marked in black triangles.*

9) Figure 2: Presented is $CH_4$ anomaly. I don't see an advantage of anomaly over $CH_4$ mole fractions. Also, how is anomaly of 0 defined for each record?

*We refer "anomaly" in Figure 2 as detrended time series after filtered with high-pass window. We used "detrended data" instead of "anomaly" to clarify it.*

10) Structure: I would suggest to avoid three levels, e.g. 3.1.1 and 3.1.2 but make 3. Millennial scale variability, 4. Latitudinal distribution…. to keep structure with max. two levels.

*We simplified the structure in two levels.*

11) In many places of the manuscript, the author review literature, e.g. on pattern of climate teleconnections, for which they allow extensive text sections. While I think it is important to review in such detail, I feel the authors could improve their discussion of how they think this is linked/relevant to their $CH_4$ interpretation. A good example for this is the entire section 3.1.3. I would like to encourage the authors to consider this throughout the entire manuscript, even though this probably means either adding to, or re-writing many sections of the manuscript.

*We re-structured the 3.1.2. section.*

12) Understanding the variability in tropical wetlands is crucial for the understanding of $CH_4$ source regions and tropical $CH_4$ fractions. (The same rule applies for boreal wetlands.) The authors fully acknowledge this throughout the manuscript. However, I note that the authors exclusively focus their interpretation on Asian/Indian monsoon systems. It has been described previously that the African monsoon system and wetland extension changed tremendously throughout the Holocene (e.g. Sahara region etc). There are also several publications on South American monsoon systems besides the Cariaco Basin reflectance, which I understand the authors only use as proxy for ITCZ migration, but not for their interpretation of hydrological changes in South American wetlands. Including further records (e.g. Cruz 2005) might allow for a more comprehensive evaluation of hydrological changes. Based on $\delta^{13}$C-$CH_4$ data, the South American monsoon system has recently been suggested to be a controlling factor in rapid $CH_4$ changes leading up to DO21 (Sperlich 2015). I would strongly recommend to either include a complete representation of tropical wetlands or to discuss why you think monsoon systems other than the Asian/Indian are not relevant.

*It is possible that the monsoon system of other regions could play a role in $CH_4$ change. However, considering that mean position of ITCZ was moved to northward than glacial condition (e.g., Deplazes et al., 2013), the rainfall and $CH_4$ emission of Asian/Indian monsoon regions should be stronger than southern hemisphere monsoon regions, for example, South America and Africa. This makes boundary condition different from those arguing that Southern Hemisphere emission leads abrupt $CH_4$ increase during Heinrich Stadial 1, 2, 4, and 5 (Rhodes et al., 2015) or DO21 (Sperlich et al., 2015). Both studies are dealing with abrupt $CH_4$ change under glacial condition. In the meanwhile, one of main conclusion of our paper is that abrupt cooling in Greenland lead ITCZ mean position change and tropical $CH_4$ emission. Therefore, it seems sufficient to support the idea with the Asian/Indian monsoon records and Cariaco Basin reflectance data.*

*Lapa Grande Cave (Eastern Brazil, 14°25'22"S) records demonstrate clear $\delta^{18}O$ depletion at 8.2 and 9.2 kyr BP (Strikis et al., 2011), indicating that ITCZ rain belts temporarily migrated southward and induced wet condition over eastern Amazonia region. This evidence agrees well with Cariaco Basin rainfall reconstructions and does not go back beyond ~10.2 kyr BP, therefore we thought that adding the South American monsoon proxy would not improve our conclusions. The monsoon record of*

*northern Peru (El Condor and Cueva del diamante, Cheng et al., 2013) and southern Brazil (Botuvera,*
*27°13'24"S, Cruz et al., 2005) are not highly resolved enough to see abrupt changes. Otherwise, the*
*reconstructions of West African monsoon (Gulf of Guinea, Ba/Ca ratio of planktonic foraminifera,*
*Weldeab et al., 2007) and Indian monsoon entire early Holocene (Qunf Cave, Fleitmann et al., 2007)*
*track well the millennial scale CH$_4$ minima. Furthermore, Australian-Indonesian monsoon rainfall*
*records (not shown) from Borneo (Partin et al., 2007) and Liang Luar (Griffiths et al., 2009) do not*
*show clear evidence of abrupt change that coincides with Greenland cooling and CH$_4$ decrease. This*
*may reflect that the rainfall in tropical western Pacific region was affected by both northern- and*
*southern hemispheric climate change (Griffiths et al., 2009; Partin et al., 2007).*

13)     Data availability: I understand that Copernicus has developed a new policy for authors to
provide either descriptions on data access, or to provide the data through international data-bases
or supplementary information. Copernicus requires a dedicated section that describes this in detail,
which the authors might want to consider during their revisions.

*Thanks for this information. If our manuscript is accepted to publish, we will make our new data*
*available online in NOAA Paleoclimate Data Center and PANGAEA Data repository.*

**Specific comments**

The manuscript may be subject to considerable re-writing. Therefore, the specific comments will not
include comments on grammar, wording or writing that might be subject to change. Since I am not a
native English language speaker myself, I am not sure to what extend my comments would help to
make it better or worse. Please understand suggested re-formulations as suggestions, only.

p1L1: Understanding processes controlling atmospheric methane

*The sentence was changed as below:*

*"Understanding  processes controlling atmospheric methane*
*(CH$_4$) mixing ratio is crucial to predict and mitigate the future climate change.*

p2L2: reference Daniau et al., is it possible to provide a reference on palaeo-fire or a reference that is
more specific on pyrogenic gas emissions?

*Citation was changed.*

p2L5: sink strength and light availability? e.g. polar winter

*We added below sentence in Introduction:*

*P36 L35-36: "Further, since the OH is produced by photo-dissociation reaction, the sink strength is*
*affected by light availability and tropospheric ozone (e.g., Levy, 1971). Polar winters may affect the*
*CH$_4$ sink strength by reducing OH production rate, but this seasonal-scale cycles are not resolvable in*
*ice core records due to gas dispersion in firn layers".*

p2L16: Lisiecki and Raymo 2005, though this reference is not on CH$_4$, general I think a reference on
CH$_4$ and Northern Hemisphere temperature would be useful here

*We changed the citation as below and added the reference:*

*(e.g., Brook et al., 2000; Chappellaz et al., 1993; Huber et al., 2006; Loulergue et al., 2008).*

*Huber, C., Leuenberger, M., Spahni, R., Fluckiger, J., Schwander, J., Stocker, T. F., Johnsen, S.,*

*Landais, A., and Jouzel, J.: Isotope calibrated Greenland temperature record over Marine Isotope*

*Stage 3 and its relationship to $CH_4$, Earth Planet. Sci. Lett., 243, 504-519, 2006.*

p2L18: See comment 5) in general comments

*See Author Comment to general comment #5.*

p2L18-19: too weak. The correlation between $CH_4$ and NH temperature ($\delta^{18}$O-ice) is well established

*It is true for longer time scales. The $CH_4 - \delta^{18}O_{ice}$ of Greenland has yet been revealed in shorter time*

*scales during the early Holocene period.*

p3L15: here and everywhere else: It is recommended to restrict the use of "concentrations" to public debate but to use "mole fraction" or "mixing ratio" in scientific literature (WMO, recommendations of GGMT experts)

*This was updated as suggested.*

p3L17: here and everywhere else, $\delta^nX$, the $\delta$ is supposed to be *italicised*, same for Δ    *Δ* (Coplen

2011, DOI: 10.1002/rcm.5129)

*This was updated as suggested.*

p3L27: there is still some ice left in Greenland

*We modified the sentence as below:*

*" It should be noted that environmental boundary conditions of the early Holocene were not*

*identical to those of the late Holocene.*

* The global*

*sea level was rising throughout the early Holocene and there is still some ice left in Greenland."*

p3L30: ensure stated, used and displayed sample numbers agree, give age interval with depth

*Please refer to our response to General Comment #3.*

p3L32: (NICL, city, state, country), (SNU, city, state, country)

*This was modified as suggested.*

p4L3: "are described in…", referring to a paper that is currently in preparation as XYZ et al., (in prep.)

seems strange as it is not useful to look it up. "A manuscript that describes the method in detail is currently in preparation."

*We removed this sentence and added more details on our analytical system. Instead, we added below*

*phrases regarding the detailed technical settings of GC and calculation of correction factors:*

*P41 L18-19: "The detailed configuration of the vacuum line and GC is described in another paper that*

*is currently in preparation (Yang et al., in preparation)."*

*P42 L16-17: "Further details on correction method will be discussed in our manuscript in preparation*

*(Yang et al., in preparation)."*

p4L7: was the standard air added before or after the bubble-free ice was melted?

*The standard air was injected before melting the bubble-free ice. We revised the wording for clarity.*

p4L7: "traditional" melt-refreeze seems misleading to me, traditional can be left out

*We removed the word "traditional".*

p4L7-17: see point 2) in general comments

*See Author Comment to general comment #2.*

p4L17: provide reference how you calculated gravitational fractionation

*Relevant citation and reference was added: Craig et al., 1988*

*Craig, H., Horibe, Y., and Sowers, T.: Gravitational separation of gases and isotopes in polar ice caps,*
*Science, 242, 1675-1678, 1988.*

p4L19: provide reference for GICC05 time scale

*Reference was added: Rasmussen et al. (2006)*

p4L25: "one of the high resolution data sets" sounds vague and strange to me, where do you draw
the line between high and low resolution? The temporal resolution of your data is higher than some
but lower than both WAIS records. "It has the currently second/third highest temporal resolution of
Antarctic $CH_4$ records covering the early Holocene."

*The sentence was modified accordingly as below:*

*"Our new Siple Dome $CH_4$  data has the currently*
*third highest temporal resolution of Antarctic $CH_4$ records covering *
* the early Holocene after*
*the WAIS Divide continuous (~2 years, Rhodes et al., 2015) and discrete (~20 years, WAIS Divide*
*members, 2015) records."*

p4L26: develop a more complete representation of analytical uncertainty

*Please refer to our response to general comment #2, as well as our response to the Referee #2.*

p4L30: describe the overlapping interval and describe why you think the OSU record should be
adjusted to match the SNU data. Why not the other way around, why not accepted as real
difference? Both should be on NOAA04?

*We attribute the SNU-OSU offset to different correction method for solubility effect. After applying*
*revised solubility correction used in OSU (Mitchell et al., 2011), the mean offset reduces 0.6 ppb*
*which lies well within analytical uncertainty of both institutes.*

p5L3: this comparison example only makes sense if you look at variations on similar time scales.
Otherwise the argument that you use the same filter as has been applied for the WAIS record is not
valuable, but could be misleading.

*Here we hesitate to remove the comparison. As we describe in this sentence and Figure 2, to ensure*
*robustness of millennial scale variability from other Antarctic ice core record is important. We used*
*filtered data because both data set have different time resolution, and they show different*
*fluctuations in short time scales as shown in Figure 1.*

p5L14: add references that show anthropogenic signal in LPIH CH$_4$, e.g. Ferretti 2005, Mischler 2009,
Sapart 2012)

*We add the relevant citations:*

*"Mitchell et al. (2011) found no significant correlation with Greenland climate in multi-decadal time*
*scale during the late pre-industrial Holocene (LPIH), possibly due to growing anthropogenic*
*emissions (e.g., Ferretti et al., 2005; Mischler et al., 2009; Mitchell et al., 2013; Sapart et al., 2012)."*

p5L16: "even though this conclusion is less robust as there are no age tie-points…"
*The sentence was modified as suggested:*

*"In contrast, we observe a moderate positive correlation between the Siple Dome CH$_4$ and NGRIP*
*$\delta^{18}O_{ice}$ during the early Holocene, which implies that natural CH$_4$ budget is closely connected with*
*Greenland climate in millennial timescales, even though this conclusion is less robust as there is no*
*age tie-point between the 8.2 ka episode and the Preboreal oscillation (Fig. S1)."*

p5L21-22: shift this sentence to after the following sentence to keep logical flow from NH to tropics
*Done.*

p5L25-29: describe the meaning for CH$_4$, develop the discussion towards CH$_4$, what does a ITCZ shift
mean for South American CH$_4$ source regions?

*South American monsoon proxy shows concurrent intensification at similar timings of ~8.2 and 9.3 ka*
*CH$_4$ drop (Strikis et al., 2011), where precipitations in Cariaco Basin and other NH monsoon regions*
*decreased. For older period it is difficult to draw robust conclusion due to lack of high resolution data*
*at the timing of preparing this paper. The southward ITCZ migration may lead reduction in NH*
*wetland emission and enhanced in SH. However, given the orbital parameters that show maximum*
*summer insolation in NH while minimum in SH during the early Holocene, it can be inferred that*
*contribution of SH wetland emission was relatively low and cancelled by reduction of NH emission.*
*We revised the paragraph as below:*

*" The $^{18}$O enrichments of Asian (Dongge) and Indian (Hoti and Qunf) cave stalagmites*
*occurred at similar timing with abrupt cooling in Greenland, which indicate the reduction of*
*monsoonal rainfall at northern tropical wetlands. The speleothem records from Chinese and Oman*
*caves seem to lag by ~100-200 years after the CH$_4$ change at ~9.3 ka, but this lies within*
*chronological uncertainties of ~200-400 years at around 9.0 ka (Dykoski et al., 2005; Fleitmann et al.,*
*2007). Moreover, sediment Ba/Ca ratio from Gulf of Guinea demonstrates concurrent decrease of*
*west African monsoon (Weldeab et al., 2007). These evidences indicate that precipitation over the*
*major wetland area was reduced and in turn it would lower the wetland CH$_4$ emissions in NH. In the*
*meanwhile, an inverse relationship is observed from the Eastern Brazilian speleothem data (Lapa*
*Grande Cave, Strikis et al., 2011) that demonstrate the increasing of precipitation at the time of*
*abrupt CH$_4$ drop occurred as a result of southward migration of ITCZ. Considering the orbital*
*parameter that shows maximum summer insolation in NH while minimum in SH during the early*
*Holocene, it can be inferred that contribution of SH wetland emission was relatively low and*
*cancelled by reduction of NH emission."*

p6L4: there are other monsoon systems that Asian/Indian that should be considered

*See our response to general comment #12.*

p6L10: the monsoon intensity change. (delete Asian, include other monsoon systems)

*We modified the sentence as below:*

*"Given the weak reduction of precipitation over the Asian, Indian, and African monsoon regions*
*(Figure R4),  it may imply $CH_4$  drop was controlled by other processes than the *
*monsoon intensity change."*

p6L20: even though the Cariaco Basin record is shown, it is presented only as indicator for
ITCZ migration, without direct connection to $CH_4$. I feel the assumed passiveness of South
American $CH_4$ source regions during the study period might not be a natural assumption
and should be explained.

*Considering the orbital configurations that show maximum summer insolation in NH while*
*minimum in SH, contribution of SH wetland emission had to be reduced during the early*
*Holocene.*

p6L23-p7L5: describe relevance for $CH_4$, what is the $CH_4$ controlling process chain? A sentence that
says "the proxies show this and that which could explain the increase/decrease in $CH_4$ during time
period XY".

*We added more details on proxies and its relevance to $CH_4$ change. Adding following sentence at the*
*end of the paragraph will be more explained: "Above evidences indicate that the early Holocene $CH_4$*
*minima were triggered by anomalous low solar activity, but future study is warranted to draw more*
*conclusive result."*

*The entire paragraph was revised as below:*

*"Below it is discussed the possible impact by external forcing. Bond et al. (1997) reported four large*
*ice-rafted debris (IRD) drifts occurred at ~8.1, 9.4, 10.3 and 11.1 ka caused by surface cooling of North*
*Atlantic Ocean. They found that the ocean surface cooling and the IRD events are closely related to*
*cooling over the Greenland. Figure 2 shows that each IRD event (maxima in hematite stained grain)*
*occurred concurrently with minima of NGRIP $\delta^{18}O_{ice}$ record within age uncertainty. Then the Greenland*
*cooling leads southward shift of ITCZ and in turn it changes wetland $CH_4$ emission in low latitudes.*
*Bond et al. (2001) found that IRD maxima during the Holocene coincide with solar activity minima. The*
*authors suggested that solar forcing could affect the climate change around the North Atlantic Ocean*
*(and Greenland), through amplification by changes in sea ice and/or deep water formation. A close*
*interplay between solar activity and monsoon intensity has been observed in previous studies using*
*the Chinese and Oman speleothem records during the Holocene (Neff et al., 2001; Wang et al., 2005;*
*Gupta et al., 2005), even on multi-decadal time scales (Agnihotri et al., 2002). However, the forcing*
*mechanism of solar activity on the North Atlantic and global climate is not well understood. Jiang et*
*al. (2015) found positive correlations between North Atlantic SST and solar forcings inferred from*
*plaeoproxies ($^{14}C$ and $^{10}Be$) for he last 4000 years, while the correlation disappears during the mid- and*
*early Holocene. They hypothesized that climate sensitivity to solar forcing is high for cooler climate. As*
*evidenced above, the early Holocene $CH_4$ minima were likely triggered by anomalous low solar activity,*
*but future study is needed to make it more conclusive."*

p7L8-9: you could add Cruz et al., 2005 to the reference list, as they discussed the interplay of solar radiation, monsoon intensity and $CH_4$ mole fractions

*Here we hesitate to cite Cruz et al. (2005) here because the time scale they are dealing with is quite*

*different.*

p7L16-17: see above comment regarding references on ENSO variability during Holocene period

*Please refer to our response to general comment #5.*

p7L23: provide temporal resolution of NEEM record, 1 sample in how many years?

*Done.*

p7L24: consider IPD reconstructions with $CH_4$ records from WAIS

*We added alternative IPD reconstructions from more reliable records; NEEM discrete, WAIS*

*continuous, and Siple Dome discrete data. Please refer to our response to Referee #2.*

p7L25-30: NICE approach!

p8L2: …show an increase by XYZ ppb from…

*We removed that sentence.*

p8L4: …in both hemispheres during…

*We removed this sentence, but we changed the word "poles" to "hemispheres" in other phrases.*

p8L6: …from both hemispheres and…

*This sentence was modified as below:*

*" By using our new IPD and the*

*reliable highly resolved $CH_4$ records (NEEM discrete – SDMA discrete / WAIS continuous), we ran a*

*simple 3-box $CH_4$ source distribution model to quantify how much the boreal and tropical source*

*strengths were changed."*

p8L8-9: extra-tropical latitudes (30N or 30S is not high latitude, rather extra-tropical)

*This was modified.*

*"Briefly, the model contains 3 boxes; northern  extra-tropical latitude (30-90°N, N-box), tropical*

*(30°S-30°N, T-box), and southern  extra-tropical latitude boxes (30-90°S, S-box). $CH_4$*

*concentrations in 3 boxes (in Tg box$^{-1}$) were determined from $CH_4$ mixing ratio of Antarctica and*

*Greenland."*

p8L13-14: develop description of model assumptions and impacts, e.g. what life times did you assume and why? did you tune life times to match previous flux estimates?

*We used the identical parameters described in Chappellaz et al. (1997) and Brook et al. (2000).*

*Previous estimates are averaged values throughout the early Holocene, so that it is difficult to*

*compare and match our higher resolution IPD to the previous results.*

p8L15-16: quantify and discuss flux estimates, otherwise meaningless

*Done.*

p8L18: in tropical emissions by XYZ Tg. (quantify)

*Done.*

p8L25-30: increased $CH_4$ emissions from boreal wetlands were previously suggested by Sowers 2010, that agreement should be acknowledged

*Please refer to our response to General Comment #5.*

p8L29: explain "conventional" northern $CH_4$ emissions

1. *We deleted this sentence.*

2. p9L1: the isotope records are already published and need to be acknowledged (Sowers, 2010). these
3. isotope data are available and should be added to the figures of this manuscript.

4. *We thoroughly revised the paragraph where we cited the Sowers 2010 publication.*

5. p9L10-15: therefore, IPD should be calculated with WAIS records as well
6. *See our response above and Figure R2 and R3.*
7. p9L30: there is also no explanation for the drop in IPD if I am not mistaken?
8. *The slight decrease in IPD between ~10.3 and 10.8 ka is not observed in the alternative IPDs.*
9. *Although it is argued above that Siple Dome CH$_4$ is more reliable data than others, we cannot rule*
10. *out the possibility that the Siple data deteriorate during this period. At present it is difficult to say the*
11. *drop in our IPD represents the real change or not.*
12. p10L5-6: why can the 10.2 ka event not be explained by low latitude hydrology, but the other events
13. can? I feel this is a section where great care has to be taken to prevent from over interpretation.
14. There is no quantitative estimate for low latitude emissions during other events, i am not convinced
15. that the presented records allow for a partial explanation of the CH$_4$ minima and that there is only a
16. missing bit. I would recommend to re-formulate. Even if the revised IPD reconstruction supports the
17. current discussion, this might seem as two results are made to fit together. I feel this can be toned
18. down and still be strong a conclusion.

19. *The 10.2 ka event does not bring corresponding abrupt change in Cariaco Basin record, Dongge Cave*
20. *speleothem data, and Siple Dome Δε$_{LAND}$. Thus we speculated that CH$_4$ drop at 10.2 ka was caused by*
21. *source reduction in boreal regions and/or Southern Hemisphere.*

22. *Assessing the latitudinal source change from IPD is not reliable, given each ice core gas record has*
23. *had own characteristic smoothing process at firn, because it could lead erroneous result for abrupt*
24. *changing intervals such as 10.2 ka event.*

25. p10L6-12: I am not sure I understand this section

26. *We reworded the sentences to better clarity.*

27. p10L11: the quantification 20-40 ppb is mentioned for the first time here. The conclusions cannot
28. include information that have not been presented earlier in the manuscript. I am not sure how you
29. quantify ppb changes? Is that from the box model?

30. *The amplitude of each millennial scale CH$_4$ drop is quantified by CH$_4$ anomaly curve shown in Figure*
31. *2. We added this into earlier part of manuscript.*

32. p16L19: I didn't check all references, but Sowers 2010 is not correct. This is "Atmospheric methane
33. isotope records covering the Holocene period, Quaternary Science Reviews 29, 213-221, 2010". The
34. title/journal name in your references refers to his 2006 paper

35. *Done.*

36. p18L4: add reference to the list or check reference

37. *Done.*

38. p18F1: show overlapping period, show minor ticks on both axes

*Please see our response to the General Comment #1.*

p19F2: define 0 ppb in anomaly or show $CH_4$ mole fractions, check width of yellow bars, can be confusingly wide

*The Figure was updated.*

p20F3: add IPD with WAIS data

*Please see our response above, and Figure R2 and R3.*

p22T1: caption is confusing to me, also what is this table supposed to add? how can this agreement be explained, life time?

*The table and caption was updated according to the revised IPD reconstructions.*

**_Referee Comment #2_**----------------------------------------------------------------------------------------------------------------

**_Interactive comment on_ "Atmospheric methane control mechanisms**

**during the early Holocene"** **_by_ Ji-Woong Yang et al.**

**Anonymous Referee #2**

Overall assessment

The manuscript provides new high-resolution CH4 data from the Siple Dome ice core over the time interval 8.5-11.5 kyr BP, extending previous work by Ahn et al., 2014. The data quality is generally very good (see comments below) and I commend the authors for their painstaking work to provide high-resolution data sets using discrete CH4 analyses. The data is interpreted with respect to millennial climate variations during this time interval and, based on correlation with other climate proxy data, a suggestive hypothesis about the influence of changes in the ITCZ is presented to explain the millennial variability in CH4.

Finally, the interpolar CH4 difference (IPD) is calculated using Greenland data from the literature and this difference is then analyzed using a simple three-box model. As outlined below, I have some fundamental questions about the reliability of the inferred IPD, which subsequently has also important implications for its interpretation. This prevents me from recommending the manuscript for publication in CP in its current form despite the nice data. Moreover, the quality of the manuscript in terms of the use of the English language has to be considerably improved. With a native English speaker on the author list, I see no problem that this can be achieved. In summary, after additional work I am confident that the manuscript will become suitable for publication in CP in a resubmitted version.

General comments a) comparison of early and late Holocene CH4 variations: In the introduction the authors make the important point that only the early Holocene period allows us to study the natural CH4 variability on centennial to millennial time scales. Unfortunately, the authors do not follow up on this in the discussion. It would be interesting to compare the centennial and millennial variability in CH4 concentrations in the early Holocene (as documented in the Siple Dome, WAIS (including the continuous CH4 data by Rhodes et al., 2015) and NEEM record with the late Holocene as documented in WAIS by Mitchell et al., 2013. Are the amplitudes of this variability different and if so, is that due to an anthropogenic influence in the late Holocene or related to the changes in seasonal and geographical distribution of solar insolation (due to orbital parameter changes) between the early and late Holocene? Note that summer insolation in high northern latitudes was several tens of W/m2 higher in the early Holocene. For this analysis it may be beneficial to use the continuous WAIS data instead of the Siple Dome data to calculate a record with comparable resolution to the Mitchell data and to compare CH4 frequency spectra for the early and late Holocene.

*We appreciate to Referee #2 for this important comment. We added a dedicated chapter based on discussion below:*

*"We compared amplitude of $CH_4$ variability between the early- and the late Holocene in multi centennial to millennial time scales. Figure 5 shows amplitude spectrum and root mean square (RMS) amplitude for the early Holocene and the late Holocene, respectively. The amplitude of the early Holocene $CH_4$ change is ~10 ppb and does not change greater except for PBO and the 8.2 ka event, while the late Holocene spectrum shows smaller amplitude than early Holocene for shorter-term change and larger for longer-term fluctuation. Late Pre-Industrial Holocene (LPIH) $CH_4$ amplitude is elevated to early Holocene level from ~0 C.E. (~2.0 ka), and increases up to higher from ~1450 C.E. (~0.5 ka).*

*The reason of low amplitude variability during 3.5 to 1.2 ka, or why the early Holocene $CH_4$ variability is larger than this period, is probably related to different orbital configuration in both time periods. Previous studies found covariation between $CH_4$ amplitude and NH summer insolation change, reflecting that mean temperature of the warmest seasons is an important factor of $CH_4$ emission, during the interstadial conditions (Flückiger et al., 2004; Baumgartner et al., 2014). Combined with elevated summer insolation in Northern Hemisphere (NH) and with climate warming in NH extratropics, the amplified variability of the early Holocene may suggest that $CH_4$ control by NH wetlands was likely stronger than the late Holocene period. Meanwhile, lower summer insolation during the late Holocene might induce diminished $CH_4$ amplitude. This evidence indicates the natural forcing in centennial- to millennial time scales is reduced in the late Holocene, given that the atmospheric $CH_4$ budget during 3.5-1.2 ka (604.9 ppb) is similar to that during 9.0-7.6 ka (628.6 ppb), and that anthropogenic emission is greater in later Holocene than the early Holocene. Abrupt increase of $CH_4$ amplitude since ~800 C.E. (1.2 ka) is likely driven by increasing anthropogenic contribution, which is consistent with anthropogenic emission scenario based on past population and agricultural activity (Mitchell et al., 2013). Also superimposed are short-term cooling events during Little Ice Age, making $CH_4$ variability greater."*

[Figure]

*Figure R5. Upper: Detrended (75 to 1800-year band-pass filtered) CH₄ for the early (a) and late (b)*
*Holocene from Siple Dome (red, this study), WAIS divide continuous (purple, Rhodes et al., 2015), and*
*WAIS divide discrete (blue, Mitchell et al., 2013) data. Dashed lines are root mean square (RMS)*
*amplitude running averaged by 75-year window. Lower: Amplitude spectrum of Early (c) and Late (d)*
*Holocene CH₄ records. Note that CH₄ data before 1750 C.E. are used for the preindustrial late*
*Holocene.*

b)        Millennial CH4 variations: The authors suggest that climate cooling in the northern
hemisphere has led to a southward shift in the ITCZ, which again led to a decline in CH4 low latitude
emissions due to changes in monsoon systems. The first part of this hypothesis (ITCZ shift) appears to
be straightforward and has been observed in models, however, the second part (CH4 emission changes)
appears not so straightforward and requires some more quantitative support. Rhodes et al. (2015)
suggest a first order relationship between CH4 emissions and intense rainfall area, where from a
certain point on also an increase in southern hemisphere wetland emissions is possible. Accordingly,
a discussion focusing on Asian monsoon systems only, as in the manuscript by Yang, seems to be too
narrow. Please explain how your hypothesis fits into this picture. Please note also the work by Bozbiyik
et al., CP, 2011, performing a North Atlantic fresh water hosing experiment under interglacial
conditions connected to a southward shift of the ITCZ, showing decreases in tropical precipitation and
the modeling work by Zurcher et al., Biogeoscience, 2013, which shows that also boreal peatland CH4
emissions are reduced during such an experiment. Finally, the discussion of the millennial CH4
variability and the corroborating proxy evidence from other archives lacks some clarity and could be
improved.

*We thank again for useful comment and paper suggestion. One of main point of Rhodes et al. (2015) is that abrupt $CH_4$ increase occurred during Heinrich Stadial 1, 2, 4, and 5 events could be induced by the southern hemisphere emission as a result of strong southward migration of ITCZ at that time. However, mean latitudinal position of ITCZ moved northward during the Holocene climate conditions, therefore monsoon intensity in northern tropics should be strengthened while tropical rainfall in southern hemisphere decreased. This is identified from Cariaco Basin reflectance record which shows increase in rainfall during the Holocene compared to glacial period (e.g., Deplazes et al., 2013), as well as from anti-correlation between Chinese- and Brazilian cave speleothem record (e.g., Wang et al., 2006). Further, Asian monsoon intensity during the Heinrich Stadials was weaker than during the early Holocene (e.g., Wang et al., 2008). For these reasons, we focused more on Asian/Indian monsoon variability than others, but we agree that our manuscript lacks detailed explanation on other monsoon systems.*

*The Figure R4 (see above our response to Reviewer #1) now shows Asian, Indian, African and Brazilian monsoon proxies, with age uncertainties indicated as black horizontal error bars. Gulf of Guinea (western tropical Africa) planktonic Ba/Ca ratio, a proxy of riverine runoff, shows decreased rainfalls at similar timings of local $CH_4$ minima at 8.2, 9.3 and 10.9 kyr BP, indicating that the abrupt Greenland cooling leads to hemispheric-wide hydroclimatic changes. Inverse pattern of South American rainfall (Lapa Grande Cave, eastern Brazil) supports that ITCZ was temporarily migrated southward at that time. High-resolution sediment reflectance records from Cariaco Basin and Arabian Sea clearly show that the strength of southward migration of ITCZ and its effect of precipitation change are smaller during the early Holocene than during the Heinrich Stadials (Deplazes et al., 2013; 2014).*

*We thank the reviewer for suggesting appropriate paper. Zurcher et al. (2013) found that abrupt cooling in Greenland and northern high latitudes by large freshwater input causes boreal peatland $CH_4$ emission to decrease substantially, which explains ~23% of abrupt $CH_4$ drop (~80 ppb) during the 8.2k event. If we assume linear scaling of model response, it implies that boreal peatland source change only accounts for ~23% of total $CH_4$ change during the rest of $CH_4$ decrease events. Given the meltwater pulses during the early Holocene before the 8.2 k event are more than 10 times smaller (Teller and Leverington, 2004) than that corresponds to the 8.2 k event, we consider the boreal emission change is not the major cause of $CH_4$ local minima.*

c)      Interpolar CH4 difference: The IPD is a tricky business and erroneous effects can be easily introduced by comparing CH4 data from different labs, different sites, or insufficient robustness of the results. Accordingly, this needs more supporting information and detail: In the method part it is mentioned that blank ice measurements show an offset with a very large scatter between 5 and 15 ppb. I read the text in such a way that a daily blank correction is applied based on 4 blank measurements per day. This needs more detailed discussion as a potentially erroneous correction, which varies by 10 ppb, has a huge influence on the IPD, which varies with a similar amplitude. Please add the following information/discussion:

- Can you be sure that the CH4 blank is coming from the extraction system and is not reflecting dissolved CH4 in the bubble-free ice? In the latter case you should not correct for this blank. Did you perform blank tests without bubble-free ice or did you repeat the extraction of bubble-free ice for a second or third time to see, whether the blank is constant or declining?

*We've performed the gas extraction test using bubble-free ices to estimate how much air dissolves in ice melt. We found less than 5 mTorr of gas extracted after second melting-refreeze procedure. Considering the typical amount of standard air injected (30-40 Torr), the air extracted from the second melting-refreezing process should have $CH_4$ mixing ratio of >~100000 ppb to cause 20 ppb of*

*blank offset. This is unlikely because (1) such high concentration cannot be explained by gas solubility effect, and (2) the bubble-free ices were trimmed outermost layer sufficiently before cutting the artificial ice samples to prevent causal contamination by ambient air dissolution. The microbial activity within bubble-free ice during storage is unrealistic, given that we use water distiller (Barnstead) and anti-bacterial membrane filter (Millipak Express) to produce deionized water, and the deionized water is boiled within a stainless steel chamber for degassing. Further, we found the blank offset is similar order from the blank test without bubble-free ices (9.4 ~ 21.6 ppb, n = 36) using various working standards (721.31, 895.03, and 1384.91 ppb $CH_4$ in NOAA04 scale). Therefore, we conclude that the $CH_4$ blank offset caused by dissolved $CH_4$ in bubble-free ice is not the case.*

-        Please add information on the time scale on which the blank changes. If my understanding is correct that a daily mean blank correction of 5-10 ppb is performed, it is important to know what the variability of the four blank ice measurements is within one day. If this intra-day variability is of the same size as the inter-day variability, then a daily blank correction varying between 5 and 15 ppb introduces offsets from one day to the other which only reflect the stochastic variability of the blank measurements themselves and not systematic day-to-day differences in the entire measurement system. In that case a long-term mean blank correction seems more appropriate. If the blank values are reproducible within one day, a daily correction seems justified.

*The daily systematic offset was determined using a standard air injected into the sample flasks which have bubble-free blank ices. Even though the systematic offset varied daily, the 4 blank results were rather in a small range, yielding the intra-day standard error of the mean of the 4 blanks of 2.0 ± 1.0 ppb on average. Our final $CH_4$ data were presented by averaging the results of duplicate sample analysis from the same depths. To test our correction method, we reanalyzed duplicate samples from the adjacent ices (~10 cm depth difference) at 8 depth intervals of 8 - 80 days after the first analysis (see the table in the response to reviewer #1). The pooled standard deviation of the average of duplicates from first and second measurements was 1.4 ppb. This implies a good reproducibility and good precision of our analytical method.*

*Despite it is true that daily blank offsets are larger than the stated precision, the good agreement between the original- and replicate measurements does reveal that our blank correction method is reliable.*

-        On the other hand if you have a long-term trend in this blank value, which may not reflect a trend in the extraction system but in the bubble-free ice quality, you introduce this error into the IPD. Did you use a randomized order to measure the samples to avoid such spurious trends.

*Siple Dome ice samples were measured in a randomized order.  As we described above, contamination by blank ice quality and/or by air occluded in blank ice is unlikely. Therefore, we decide not to apply this correction.*

-        There was an average offset of 3 ppb observed between Siple Dome data measured at SNU and OSU and the OSU data have been corrected by subtracting 3 ppb, but it is not discussed what the influence of this correction may be on the IDP. Note that the NEEM discrete data used to calculate the IPD are also measured at OSU. Accordingly, to avoid any systematic errors in the IPD it is mandatory to add the 3 ppb to the Siple Dome data measured at SNU and not to subtract 3 ppb from OSU data.

*The offset comes from different correction methods between the two laboratories. As described in our response to the Reviewer #1, instead of simply adding 3 ppb to SNU data, we applied the similar solubility correction used in Mitchell et al. (2011) at OSU, and found that the average offset between SNU-OSU reduces to ~0.6 ppb, which lies within analytical uncertainty range of both institutes.*

*Instead of adding 3 ppb, now we apply the OSU solubility correction methods to our data for IPD calculation.*

-        Comparing the continuous WAIS data with the discrete (or continuous) CH4 data from NEEM, it becomes apparent that the relative changes in CH4 concentrations in the northern and southern hemisphere are much more similar than when comparing Siple Dome and NEEM data after the Monte Carlo synchronization in Figure 3. For example the downward trend in NEEM between 11.3 and 10.9 kyr BP is also seen in the continuous WAIS data, while in Siple this time interval shows essentially constant CH4 after a first short peak. Consequently, the constant values in the Siple data lead to an erroneous downward trend in the IPD in this time interval. Vice versa, there is an upward trend from 10.9 to 10.4 kyr BP found in NEEM and continuous WAIS data. The same time interval in Siple looks more like a broad maximum, again with implications on the derived IPD. A similar observation holds for the maximum around 10 kyr BP. Note that on these centennial to millennial time scales, which are much longer than the atmospheric lifetime and the interhemispheric exchange time, you may have changes in the size of the IPD, however, it is extremely difficult to create a millennial trend in one hemisphere without a trend in the other. This is nicely illustrated in the high-resolution data by Mitchell et al., 2013. Obviously, the resolution and quality of the data in Figure 3 does not suffice to gain a robust IPD and/or the Monte Carlo synchronization fails to synchronize the records sufficiently. In fact, it seems crucial that the IPD analysis is performed not only on the Siple but also on the WAIS discrete and continuous data to study the robustness of the results gained from the Siple Dome core. Note that the WAIS very high-resolution data from continuous measurements can be used to much better synchronize WAIS to the continuous records available from NEEM. This would circumvent the synchronization problems apparent between Siple and NEEM. As a final remark on this topic, I do not agree with the authors' statement that the IPD values in Siple Dome over the time interval 9.5-11.5 kyr BP are in agreement with previous results. If you calculate the mean over this time interval in the Siple IPD data and calculate the standard error, this appears to clearly higher than the literature values. In summary, the IPD discussion needs more work before the manuscript should be published in CP.

*We thank to the Reviewer #2 for pinpoint comment and useful suggestions. We revised the IPD section of our manuscript based on the following discussions:*

*To test robustness of the early Holocene IPD change, we revised the previous IPD and calculated various IPD curves by using different data sets, including NEEM continuous, NEEM discrete, Siple Dome discrete, WAIS continuous, and WAIS discrete data (Figure R3 above). We calibrated NEEM and WAIS continuous data against to discrete measurements, as discrete measurements are more accurate than continuous ones in absolute values although they are worse in precision. However, we hesitate to equally consider a*

*ll the IPDs because some CH$_4$ data sets are not sufficient for centennial to millennial IPD estimates (see below for detailed discussion on data reliability). Here we consider NEEM discrete, WAIS Divide continuous, and Siple Dome discrete records are reliable to draw IPDs. Also we address the inconsistency among the different ice core records for interval older than ~10.3 kyr BP, which make IPDs different each other.*

*Resulting IPD curve from NEEM discrete and WAIS continuous (green) shows long-term increase from the onset of Holocene to ~9.9 kyr BP, and it supports the NEEM-Siple IPD reconstruction (black). The good agreement implies that the millennial scale IPD increase trend during the early Holocene is robust. However, the IPD fluctuation during 10.8 – 11.2 kyr BP is not reproduced in the alternative IPD, hence we modified our original argument that IPD increase from ~10.7 to 9.8 kyr BP. Instead, we focused on long term IPD increase. Except for during 10.8 – 11.2 kyr BP, both IPDs show concomitant*

*increase with NH extratropical temperature and thermokarst lake $CH_4$ emission increase. These*

*evidences show that boreal emission increased while tropical emission decreased (Table R1).*

*Finally, we'd like to note that mean value of the Siple Dome IPD is 41 ± 6 ppb over the 9.5-11.5 kyr BP*

*period, which is consistent with previous results within uncertainty range.*

*We excluded some dataset from our discussion for the following reasons. Rhodes et al. (2015)*

*reported that WAIS continuous data are lower than OSU discrete measurements by 1.5-2.5% for*

*1804-2420m (9.8 – 17.1 kyr BP in WD2014 scale) interval. WAIS continuous data were calibrated*

*against to Siple discrete data instead of WAIS data, because we consider that Siple data are more*

*reliable during the early Holocene period. Even though the analytical method of Mitchell et al. (2011)*

*has been regarded as a "benchmark" of discrete wet-extraction technique, but unfortunately, none of*

*existing Antarctic $CH_4$ data for the early Holocene was measured by Mitchell et al. (2011) method.*

*Most of WAIS discrete data covering the early Holocene were measured in a different institute (Penn*

*State University, PSU) showing a noisy trend with pooled standard deviation of ~7.3 ppb (1σ), and*

*there is an unexplained offset of 9.9 ppb between WAIS discrete data measured in OSU and PSU lab*

*(Rhodes et al., 2015). As it lacks rigorous comparison between the two data sets during the early*

*Holocene, there is no evidence to show that WAIS discrete data are more reliable. Meanwhile, it has*

*been revealed that during the early Holocene interval, SNU Siple data (this study) agree well with*

*OSU Siple data (Ahn et al., 2014) by comparison of the nearest data point from both labs.*

*Furthermore, it should be noted that we used NEEM continuous data obtained by WS-CRDS*

*(Wavelength Scanned Cavity Ring Down Spectroscopy, CFADS36, Picarro Inc.)) because the OF-CEAS*

*(Optical Feedback Cavity Enhanced Absorption Spectroscopy, SARA, Laboratoire Interdisciplinaire de*

*Physique, Universite Joseph Fourier, Grenoble, France) instrument was calibrated against different*

*standard scale (CSIRO standard, Chappellaz et al., 2013), and WAIS continuous data were measured*

*by the same instrument (WS-CRDS, 1804-2621m depth, 2012 campaign, Rhodes et al., 2015). In*

*summary, we regard the NEEM discrete, Siple Dome discrete, and WAIS Divide continuous records as*

*more reliable than the others during the early Holocene period.*

Specific comments

I started to correct for English language issues, but had to stop at some point. Please ask your English speaking co-author for a thorough language check not only for typos but also to improve the clarity of the arguments. As major textual changes are still required for this manuscript, I will not comment on language issues here.

P(age) 2 l(ine) 2: Daniau et al., 2012 is not an appropriate reference in this respect (CH4 emissions)

*Citation has been changed.*

P2 l20: Cite recent work by Baumgartner et al. CP 2014

*Citation added.*

P2: discuss in more detail previous work on the relationship between ITCZ changes and CH4

emissions

*Following paragraphs have been added in the introduction section:*

*P37 L9-20: "Relationship between the latitudinal shift of ITCZ and $CH_4$ emission varies with time*

*scales. Landais et al. (2010) and Guo et al. (2012) suggested that ITCZ migration is not a dominant*

*control of glacial-interglacial $CH_4$ cycle because long-term $CH_4$ trend does not follow well the*

*precessional insolation change in the northern hemisphere. Modelling studies found the southward*

*shift of ITCZ coincides with reduced $CH_4$ in Last Glacial Maximum (LGM) and Heinrich Stadial (HS)*

*events, but changes in wetland area and surface hydrology were small (Weber et al., 2010; Hopcroft*

*et al., 2011). They instead suggested that changes in temperature and/or plant productivity affected*

*$CH_4$ production during those events. Rather, ITCZ migration appears to be related with millennial- or*

*sub-millennial scale $CH_4$ change. Brook et al. (2000) found that submillennial-scale $CH_4$ minima*

*during the last deglacial period correspond with reduced precipitation recorded in Cariaco Basin*

*sediment data, which indicates southward displacement of ITCZ (Hughen et al., 1996). It is supported*

*by spectral analysis of $CH_4$ during the past 800 ka record that found that ITCZ change becomes an*

*important driver of millennial scale $CH_4$ change (Tzedakis et al., 2009; Guo et al., 2012)."*

P3: discuss the difference in orbital parameters for the early and late Holocene and the potential implications for CH4 emissions

*Please refer to our response to General comment a).*

P4 methods: Is it correct that you use only a one standard calibration? Comment on the potential systematic error introduced by this approach

*The GC linearity was tested by using working standards of 395.5, 721.3, 895.0, and 1384.9 ppb (in*

*NOAA04 scale). This has been added in method section.*

P4 l25-32: This paragraph should be moved to the methods section

*The paragraph was moved to Method section.*

P5 1st paragraph. You say that you use a 250 year running average (and similar a high-pass filter with

1800 cut-off), however, your data is not equidistant. Please explain in more detail how you averaged the data

*We interpolated the data annually and then averaged each 250-year interval. This has been added.*

P5 l14: Is the significance level of the correlation coefficient really taking the reduced degrees of freedom into account after averaging the data? Looking at the value, I am afraid it didn't and the significance is highly overestimated.

*We add more details on it. Before the synchronization to GICC05, the correlation coefficient between*

*the filtered SDMA $CH_4$ and NGRIP $\delta^{18}O_{ice}$ was calculated as r = 0.57 (p = 0.06). After adjusting to*

*GICC05 scale by matching with NGRIP $\delta^{18}O_{ice}$ at 8.2 ka event and PBO interval (three tie-points), this*

*correlation increases to r = 0.74 (p < 0.01).*

P5: see comment on insufficient discussion of the effect of an ITCZ shift on CH4 emissions north and south of the equator. Please discuss also in more detail the dating uncertainties of the various archives and their potential impacts on the conclusions.

*We added age uncertainty (1σ) of each proxy used.*

P5 l30: Reference Bjorck et al. is not in the reference list

*The reference was added.*

*Björck, S., Muscheler, R., Kromer, B., Andresen, C. S., Heinemeier, J., Johnsen, S. J., Conley, D., Koç, N., Spurk, M., and Veski, S.: High-resolution analyses of an early Holocene climate event may imply decreased solar forcing as an important climate trigger, Geology, 29, 1107-1110, 2001.*

P6 l5-7: There is also variability in GRIP and GISP2. Please explain in more detail what you refer to.

*Unlikely to NGRIP data, GRIP and GISP2 records show even smaller variability at 10.9 ka than that of 10.3 ka.*

P7 l1-17: This paragraph is highly speculative and lacks clarity and detail.

*In this paragraph we intended to discuss possibility of solar forcing to observed $CH_4$ change. Several previous studies found evidences solar-induced climate change, but we observed that the timings of solar activity minima differ by 195 (8.2 ka), 278 (9.3 ka), 110 (10.3 ka), and 250 (11.0 ka) years to $CH_4$ minima and Greenland cooling. The maximum layer counting error of GRIP age scale (GICC05) is less than 100 years (Rasmussen et al., 2006), and the maximum gas age uncertainty of Siple Dome is ~150 years (This study). Therefore, age difference larger than ~180 years is not explained by chronological uncertainty.*

P7 following l24: You disturbed the age of the data points by a Gaussian distribution with sigma=30 years. How did you make sure that the chronological order of all data points was ensured in your approach? How did you take the measurement uncertainty in each data point into account? Please explain in more detail.

*We chose the sigma of 30 years given the mean temporal resolution of Siple data is ~27 years.*

P8 l1: there is a significant offset between your average and previous IPD estimates

*Figure R2 (above) shows the IPD calculated from NEEM discrete and SDMA discrete data together with alternative IPDs from different data set. Previous IPD estimates lie within range of the new IPDs. The IPDs calculated from NEEM continuous data show higher values, which reflects the offset between NEEM continuous and discrete record.*

P8 l10-12: not entirely clear to the outsider what you did, please clarify

*We modified the paragraph as below:*

*P47 L33-38: "~~To calculate the N-box $CH_4$, we subtracted the 7 % of IPD from Greenland $CH_4$ concentration, assuming the difference between Greenland and the mean latitude of N-box is ~7 % of IPD (Chappellaz et al., 1997).~~ The mean $CH_4$ mole fraction of N-box (30-90N) is not identical to that of Greenland ice core record, given the latitudinal $CH_4$ distribution (e.g., Fung et al., 1991). To derive the N-box $CH_4$, we followed the assumption of Chappellaz et al. (1997), where the authors assumed that difference between Greenland and the mean N-box $CH_4$ is 7% of IPD. Hence here the N-box CH4 is calculated by subtracting 7% of IPD from the Greenland mixing ratio."*

P8 l19: the boreal sources increased

*We removed the paragraph.*

P9 l10. You discuss the effect of the different age distributions in the Siple Dome and NEEM cores, but you do not follow up on this in your analysis. Either you use WAIS to compare with NEEM (as it has essentially the same enclosure characteristics) or you low-pass filter NEEM to the same enclosure characteristics as Siple. I would strongly recommend to do both to study the robustness of the results.

*See our response to general comment above. We present alternative IPD reconstructions including WAIS and NEEM data.*

P9 l20-21: The results by Fischer et al. (2008) on LGM biomass burning emissions result from the use of temporally constant isotopic source signatures in the box model approach. Moller et al., Nature Geoscience, 2013 showed that also the source signatures changed significantly over time and they revised the biomass burning estimates, showing that LGM emissions were lower than Holocene emissions.

*We removed the entire paragraph on LGM-Holocene transition, because it is out of main focus of this paper.*

P9 l26: why do you only refer to biomass burning in the tropics?

*We removed the entire paragraph on LGM-Holocene transition, because it is out of main focus of this paper.*

P20 l5 Chappellaz et al., 1997 not 2013
*We removed the comparison.*

**References that are not cited in discussion paper**---------------------------------------------------------

Andreae, M. O., and Merlet, P.: Emission of trace gases and aerosols from biomass burning, Global

Biogeochem. Cycles, 15, 955-966, 2001.

Baumgartner, M., Kindler, P., Eicher, O., Floch, G., Schilt, A., Schwander, J., Spahni, R., Capron, E.,

Chappellaz, J., Leuenberger, M., Fischer, H., and Stocker, T. F., NGRIP $CH_4$ concentration from 120 to

10 kyr before present and its relation to an $\delta^{15}N$ temperature reconstruction from the same ice core,

Clim. Past, 10, 2014.

Björck, S., Muscheler, R., Kromer, B., Andresen, C. S., Heinemeier, J., Johnsen, S. J., Conley, D., Koç, N.,

Spurk, M., and Veski, S.: High-resolution analyses of an early Holocene climate event may imply decreased solar forcing as an important climate trigger, Geology, 29, 1107-1110, 2001.

Brook, E. J., Harder, S., Severinghaus, J., Steig, E. J., and Sucher, C. M.: On the origin and timing of rapid changes in atmospheric methane during the last glacial period, Global Biogeochem. Cycles, 14,

559-572, 2000.

Buiron, D., Chappellaz, J., Stenni, B., Frezzotti, M., Baumgartner, M., Capron, E., Landais, A., Lemieux-

Dudon, B., Masson-Delmotte, V., Montagnat, M., Parrenin, F., and Schilt, A.: TALDICE-1 age scale of the Talos Dome deep ice core, East Antarctica, Clim. Past, 7, 1-16, 2011.

Cheng, H., Sinha, A., Cruz, F. W., Wang, X., Edwards, R. L., d'Horta, F. M., Ribas, C. C., Vuille, M., Stott,

L. D., and Auler, A. S.: Climate change patterns in Amazonia and biodiversity, Nat. Commun. 4, 1411,

2013.

Craig, H., Horibe, Y., and Sowers, T.: Gravitational separation of gases and isotopes in polar ice caps,

Science, 242, 1675-1678, 1988.

Cruz, F. W., Burns, S. J., Karmann, I., Sharp, W. D., Vuille, M., Cardoso, A. O., Ferrari, J. A., Silva Dias, P.

L., and Vianna, O.: Insolation-driven changes in atmospheric circulation over the past 116,000 years in subtropical Brazil, Nature, 434, 63-66, 2005.

Dai, A., and Wigley, T. M. L.: Global patterns of ENSO-induced precipitation, Geophys. Res. Lett., 27,

1283-1286, 2000.

Ferretti, D. F., Miller, J. B., White, J. W. C., Etheridge, D. M., Lassey, K. R., Lowe, D. C., MacFarling

Meure, C. M., Dreier, M. F., Trudinger, C. M., van Ommen, T. D., and Langenfelds, R. L.: Unexpected changes to the global methane budget over the past 2000 years, Science, 309, 1714-1717, 2005.

Flückiger, J., Blunier, T., Stauffer, B., Chappellaz, J., Spahni R., Kawamura, K., Schwander, J., Stocker, T.

F., and Dahl-Jensen, D., $N_2O$ and $CH_4$ variations during the last glacial epoch: Insight into global processes, Global Biogeochem. Cycles, 18, GB1020, 2004.

Griffiths, M. L., Drysdale, R. N., Gagan, M. K., Zhao, J. –X., Ayliffe, L. K., Hellstrom, J. C., Hantoro, W. S.,

Frisia, S., Feng, Y. –X., Cartwright, I., Pierre, E. St., Fischer, M. J., and Suwargadi, B. W.: Increasing

Australian-Indonesian monsoon rainfall linked to early Holocene sea-level rise, Nature Geosci., 2,

636-639, 2009.

Guo, Z., Zhou, X., and Wu, H.: Glacial-interglacial water cycle, global monsoon and atmospheric methane changes, Glim. Dyn., 39, 1073-1092, 2012.

Hao, W. M., and Ward, D. E.: Methane production from global biomass burning, J. Geophys. Res., 98,
20657-20661, 1993.

Hopcroft, P. O., Valdes, P. J., and Beerling, D. J.: Simulationg idealized Dansgaard-Oeschger events and
their potential impacts on the global methane cycle, Quat. Sci. Rev., 30, 3258-3268, 2011.

Huber, C., Leuenberger, M., Spahni, R., Fluckiger, J., Schwander, J., Stocker, T. F., Johnsen, S., Landais,
A., and Jouzel, J.: Isotope calibrated Greenland temperature record over Marine Isotope Stage 3 and
its relationship to $CH_4$, Earth Planet. Sci. Lett., 243, 504-519, 2006.

Hughen, K. A., Overpeck, J. T., Peterson, L. C., and Anderson, R. F.: The nature of varved
sedimentation in the Cariaco Basin, Venezuela, and its palaeoclimatic significance, Geol. Soc. London
Spec. Publ., 116, 171-183, 1996.

Kirby, M. E., Lund, S. P., Patterson, P., Anderson, M. A., Bird, B. W., Ivanovici, L., Monarrez, P., and
Nielsen, S.: A Holocene record of Pacific Decadal Oscillation (PDO) – related hydrologic variability in
Southern California Lake Elsinore, CA, J. Paleolim., 44, 819-839, 2010.

Landais, A., Dreyfus, G., Capron, E., Masson-Delmotte, V., Sanchez-Goñi, M. F., Desprat, S., Hoffman,
G., Jouzel, J., Leuenberger, M., and Johnsen, S.: What drives the millennial and orbital variations of
$\delta^{18}O_{atm}$?, Quat. Sci. Rev., 29, 235-246, 2010.

Levy II, H.: Normal atmosphere: large radical and formaldehyde concentrations predicted, Science,
173, 141-143, 1971.

Loulergue, L., Schilt, A., Spahni, R., Masson-Delmotte, V., Blunier, T., Lemieux, B., Barnola, J.-M.,
Raynaud, D., Stocker, T. F., and Chappellaz, J.: Orbital and millennial-scale features of atmospheric
$CH_4$ over the past 800,000 years, Nature, 453, 383-386, 2008.

Lyon, B., and Barnston, A. G.: ENSO and the spatial extent of interannual precipitation extremes in
tropical land areas, J. Clim., 18, 5095-5108, 2005.

Moy, C. M., Seltzer, G. O., Rodbell, D. T.: Variability of El Niño/Southern Oscillation activity at
millennial timescales during the Holocene epoch, Nature, 420, 162-165, 2002.

Partin, J. W., Cobb, K. M., Adkins, J. F., Clark, B., and Fernandez, D. P.: Millennial-scale trends in west
Pacific warm pool hydrology since the Last Glacial Maximum, Nature, 449, 452-456, 2007.

Renssen, H., Goosse, H., and Muscheler, R.: Coupled climate model simulation of Holocene cooling events:
oceanic feedback amplifies solar forcing, Clim. Past, 2, 79-90, 2006.
Rodbell, D. T., Seltzer, G. O., Anderson, D. M., Abbott, M. B., Enfield, D. B., and Newman, J. H.: An
~15,000-year record of El Niño-driven alleviation in southwestern Ecuador, Science, 22, 516-520,
1999.
Schulz, M., and Mudelsee, M.: REDFIT: estimating red-noise spectra directly from unevenly spaced
paleoclimatic time series, Computers & Geosciences, 28, 421-426, 2002.

Sperlich, P., Schaefer, H., Mikaloff Fletcher, S. E., Guillevic, M., Lassey, K., Sapart, C. J., Röckmann, T.,
and Blunier, T.: Carbon isotope ratios suggest no additional methane from boreal wetlands during
the rapid Greenland Interstadial 21.2, Global Biogeochem. Cycles, 29, 1962-1976, 2015.

Strikis, N. M., Cruz, F. W., Cheng, H., Karmann, I., Edwards, R. L., Vuille, M., Wang, X., Paula, M. S.,

Novello, V. F., and Auler, A. S., Abrupt variations in South American monsoon rainfall during the

Holocene based on a speleothem record from central-eastern Brazil, Geology, 39, 2011.

Taylor, K. C., Alley, R. B., Meese, D. A., Spencer, M. K., Brook, E. J., Dunbar, N. W., Finkel, R. C., Gow, A.

J., Kurbatov, A. V., Lamorey, G. W., Mayewski, P. A., Meyerson, E. A, Nishiizumi, K., Zielinski, G. A.:

Dating the Siple Dome (Antarctica) ice core by manual and computer interpretation of annual layering, J. Glaciol., 50, 453-461, 2004.

Tzedakis, P. C., Palike, H., Roucoux, K. H., and de Abreu, L.: Atmospheric methane, southern European vegetation and low-mid latitude links on orbital and millennial timescales, Earth Planet. Sci. Lett.,

277, 307-317, 2009.

Wang, X., Auler, A. S., Edwards, R. L., Cheng, H., Ito, E., and Solheid, M., Interhemispheric anti- phasing of rainfall during the last glacial period, Quat. Sci. Rev., 25, 3391-3403, 2006.

Wang, Y., Cheng, H., Lawrence Edwards, R., Kong, X., Shao, X., Chen, S., Wu, J., Jiang, X., Wang, X., and An, Z.: Millennial- and orbital-scale changes in the East Asian monsoon over the past 224,000

years, Science, 451, 1090-1093, 2008.

Weldeab, S., Lea, D. W., Schneider, R. R., and Andersen, N., Centennial scale climate instabilities in a wet early Holocene West African monsoon, Geophys. Res. Lett., 34, L24702, 2007.

[revised manuscript text omitted]

Our new $CH_4$ data confirms the abrupt doubling at the Younger Dryas termination. Previous studies using stable isotopes of $CH_4$ have shown contradictory results. Previous studies that, using the stable isotopic composition of C and H in $CH_4$ that aimed to disentangle the cause of abrupt $CH_4$ increase during the earliest period of the Holocene have shown contradictory results. Schaefer et al. (2006) calculated isotopic ($\delta^{13}C$-$CH_4$) mass balance model to discern major source term that caused a slight enrichment in $^{13}C$ during the Younger Dryas termination, suggesting tropical wetland emission as a dominant source. The authors also proposed biomass burning, geologic $CH_4$ and enhanced sink process at marine boundary layer as alternatives, but less probable scenarios. On the other hand, Fischer et al. (2008) argued nearly constant biomass burning emission of ~45 Tg yr$^{-1}$ throughout the last glacial termination with a slight increase in PB, and also showed that the boreal sources were expanded during the YD PB transition. However, Möller et al. (2013) pointed out the possibility of changing isotopic signature of each source itself, and they found that less pyrogenic emission is required for LGM condition if they increased the $\delta^{13}C$-$CH_4$ signatures of tropical wetland and of biomass burning. 
[revised manuscript text omitted]

**Supplementary text**

**Reproducibility test from Styx glacier ice core**

 Here we present results of reproducibility test by using different ice core samples in the same manner as used for Siple Dome ices, to demonstrate the reliability of our analytical system. The ice core was drilled at Styx glacier (73° 51.10'S, 163° 41.22'E, 1623 m a.s.l) in 2014-2015 austral summer, and mean snow accumulation rate was estimated as 0.13 Mg m$^{-2}$ yr$^{-1}$ (Han et al., 2015). The replicate measurements were carried out at randomly-chosen 7 depths with time interval of 51 to 226 days. Depth difference between the replicate pairs is less than 10 cm. Results show the mean absolute difference between the original- and replicate measurements of 1.9, which is as good as the Siple Dome results. The daily blank offset (see the main text) ranges from ~6 to 19 ppb, and the intra-day blank offset is 2.4 ppb (standard error of the mean). Again, these results reveal the robustness of our blank correction methods.

**Maximum SDMA gas age unceratinty**

 In this paper we used a modified gas age scale from the previous one based on $\delta^{15}$N measurements (Severinghaus et al., 2009) by setting 3 age tie-points and interpolating the age offset between the tie-points (Fig. S1). To estimate gas age uncertainty, we compared the SDMA modified gas age (this study) to the new gas age determined by CH$_4$ correlation with NEEM discrete CH$_4$ data measured at OSU (Chappellaz et al., 2013). Figure S3 shows the offset between the two age scales, which it should be moved to adjust the NEEM CH$_4$ in GICC05modelext-NEEM-1 age. In addition, we take into account the maximum layer counting uncertainty of 99 years (Rasmussen et al., 2006) and delta-age uncertainty of 30 years (Rasmussen et al., 2013) during the early Holocene. Error progagation gives us the maximum uncertainty of the early Holocene SDMA gas age of ~147.4 years.

[Figure]

**Figure S1. Upper: Comparison between Siple Dome CH₄ anomalies plotted with gas age adjusted to GICC05 (red, solid)**

**and previous gas age (red, dashed; Brook et al., 2005). Lower: NGRIP δ¹⁸O anomaly in GICC05 scale. The horizontal**

**error bars denote the age uncertainty of GICC05 chronology (Rasmussen et al., 2006), and the black triangles are age**

**tie points used to adjusting the Siple Dome age scale to GICC05 scale.**

[Figure]

**Figure S2. Comparison of Greenland oxygen isotope ratios from NGRIP (blue, Rasmussen et al., 2006), GRIP (grey,**

**Rasmussen et al., 2006) and GISP2 (red, Stuiver and Grootes, 2000). All time series were high-pass filtered with 1/1800-**

**year window. Note that the cooling amplitude at 10.3 ka is smaller than 8.2 and 9.3 ka events in NGRIP records, but**

**this is not clear in GRIP and GISP2 ice cores.**

[Figure]

[Figure]

**Figure S3.** 3-box source distribution model results from IPD-2.

[Figure]

**Figure S4. Age difference between the new gas age scale adjusted to GICC05 by Monte Carlo matching with NEEM**

**discrete CH$_4$ (Chappellaz et al., 2013) and the original gas age based on $\delta^{15}$N records (Severinghaus et al., 2009).**

**Table S1. 3-box source distribution model results of tropical (green, T) and boreal (red, N) boxes and boreal source fraction obtained from IPD-2. Errors denote 95% confidence interval.**

| Ref. | N box | T box | Boreal source fraction |
|---|---|---|---|
| (ka) | (Tg yr$^{-1}$) | | (%) |
| This study (11.5 ka) | 60 ± 7 | 134 ± 16 | 29 ± 4 |
| This study (10.0 ka) | 71 ± 7 | 115 ± 11 | 35 ± 4 |

---

## Referee Report (RR1)

Thanks a lot to the authors for their considerable revisions of the original manuscript. In my view, the new version is a great improvement. The structure gained clarity and the refined focus on the time scale makes the entire paper appear more concise. Again, this manuscript might require some language editing or input by the native speaker of the author list. On top of the good development, I recommend major revisions on several aspects of the new manuscript. These include i) the description of the analytical system and its uncertainties, ii) the use of IPD-1 and IPD-2, iii) Section 3.3, Comparison with late Holocene variability, iv) conclusions, v) Supplements.

**i) the description of the analytical system and its uncertainties**

I find the part on the analytical uncertainty of the methods section extremely hard to follow and even after careful re-reading, I am not sure what data were used to calculate the stated averages, pooled standard deviations etc. The authors have chosen to publish the data paper before their new method is published in a separate publication. In my point of view, this requires a sufficient description of method and uncertainty if it is claimed that the new method is superior to existing methods, even if a dedicated publication follows. It seems to me that the authors should increase their resistance to the temptation to choose statistical methods that understate the uncertainty of their analytical system, as this leaves the reader rather suspicious.

Please make sure it is well described how you achieve the values you present and what data they were calculated from. For example on page 42, line 6-7 of the document including the tracked changes, you state a standard error of the mean (SEM) of 2.0±1.0 ppb. To my understanding, the SEM is the uncertainty measure, it has no uncertainty itself. What has been calculated here?

In general, SEM's may be used as measure of uncertainty when a sufficient number of samples have been measured. In this manuscript, the authors seem to apply SEM to sample sizes of n=2. If the authors have a good reason to use SEM for n=2, they should clearly state why and account for the obvious lack of sample size (student's t). I feel very strongly that using SEM on n=2 is not OK and needs changing in every uncertainty calculation (e.g. p. 42, L10). Presenting a formula may help clarify what has been calculated. An understandable and ethically correct uncertainty demonstration will provide more credibility to the technical aspects of this manuscript than unrealistically small values. If the system is a new game changer, it should be possible to show outstanding performance using appropriate statistical methods.

This manuscript version has a lot of emphasis on the use of bubble-free blank ice to determine the system blank. The authors state that based on their method to produce the blank ice, they can exclude that CH4 is introduced with the blank ice. From personal experiences, I wouldn't trust that statement. Dedicated blank ice tests including several labs showed that while it is easily possible to produce bubble free ice, there is no such thing as gas-free blank ice. The gas content of the blank ice varied both between the labs and between the batches produced in each lab. All labs used deionised water and pumped on the water for 90 minutes and froze large crystals bottom up… While blank ice allows for a valuable tests, great care has to be taken in the use and interpretation of the results. Based on 4 daily blank ice analyses, the authors state an intra-day SEM of 2.0±1.0 ppb and a variation of the inter-day mean between 5 and 15 ppb. I think the authors intention to present these data is to show that even if the inter-day variation is as large as 15 ppb, the daily blank can be quantified with low uncertainty (SEM 2.0±1.0 ppb) and a correction can therefore be accurately determined. I find this hard to understand. If I chose 4 random values between 5 and 15 (as representative for the large variability of the inter-day blank) and calculate the SEM for these 4

values, the SEM is <2.5. The authors state that the blank estimate of OSU has a high uncertainty of 10% due to the small peak size, but don't seem to think this is a problem when quantifying the blank ice contribution with higher relative uncertainties. Please clarify.

I would think that introducing a known amount of air with a known CH4 mixing ratio into the melt chambers and to analyse that air in the same way as an ice core sample could provide a useful measure of analytical error. This could be done in a dry chamber and over melt water to test for differences. Such experiments could be useful to determine the **accuracy of the measurements**.

The authors use table R1 to show the reproducibility of the analytical system. I find this is a very useful way to state the potential uncertainty/reproducibility of the system and would suggest to improve the clarity and possibly give this more weight in the formulation of the uncertainty estimate. On p. 42 L. 12, the authors refer to the data in table R1 and state, that the differences between the averages from the 1$^{st}$ and the 2$^{nd}$ measurement pair is on average 1.9 ppb. Even though this is nice, it ignores the uncertainty of each single measurement as well as the differences within each sample pair. Again, the authors do not clearly state how the uncertainty in R1 is calculated for 1$^{st}$ and 2$^{nd}$ measurement, is it SEM again for n=2 or 1σ? The authors provide one depth interval for all for samples in table R1, where the depth is sometimes stated to the mm. Are really all four samples from the same mm depth? In my view, this table should include all 4 measurements per depth level. You can then add columns for 1σ or SEM for all four, and calculate the pooled standard deviation (PSD) of the whole lot afterwards as total uncertainty estimate for **measurement precision**. I am not sure if this is what the authors describe on p. 42, L. 13 (PSD of 1.4 ppb)? Did you calculate the PSD based on 4 individual measurements per depth or based on the averages from 1$^{st}$ and 2$^{nd}$ measurements? If this is the "final" uncertainty that the authors intend to state for their analytical system, they should clearly say so in a dedicated sentence, not hidden in a sentence and furthermore in brackets.

P. 42, L 30: You mention the difference OSU-SNU of 3ppb and present a hypothesis on what might have caused this offset. However, this dilemma shows that an analytical uncertainty of 1.4 ppb cannot include the entire uncertainty to the NOAA scale. It would be great to see this resolved, i.e. precision and accuracy, or at least clearly discussed if unresolvable.

On p. 42 L. 34, the authors state that their new Siple Dome data have the third highest temporal resolution. However, they don't say how high it is, as they do for the others. Please quantify, otherwise your statement of 3$^{rd}$ highest is meaningless.

**ii) Use of IPD-1 and IPD-2**

The authors calculate IPD's based on NEEM discrete and Siple discrete (IPD-1) and NEEM discrete and WAIS continuous (IPD-1). Unfortunately, the authors do not describe how the uncertainty envelope is calculated. Furthermore, you do not state the temporal resolution of the Siple data, as compared to the WAIS continuous (p. 42, L. 34). There are a few periods between 11.4 and 10.5 ka where WAIS and Siple are different by ~20 ppb, which produces the different IPD shapes. While WAIS continuous includes several 100 data points (mean resolution of 2 years), I manually count ~15 in the Siple record in Figure 3. Moreover, both the WAIS and NEEM records share several features, that are not resolved at all in the Siple record. It seems to me, that focussing your IPD interpretation on IPD reconstructions that are best constraint makes most sense. I understand the temptation to focus your analysis on the new data. However, I would strongly recommend to shift IPD-2 from Supplements into the main figure and to use IPD-2 for interpretation between 11.5 and 10.0 ka. This

will still leave you with the IPD increase until 9.5 ka, but is better constraint and has less artefacts. In the box model, this would create rather stable emissions from tropics until 10.0 and a decrease thereafter. To me, this would seem the best IPD reconstruction possible.

**iii) Section 3.3, Comparison with late Holocene variability**

This section feels completely unattached to the rest of the manuscript. The figures are described but any quantification is hard to follow. The section title is meaningless and confusing.

**iv) Conclusions**

Please quantify your statements and avoid generalisations, such as high resolution (how high), agree well with previous measurements (how well), four CH4 minima (how big are anomalies, how long in duration), first reporting of IPD increase from 11.5 to 9.9 (how much increase), elevated emissions from NH extra-tropics (how much did emissions from NH extra tropics increase), RMS amplitude smaller (how much) and what does this even mean in conclusions?

**v) Supplements**

I would like to encourage the authors to shift all these important information on analytical reproducibility, gas age uncertainty, IPD-2, age model effect on previous gas age scale and box model data in the main text. Figures can be reduced size, e.g. one column etc, but including those information will increase value of paper! I don't recall being frequently referred to supplements in the main text. Having a lot to read every day, I would probably never get to read supplements that aren't constantly advertised in the main paper, unless this is the most critical paper for my current work.

**General comments**

P36 L11: human influence was substantially smaller

P37 L1: is geological CH4 really the 2$^{nd}$ most important natural source? Not sure if most recent 14C-CH4 tells the same story. The authors might want to consider to tone this statement down a bit.

P37 L4: The CH4 flux from the ocean to the atmosphere

P37 L7: not to forget CH4 itself

P37 L15: …during the past climate changes… could you be more specific on time scales?

P37 L15-16: This sentence should be in the section that describes the box model method

P37 L18: polar firn and ice

P37 L20: cite Loulergue 2008

P37 L29: Greenland temperature change

P37 L30: …around Greenland is linked to the….

P37 L32: with abrupt Northern Hemispheric warming during DO…

P38 L8-11: Please develop this sentence, I cannot follow the line of argumentation

P38 L25: cover only a part…

P38 L34: To my knowledge, there is no plural for "ice", "ices" doesn't exist. Please correct here and everywhere else.

P38 L37: On the other hand…

P39 L20: record from the early Holocene and investigate…

P40 L30: starts below 400 m and continues to the bottom of the core at 1004 m

P41 L3: SEM of n=2???

P41 L25-27: How do you expand the gas into the GC and ensure 100% sample transfer into the GC? Do you flush the headspace with He? Could this He create a blank that varies with He cylinder?

P42 L6-7: SEM of 2.0±1.0ppb… please explain your calculations

P42 L10: SEM of n=2???

P42 L12: Everything will agree well if you average often enough. Please develop transparent and unbiased approach.

P42 L13: If this is your final uncertainty estimate, please state this in a clear and dedicated sentence.

P42 L17: You might want to consider leaving the "M" at the beginning of the sentence.

P42 L34: Please quantify the temporal resolution of your Siple Dome data

P44 L31: from Asian and Amazon wetlands

P47 L29: Another argument to use NEEN discrete is that these data were measured at OSU as well. No?

P48 L9: Please provide a transparent description of how you calculate 1.4 ppb.

P48 L13: "regarded as more accurate", not exactly a scientific term. Haven't the uncertainties of 1.4 ppb and 1.5 ppb just been stated for Siple and WAIS, respectively, in the previous sentence? I strongly recommend to re-think the presented uncertainty model. Accuracy is part of the uncertainty but doesn't seem to be considered in the uncertainties presented in this manuscript. If ice core specific accuracy problems infiltrate IPD analysis, how reliable is the magnitude of IPD reconstructions?

P48 L17: Describe the calculation of the envelope

P49 L3: State lifetime and transport time you assume in model

P49 L7: 15 Tg

P49 L8: IPD-2, 134 to 115 Tg, I don't see anything >125 Tg in Figure S3. I am confused with some of the quantifications in the following text, often, the stated numbers don't seem to match the values in the figures.

P49 L9: What trend? I can't see a trend in Fig. S3.

P49 L12: Where are these numbers from? Fig S3? Fig 4?

P49 L12: The minima at 10.7 ka is only a feature in IPD-1, not in IPD-2. Again, this is misleading, as IPD-1 is based on much lower temporal resolution. Please use IPD-2 where possible.

P49 L16: "…from 29 to 35% during the 11.5 to 10.0 interval." I cannot even see a value <30% or >34% in Fig. S3, even including the envelope.

P49 L7-16: This entire section needs major revision. The numbers don't seem to match the presentation in the figures, the description jumps back and forth between Figures in supplements and main text, the text flow makes it hard to understand.

P49 L22: What conclusion?

P49 L30: Your 13 Tg estimate is based on IPD-1, which matches the 8.2 Tg within uncertainties. What value would you get from IPD-2?

P51 L4: Please find a meaningful section title

P51 L27ff: please generalise less and quantify more.

P62, Figure 2: Please add next to axes what these proxies actually show, e.g. warmer, wetter, colder dryer with arrows.

P64, Figure 4: Please add reconstructions based on IPD-2

P65, Figure 5: Please synchronize x-axes directions in top panels

P66 L2: Uncertainties or errors?

P67ff: Please include supplements in main text

---

## Author Response (AR2)

**Editor Decision: Reconsider after major revisions** (08 Jun 2017) by Eric Wolff

Comments to the Author:

Thank you for the considerable changes you have made to the paper. These have certainly improved it. I apologise for the delay in obtaining reviews on this version. As you will see, I sent the new version to two reviewers - one was the same as for the previous version, the other was a new reviewer. Both reviewers agree that your paper is valuable and should eventually be published. However, although they go into different levels of detail, both still have significant issues on two aspects of your paper: the analytical uncertainty you quote, and the use of SD data for the IPD. You need to address their detailed comments as well as these two major issues.

I would like to particularly emphasise the need to either alter or explain better the uncertainty calculation. Your paper has been seen by 3 reviewers, all of them VERY familiar with measurements of methane in ice and with the sources of error in such measurements. Despite your explanations all 3 of them (and me as editor) still don't understand how it is possible, when you have a strongly varying blank, to arrive at such a small uncertainty. As one reviewer says, you need either to come up with a convincing discussion of this, or resist the temptation to claim a very low uncertainty.

I would also like to expand further rev 1's comment about the blank. They correctly point out that, if you draw 4 samples from the range 5-15 ppb, the SE of the mean (SD/2) is indeed around 2 ppb. This implies that the intraday uncertainty is no less than the inter-day uncertainty. The uncertainty on each sample is controlled by the blank of that sample, not by the uncertainty of the daily average so you should not be dividing by root(4) to obtain it. It seems impossible that the uncertainty on a measurement can actually be less than the standard deviation of the blank measurements, which presumably is 4 ppb.

I look forward to seeing a revised manuscript.

*We appreciate the editor for thoughtful comments and patient guidance. Agreeing with both reviewers, we added more details on analytical method, and we re-defined data uncertainty including the uncertainties due to corrections for daily systematic offset (e1) and the other effects (e2). The uncertainty of daily systematic offset (e1) was estimated from results of bubble-free ice with a standard air (now we use "bubble-free ice" instead of "blank ice" because the ice was used only for e1 estimation). As we used the average of four "bubble-free ice" results for the daily systematic offset, the best estimation of the daily e1 is the standard error of the mean for the day. The e2 includes uncertainty from solubility correction and inhomogeneous $CH_4$ distribution in ice. We estimated the e2 with intra-day duplicates from the same depths. Because a daily systematic uncertainty was applied to all the samples for the day, any intra-day distribution of $CH_4$ from the same depth should indicate e2. The total uncertainty of individual ice is now calculated by $((e1)^2 + (e2)^2)^{1/2} = (1.9^2 + 3.3^2)^{1/2} = 3.8$ ppb. Because we used the average of a duplicate ice samples for the same depth, the data uncertainty for data plot and IPD calculations is $3.8/2^{1/2} = 2.7$ ppb. The details are described in our response to the reviewers' comments.*

*We revised solubility correction method. Instead of calculating Henry's law and applying a correction factor of 1.0058, we estimated solubility effect from the results of direct measurement of $CH_4$ mixing ratio of air remained in refrozen meltwater. Mean difference between before and after revising the solubility correction is less than 1.0 ppb, while intra-day scattering among the four bubble-free ice samples remains unchanged (average SEM = 1.9 ppb). We also clearly stated the data rejection scheme, and carefully reviewed our data set. Due to changes in correction method, two data points previously rejected are now included in data set, while five data points were newly rejected. Regarding IPDs, we present a combined IPD by using IPD-1 for 9.0 – 10.0 ka and IPD-2 for 10.0 – 11.5 ka, because IPD-2 is better constrained in this period as the reviewer recommended. However, it does not change the major findings of our manuscript.*

*Finally, we included the important information in Supplements into main text and the other contents that are not essential were removed. The English was improved as the reviewer suggested.*

*Below our responses are written in red italics, and* the point-to-point revisions are in Times New Roman*, where* the words of the original manuscript version are black*,* the changes of the first major revisions are shown in blue *and* those of the second major revisions appears in green.

Hello,

The manuscript improved considerably, however, major aspects of the analysis are presented in a confusing or even misleading way. I have no doubt that the author group can iron this out and that this manuscript will eventually be suitable for publication in CP.

Cheers

Thanks a lot to the authors for their considerable revisions of the original manuscript. In my view, the new version is a great improvement. The structure gained clarity and the refined focus on the time scale makes the entire paper appear more concise. Again, this manuscript might require some language editing or input by the native speaker of the author list. On top of the good development, I

recommend major revisions on several aspects of the new manuscript. These include i) the description of the analytical system and its uncertainties, ii) the use of IPD-1 and IPD-2, iii) Section

3.3, Comparison with late Holocene variability, iv) conclusions, v) Supplements.

**i) the description of the analytical system and its uncertainties**

I find the part on the analytical uncertainty of the methods section extremely hard to follow and even after careful re-reading, I am not sure what data were used to calculate the stated averages, pooled standard deviations etc. The authors have chosen to publish the data paper before their new method is published in a separate publication. In my point of view, this requires a sufficient description of method and uncertainty if it is claimed that the new method is superior to existing methods, even if a dedicated publication follows. It seems to me that the authors should increase their resistance to the temptation to choose statistical methods that understate the uncertainty of their analytical system, as this leaves the reader rather suspicious.

➔ We agree that our description of analytical method and uncertainty was not sufficient. Now we elaborated more description on analytical method, especially on correction methods.

Our new estimation of analytical uncertainty for each individual ice is 3.8 ppb and 2.7 ppb for the mean of duplicates used in the data plot and IPD calculation. Details on revised analytical uncertainty are found in our responses to the following comments. To improve our description of the analytical method, we added sentences below:

➔ "Different solubilities of each air component cause preferential dissolution during melting procedure.

As the solubility of $CH_4$ is higher than the other major components of air – nitrogen ($N_2$), oxygen ($O_2$), Argon (Ar), the $CH_4$ mole fraction of the extracted air is lower than originally enclosed air (solubility effect). The original $CH_4$ mole fraction of air enclosed in ice sample is estimated from residual gas fraction and $CH_4$ mixing ratio in air remained in refrozen meltwater (retrapped air). Residual gas fraction is a measure of how much air is retrapped during refreeze, which is defined as ratio of amount (pressure)

of air extracted from the $2^{nd}$ gas extraction to the $1^{st}$ extraction. The $2^{nd}$ gas extraction was carried out by an additional melting-refrezing process after the $1^{st}$ extraction and evacuation of the sample cup. Residual gas fraction was measured during the 10 experimental days.

Mean residual gas fraction is $1.05 \pm 0.13\%$ ($1\sigma$, n=60) for SDMA ice samples and $0.38 \pm$

$0.08\%$ ($1\sigma$, n=40) for bubble-free ice. The test with ice samples from Styx glacier, Antarctica revealed that $CH_4$ mixing ratio in retrapped air is enriched 3.11 times (n=12) for glacial ice and 2.98 times (n=7) for bubble-free ice. Then the solubility effect is corrected by using a simple mass balance calculation."

➔ "Daily systematic offset correction was applied to account for the daily-varying system condition. To do this, we measured four bubble-free ice samples every day with SDMA ice samples. The experimental procedures for the bubble-free ice were identical to the SDMA ice.

After the sample flasks are evacuated, standard air is injected into the flasks containing bubble-free ice, so that it returns similar amount of air into the sample loop to typical size of

SDMA ice when the extracted air inside the bubble-free ice flasks is expanded. The solubility correction for the bubble-free ice was done by the same formula as SDMA ice samples, but using different $CH_4$ mixing ratio (see above) and residual gas fraction. After corrected for solubility effect, the daily systematic offset is calculated by difference between $CH_4$ mixing ratio of the injected standard air and results from the four flasks containing bubble-free ice.

The daily systematic offset fluctuates from 5 to 15 ppb during SDMA measurements, and is subtracted from the samples corrected for gas solubility effect, including bubble-free ice and

SDMA samples. This is one of the major difference with OSU wet extraction system, where the daily systematic offset was interpolated from the results of blank tests carried out between several days (Mitchell et al., 2011).

The bubble-free  ice was made by chilling the degassed ultrapure water (resistivity >

18.2 $M\Omega \cdot cm$ at 25°C) slowly from the bottom in a closed stainless steel chamber. From gas extraction test using our bubble-free ice without injecting standard air, we observed that no significant pressure increase at the pressure gauge with a detection limit of 0.01 Torr (corresponding to less than 0.03% of sample air pressure in the extraction line) after melting- refreezing the bubble-free ice."

Please make sure it is well described how you achieve the values you present and what data they were calculated from. For example on page 42, line 6-7 of the document including the tracked changes, you state a standard error of the mean (SEM) of 2.0±1.0 ppb. To my understanding, the SEM is the uncertainty measure, it has no uncertainty itself. What has been calculated here? In general, SEM's may be used as measure of uncertainty when a sufficient number of samples have been measured. In this manuscript, the authors seem to apply SEM to sample sizes of n=2. If the authors have a good reason to use SEM for n=2, they should clearly state why and account for the obvious lack of sample size (student's t). I feel very strongly that using SEM on n=2 is not OK and needs changing in every uncertainty calculation (e.g. p. 42, L10). Presenting a formula may help clarify what has been calculated. An understandable and ethically correct uncertainty demonstration will provide more credibility to the technical aspects of this manuscript than unrealistically small values. If the system is a new game changer, it should be possible to show outstanding performance using appropriate statistical methods.

➔ To derive a more comprehensive and conservative error estimation, we address two types of uncertainty sources: uncertainty of systematic offset ($e1$) and other causes ($e2$). The daily systematic offset uncertainty ($e1$) is estimated as standard error of the mean (SEM) of four flasks containing bubble-free ice and standard air (Now we use "bubble-free ice" instead of "blank ice" because we use the "bubble-free ice" only to estimate daily systematic offset and its uncertainty ($e1$)). Here we use SEM (n=4) because the daily systematic offset is calculated from the mean of the four bubble-free ice samples. The average of daily $e1$ was 1.9 ppb. To estimate the $e2$, (e.g., uncertainty from variable degree of solubility equilibrium of $CH_4$ and inhomogeneous $CH_4$ distribution in ice at the same depth), we used results from duplicates. Because any duplicates from same depths were analyzed on the same days, the differences in duplicate ice measurements should reflect uncertainty other than $e1$. Thus, the $e2$ was determined by pooled standard deviation (PSD) of ice duplicates of all depths. PSD of SNU data set is 3.3 ppb. Taking in to account the above two uncertainty terms ($e1$ and $e2$), we state a revised analytical uncertainty of 3.8 ppb ($((e1)^2 + (e2)^2)^{1/2}$) for individual ice measurement. With regard to the data used in graphs and IPD reconstruction, we used the mean values of duplicates with uncertainty of 2.7 ppb ($3.8/(2^{1/2})$). Based on this argument, we added the paragraph below:

➔ "Here we consider two types of uncertainty sources: uncertainty in (1) estimating daily systematic offset and (2) other causes. The former indicates uncertainty of the daily systematic offset ($e1$). As the daily systematic offset is calculated from the mean of the four flasks with bubble-free ice and standard air, scattering of the bubble-free ice samples can induce uncertainty in the systematic offset correction. The daily $e1$ is estimated with standard error of the mean (SEM, n=4), because the daily systematic offset is calculated from the mean of the four bubble-free ice samples. The average of daily $e1$ is 1.9 ppb. The latter ($e2$) includes uncertainty due to solubility correction and inhomogeneous distribution of $CH_4$. Given our solubility correction uses the mean value of residual gas fraction and the ratio at which $CH_4$ enriches in retrapped air, different solubility effect and/or inhomogeneous $CH_4$ distribution in individual ice causes offset between adjacent duplicate ice samples analysed on the same day. As the duplicates from same depths were measured on the same day, we estimated the $e2$ with pooled standard deviation (PSD) between duplicate measurements from entire depths, which yields 3.3 ppb. Taking the $e1$ and $e2$ into account together, the final uncertainty of individual measurement is given as 3.8 ppb by error propagation. The uncertainty for the mean of duplicate results is obtained by dividing the individual uncertainty by square root of 2, yielding 2.7 ppb."

This manuscript version has a lot of emphasis on the use of bubble-free blank ice to determine the system blank. The authors state that based on their method to produce the blank ice, they can exclude that CH4 is introduced with the blank ice. From personal experiences, I wouldn't trust that statement. Dedicated blank ice tests including several labs showed that while it is easily possible to produce bubble free ice, there is no such thing as gas-free blank ice. The gas content of the blank ice varied both between the labs and between the batches produced in each lab. All labs used deionised water and pumped on the water for 90 minutes and froze large crystals bottom up… While blank ice allows for a valuable tests, great care has to be taken in the use and interpretation of the results.

➔ From our test using bubble-free ice samples without injecting any standard, we found that the pressure of the extracted air is below the detection limit (0.01 Torr) of our system. Compared to the air pressure of the typical SDMA ice samples is ~30 Torr when expanded into the vacuum line, air content in bubble-free ice is less than 0.03% of the SDMA ice. We added this information on our method section.

Based on 4 daily blank ice analyses, the authors state an intra-day SEM of 2.0±1.0 ppb and a variation of the inter-day mean between 5 and 15 ppb. I think the authors intention to present these data is to show that even if the inter-day variation is as large as 15 ppb, the daily blank can be quantified with low uncertainty (SEM 2.0±1.0 ppb) and a correction can therefore be accurately determined. I find this hard to understand. If I chose 4 random values between 5 and 15 (as representative for the large variability of the inter-day blank) and calculate the SEM for these 4 values, the SEM is <2.5. The authors state that the blank estimate of OSU has a high uncertainty of 10% due to the small peak size, but don't seem to think this is a problem when quantifying the blank ice contribution with higher relative uncertainties. Please clarify.

I would think that introducing a known amount of air with a known CH4 mixing ratio into the melt chambers and to analyse that air in the same way as an ice core sample could provide a useful measure of analytical error. This could be done in a dry chamber and over melt water to test for differences. Such experiments could be useful to determine the **accuracy of the measurements**.

➔ Here we disagree with the Reviewer in including uncertainty induced from inter-day blank fluctuations because we measured the systematic offset daily. This is one of main difference from OSU method. If we estimate the systematic offset once in several days, it should introduce a certain amount of uncertainty in interpolating the inter-day blank.

The authors use table R1 to show the reproducibility of the analytical system. I find this is a very useful way to state the potential uncertainty/reproducibility of the system and would suggest to improve the clarity and possibly give this more weight in the formulation of the uncertainty estimate.

On p. 42 L. 12, the authors refer to the data in table R1 and state, that the differences between the averages from the 1st and the 2nd measurement pair is on average 1.9 ppb. Even though this is nice, it ignores the uncertainty of each single measurement as well as the differences within each sample pair. Again, the authors do not clearly state how the uncertainty in R1 is calculated for 1st and 2nd measurement, is it SEM again for n=2 or 1σ? The authors provide one depth interval for all for samples in table R1, where the depth is sometimes stated to the mm. Are really all four samples from the same mm depth? In my view, this table should include all 4 measurements per depth level.

You can then add columns for 1σ or SEM for all four, and calculate the pooled standard deviation (PSD) of the whole lot afterwards as total uncertainty estimate for **measurement precision**. I am not sure if this is what the authors describe on p. 42, L. 13 (PSD of 1.4 ppb)? Did you calculate the PSD

based on 4 individual measurements per depth or based on the averages from 1st and 2nd measurements? If this is the "final" uncertainty that the authors intend to state for their analytical system, they should clearly say so in a dedicated sentence, not hidden in a sentence and furthermore in brackets.

➔ Now we include all the four measurements in Table R1. The uncertainties in Table R1

indicates 1 standard deviation between duplicates. PSD from quadruplicate measurements is 3.0 ppb. The PSD among the four individual ice measurements is similar to the PSD of duplicate measurements of the SNU data ($e2$ = 3.3 ppb). The PSD among the mean of duplicate analyses of the 1st and 2nd measurements is 1.1 ppb. The good agreement among the duplicate means indicates good reproducibility. The better PSD obtained from the mean of duplicates than from the four individual measurements could be attributed to casual underestimation due to insufficient number of replicate pairs, as well as to non-Gaussian distribution of individual measurement that might results better agreement when averaged

2 to 4 replicates, even though the error of each data point is larger. There is no evidence that each duplicate analysis (consist of 3 injections) is Gaussian, and even if the duplicates individually follow Gaussian distribution, the final results (mean of duplicates) are not necessarily Gaussian (e.g., Anderson, 2011). We added below paragraph and Table:

➔ "We made additional measurements using adjacent samples (depth difference of 10 cm) at randomly selected 8 depth intervals to examine reproducibility and long-term stability of our system. The second measurements of duplicates were performed 8 to 80 days after the first analysis. Table 1 displays quadruplicate results at each depth. PSD between the mean of duplicate analyses of the first and second measurements on different days yields 1.1 ppb. The good agreement between duplicate means indicates good reproducibility of our system. In the meanwhile, PSD of the quadruplicate measurements is 3.0 ppb, which is similar to PSD of duplicate samples for the entire data set (3.3 ppb)."

➔ Table 1 is as below:

| | 1st measurements | | | | | 2nd measurements | | | | |
|---|---|---|---|---|---|---|---|---|---|---|
| Depth | Dup.1 | Dup.2 | Mean | 1sigma | Date | Dup.1 | Dup.2 | Mean | 1sigma | Date |
| (m) | (ppb) | (ppb) | (ppb) | (ppb) | (dd/mm/yy) | (ppb) | (ppb) | (ppb) | (ppb) | (dd/mm/yy) |
| 523.150 | 634.8 | 634.7 | 634.7 | 0.1 | 27-1-14 | 637.5 | 634.3 | 635.9 | 1.6 | 24-2-14 |
| 530.950 | 669.0 | 665.8 | 667.4 | 1.6 | 03-2-14 | 669.4 | 670.7 | 670.0 | 0.7 | 24-2-14 |
| 558.295 | 682.5 | 678.2 | 680.3 | 2.2 | 14-3-14 | 687.5 | 678.3 | 682.9 | 4.6 | 02-4-14 |
| 559.850 | 689.8 | 680.3 | 685.0 | 4.7 | 03-2-14 | 683.8 | 690.0 | 686.9 | 3.1 | 26-3-14 |
| 561.150 | 687.8 | 689.2 | 688.5 | 0.7 | 14-3-14 | 684.0 | 690.4 | 687.2 | 3.2 | 02-4-14 |
| 562.407 | 687.2 | 685.5 | 686.4 | 0.8 | 26-3-14 | 689.4 | 686.4 | 687.9 | 1.5 | 02-4-14 |
| 575.913 | 679.2 | 679.2 | 679.2 | 0.0 | 07-2-14 | 686.7 | 678.9 | 682.8 | 3.9 | 28-3-14 |

P. 42, L 30: You mention the difference OSU-SNU of 3ppb and present a hypothesis on what might have caused this offset. However, this dilemma shows that an analytical uncertainty of 1.4 ppb cannot include the entire uncertainty to the NOAA scale. It would be great to see this resolved, i.e.

precision and accuracy, or at least clearly discussed if unresolvable.

➔ The uncertainty of standard air affects accuracy of measurements, not analytical precision.

According to the results of calibration of the standard air that was carried out at NOAA

shows PSD of about 0.4 ppb, while the repeatability of NOAA04 scale has been reported as <

1.5 ppb for ambient air mole fractions (~1700 ppb) (Dlugokencky et al., 2005). As described above, our new uncertainty model yields average offset between OSU and SNU data set of ~0.1 ppb, which lies within the precision of our working standard.

On p. 42 L. 34, the authors state that their new Siple Dome data have the third highest temporal resolution. However, they don't say how high it is, as they do for the others. Please quantify, otherwise your statement of 3rd highest is meaningless.

➔ The mean temporal resolution of our Siple Dome composite record is ~26 years, while those of WAIS discrete and continuous data is ~20 and ~2 years, respectively. The sentence was changed as below:

➔ "Our new Siple Dome $CH_4$ composite data have mean temporal resolution of ~26 years. The WAIS Divide continuous $CH_4$ records shows much higher resolution (~2 years), but does not cover the entire early Holocene period (Rhodes et al., 2015)."

**ii) Use of IPD-1 and IPD-2**

The authors calculate IPD's based on NEEM discrete and Siple discrete (IPD-1) and NEEM discrete and WAIS continuous (IPD-1). Unfortunately, the authors do not describe how the uncertainty envelope is calculated. Furthermore, you do not state the temporal resolution of the Siple data, as compared to the WAIS continuous (p. 42, L. 34).

➔ The error range of IPD was calculated from synchronization uncertainty and $CH_4$ measurement uncertainty. The Monte Carlo-based synchronization routine produces 20 sets of age offset at each of the tie points that were resampled every 30 years. The age offsets were linearly interpolated and added to the initial synchronization ages, creating 20 sets of synchronized age scales. The standard deviation of the 20 IPD records calculated from the 20 synchronized age scales is taken as synchronization uncertainty. The $CH_4$ measurement uncertainty was estimated with the stated uncertainty of each data set (4.3 ppb for NEEM discrete / 2.7 ppb for SDMA / 1.5 ppb for WAIS continuous, 1 sigma). To check the sensitivity of the uncertainties, we carried out Monte Carlo simulations. We produced 1000 different sets of IPD, which vary randomly with Gaussian propagation in their ages and $CH_4$ concentration uncertainties. Each IPD was annually interpolated and smoothed by a $1/1000$ $year^{-1}$ low-pass filter. The standard deviation of the 1000 smoothed IPDs was taken as the uncertainty envelop of IPD. In Figure R1 and R2, the uncertainties were reported as 95% confidence interval by multiplying 1.96 to standard deviation. We also added the mean temporal resolution of each data set. The paragraph was updated as below:

➔ "Temporal uncertainty (synchronizing error) was determined for each point as 1 standard

 The uncertainty range of IPD was calculated from synchronization uncertainty and CH$_4$ data uncertainty. To estimate synchronization uncertainty, we created 20 IPDs from the 20 sets of maximum correlation time series, and the standard deviation of the 20 records was taken as synchronization uncertainty for each of the data points. The CH$_4$ data uncertainty was estimated with the stated uncertainty of each data set (4.3 ppb for NEEM discrete / 2.7 ppb  for SDMA / 1.5 ppb for WAIS continuous, 1 sigma). To check the sensitivity of the uncertainties, we carried out Monte Carlo simulations. We produced 1000 different sets of IPD, which vary randomly with Gaussian propagation in their ages and CH$_4$ concentration uncertainties. Each IPD was annually interpolated and smoothed by a 1/1000 year$^{-1}$ low-pass filter. The cutoff frequency of 1000 years was chosen to examine multi-centennial to millennial scale change, because  the IPD calculation is very sensitive to high frequency variability of CH$_4$ records from both poles. To report 95% confidence interval, we multiplied the standard deviation by 1.96 and enveloped the IPD."

There are a few periods between 11.4 and 10.5 ka where WAIS and Siple are different by ~20 ppb, which produces the different IPD shapes. While WAIS continuous includes several 100 data points (mean resolution of 2 years), I manually count ~15 in the Siple record in Figure 3. Moreover, both the WAIS and NEEM records share several features, that are not resolved at all in the Siple record. It seems to me, that focussing your IPD interpretation on IPD reconstructions that are best constraint makes most sense. I understand the temptation to focus your analysis on the new data. However, I would strongly recommend to shift IPD-2 from Supplements into the main figure and to use IPD-2 for interpretation between 11.5 and 10.0 ka. This will still leave you with the IPD increase until 9.5 ka, but is better constraint and has less artefacts. In the box model, this would create rather stable emissions from tropics until 10.0 and a decrease thereafter. To me, this would seem the best IPD reconstruction possible.

➔ We agree with the Reviewer and we create a combined IPD using IPD-1 for 10.0 – 9.0 ka and IPD-2 for the rest of the studied interval (Fig. R1). The new IPD composite maintains major findings of our manuscript, but shows less fluctuation for 11.5 – 10.0 ka period. Figure R2 presents revised box model results using combined IPD. It does not change our findings, showing a gradual strengthening of boreal sources while tropical emission was reduced along with insolation in northern hemisphere. We also changed colors of Greenland- and Antarctic CH$_4$ records in Figure R1 and R3 for better readability.

[Figure]

**Figure R1.** Inter-polar difference (IPD) reconstructions. Top: high resolution CH₄ records from Greenland and Antarctica, synchronized to NEEM gas age scale by Monte Carlo procedure. Middle: Millennial-scale IPD composite derived from IPD-1 and IPD-2. Shaded area indicates 95% significance interval. Bottom: Proxy-based temperature reconstruction for northern mid to high latitude and boreal CH₄ emission from northern thermokarst lakes.

[Figure]

**Figure R2.** Revised 3-box model results from the combined IPD and its 95% significance interval.

**iii) Section 3.3, Comparison with late Holocene variability**

This section feels completely unattached to the rest of the manuscript. The figures are described but any quantification is hard to follow. The section title is meaningless and confusing.

➔ Actually, the Section 3.3 was added following comment of one of the previous reviewers for the first open discussion. However, we agree with the reviewer that this section is not in line with the main text, and therefore we deleted this section.

**iv) Conclusions**

Please quantify your statements and avoid generalisations, such as high resolution (how high), agree well with previous measurements (how well), four CH4 minima (how big are anomalies, how long in duration), first reporting of IPD increase from 11.5 to 9.9 (how much increase), elevated emissions from NH extra-tropics (how much did emissions from NH extra tropics increase), RMS amplitude smaller (how much) and what does this even mean in conclusions?

➔ To avoid ambiguity, we revised the conclusion paragraph as below:

➔ "We reconstructed a new high resolution $CH_4$ record during the early Holocene from Siple Dome ice core, Antarctica, to study millennial $CH_4$ variability and its natural controls under Holocene interglacial condition. The new Siple Dome record agrees well with previous records measured at OSU within analytical uncertainty, showing a mean difference of 0.1 ppb. By combining the two data sets, we present a SDMA $CH_4$ composite record covering from ~7.7 to 11.6 ka. ~~the early Holocene $CH_4$ time series in high resolution to discuss natural processes that control the millennial scale $CH_4$ variations in the past atmosphere. Since the new SDMA data agree well with previous measurements at OSU, we made SDMA $CH_4$ composite data covering ~7.7 to 11.6 ka.Our results show a series of theand the evidence suggests that the low latitude source changes were the major causes of the early Holocene $CH_4$ minima.~~ Further, this study presented the millennial scale change of IPD, which was calculated from high resolution discrete data set of NEEM and SDMA, and a continuous record of WAIS Divide. Here we reported that the IPD increased by ~13 ppb  from the onset of the Holocene to  ~9.5 ka following the temperature rise in NH extra-tropical regions. The three-box model demonstrates that  NH extratropical emissions elevated by ~11 Tg $yr^{-1}$, while tropical emission was reduced by ~9 Tg $yr^{-1}$, resulting the increased contribution of the NH extra-tropical sources by ~5%.  However, the North Atlantic induced changes in low latitude hydrology cannot fully explain the $CH_4$ minimum at ~10.2 ka. High resolution IPD and 3 box source distribution model results indicate that fraction of boreal sources increased by 5 % during the early Holocene, which indicates that fraction of boreal sources increased from ~10.7 ka and remained high until ~9.3 ka. To summarize, the millennial scale variability of $CH_4$ during the early Holocene was primarily controlled by low latitude climatic and surface hydrological conditions, while relative boreal source contribution increased during 10.7-9.3 ka by newly developed high latitude sources following terrestrial deglaciation. Further, our observations imply that ~20-40 ppb of $CH_4$ change could be induced naturally by low latitude hydroclimate changes."

**v) Supplements**

I would like to encourage the authors to shift all these important information on analytical reproducibility, gas age uncertainty, IPD-2, age model effect on previous gas age scale and box model data in the main text. Figures can be reduced size, e.g. one column etc, but including those information will increase value of paper! I don't recall being frequently referred to supplements in the main text. Having a lot to read every day, I would probably never get to read supplements that aren't constantly advertised in the main paper, unless this is the most critical paper for my current work.

➔ Here we moved part of the supplements into main text, because some of them are essential for the discussion of this manuscript. We included information on chronology and gas age uncertainty in the main text. The IPD-2 and corresponding box model data were deleted because this information is already included in our new IPD. However, description of the results from the Styx glacier ice core was not included in the main text, because comparison of data set measured in different period and from different ice core may not be directly related to data interpretation.

[Figure]

**Figure R3**. Various IPD scenarios during the early Holocene. Top: High resolution CH₄ records available covering the early Holocene. Bottom: Different IPDs derived from various pairs of data set.

**General comments**

P36 L11: human influence was substantially smaller

➔ The sentence was modified as below:

➔ "In contrast, the early Holocene was a period when human influence should have been was substantially smaller, so that it allows allowing us to elucidate the natural controls under interglacial conditions more clearly."

P37 L1: is geological CH4 really the 2nd most important natural source? Not sure if most recent 14C-CH4 tells the same story. The authors might want to consider to tone this statement down a bit.

➔ According to synthesis of IPCC 5th report, geological sources (including oceans) are the largest natural emission except for natural wetlands, but the range of minimum and maximum estimations in literatures is largely overlapped with freshwater sources. We modified the sentence as below:

➔ "Other, more minor sources include Geological CH4 released from mud volcanoes and gas seepages through faults is the second most important natural source (e.g., Etiope et al., 2008 and references therein). pyrogenic sources such as wildfire and biomass burning (Andreae and Merlet, 2001; Ferretti et al., 2005; Hao and Ward, 1993), and microbial digestion by wild animals and termites (e.g., Sanderson, 1996)."

P37 L4: The CH4 flux from the ocean to the atmosphere

➔ We changed the sentence as below:

➔ "The  $CH_4$ flux from the ocean to the atmosphere is considered as too small to create a significant change in global budget compared to the other sources (e.g., Rhee et al., 2009)."

P37 L7: not to forget CH4 itself

➔ The sentence was modified as below:

➔ "The major sink of atmospheric $CH_4$ is photochemical reactions (oxidation) with the hydroxyl radical (OH), which is mainly controlled by atmospheric temperature, humidity, and  the mixing ratio of $CH_4$ itself and non-methane volatile organic compound (NMVOC) (e.g., Levine et al., 2011 and references therein)."

P37 L15: …during the past climate changes… could you be more specific on time scales?

➔ The cited references are dealing with $CH_4$ sink contribution on glacial-interglacial (LGM-PI) time scales. The sentence was changed as below:

➔ "However, recent model studies suggested  that $CH_4$ changes between glacial- and interglacial conditions were driven mostly by source changes, rather than sink changes (Weber et al., 2010; Levine et al., 2011)."

P37 L15-16: This sentence should be in the section that describes the box model method

➔ We removed this sentence because the box model does account for sink term.

P37 L18: polar firn and ice

➔ We modified following the comment, but also reworded thoroughly.

➔ "Since direct monitoring of $CH_4$ mixing ratio in modern atmosphere  only covers the late 20$^{th}$ and early 21$^{st}$ centuries (Dlugokencky et al., 1994, 2011), polar firn and ice is the unique archive that preserves the ancient atmosphere for the research of fossil air older than 20$^{th}$ century. "

P37 L20: cite Loulergue 2008

➔ We cited Loulergue et al. (2008) in the sentence.

P37 L29: Greenland temperature change

➔ The sentence was modified as below:

➔ "The resemblance between water stable isotope records from Greenland ice cores, a proxy for Greenland temperature  change, and global CH$_4$ mixing ratio on millennial time scales is also well known. "

P37 L30: …around Greenland is linked to the….

➔ This was changed as below:

➔ "This implies that local temperature change around Greenland is linked to  the major CH$_4$ sources in low latitudes (e.g., Brook et al., 1996; Chappellaz et al., 1993; Huber et al., 2006; EPICA Community Members, 2006; Grachev et al., 2007, 2009)."

P37 L32: with abrupt Northern Hemispheric warming during DO…

➔ We deleted the paragraph.

P38 L8-11: Please develop this sentence, I cannot follow the line of argumentation

➔ We removed the sentence.

P38 L25: cover only a part…

➔ We removed the sentence.

P38 L34: To my knowledge, there is no plural for "ice", "ices" doesn't exist. Please correct here and everywhere else.

➔ Done.

P38 L37: On the other hand…

➔ Here we thought "in principle" is better than "on the other hand".

➔ "In principle, stable isotope ratios of CH$_4$ help us to distinguish the types of sources – biogenic, pyrogenic, and geologic."

P39 L20: record from the early Holocene and investigate…

➔ The sentence was modified as follows:

➔ "Therefore, in this study we present a new high-resolution CH$_4$ record from  the early Holocene and  investigate natural control mechanisms under interglacial condition."

P40 L30: starts below 400 m and continues to the bottom of the core at 1004 m

➔ The sentence was changed as below:

➔ " The brittle zone of SDMA ice starts below 400 m  and continues to the bottom of the core at 1004 m (Gow and Meese, 2007) and samples from this region are more likely to be fractured. "

P41 L3: SEM of n=2???

➔ As we re-defined the analytical uncertainty, the sentence was modified as below:

➔ "All samples were duplicated, so that our final CH$_4$ data were presented by averaging the results of duplicate analysis from the same depth.  The analytical uncertainty of each data point is estimated by the uncertainty of individual ice measurement divided by square root of 2 (see below)."

P41 L25-27: How do you expand the gas into the GC and ensure 100% sample transfer into the GC?

Do you flush the headspace with He? Could this He create a blank that varies with He cylinder?

➔ No He flush is used in our method. The sample gas in flask is expanded into evacuated vacuum line and sample loop of GC. We added this into the sentence to clarify it:

➔ "The extracted air in the headspace was expanded into the evacuated vacuum line and sample loop of a gas chromatograph (GC) equipped with a flame ionization detector (FID) to measure $CH_4$ mixing ratio. After detecting the $CH_4$ peak in the GC chromatogram (retention time of ~1.6 minutes), the vacuum line and sample loop is evacuated again prior to the next injection."

P42 L6-7: SEM of 2.0±1.0ppb... please explain your calculations

➔ This value was calculated from average of the daily SEM of four bubble-free ice samples, indicating the uncertainty of the daily systematic offset correction ($e1$). Therefore, we added below paragraph to explain it:

➔ "Here we consider two types of uncertainty sources: uncertainty in (1) estimating daily systematic offset and (2) other causes. The former indicates uncertainty of the daily systematic offset ($e1$). As the daily systematic offset is calculated from the mean of the four flasks with bubble-free ice and standard air, scattering of the bubble-free ice samples can induce uncertainty in the systematic offset correction. The daily $e1$ is estimated with standard error of the mean (SEM, n=4), because the daily systematic offset is calculated from the mean of the four bubble-free ice samples. The average of daily e1 is 1.9 ppb. The latter ($e2$) includes uncertainty due to solubility correction and inhomogeneous distribution of $CH_4$. Given our solubility correction uses the mean value of residual gas fraction and the ratio at which $CH_4$ enriches in retrapped air, different solubility effect and/or inhomogeneous $CH_4$ distribution in individual ice causes offset between adjacent duplicate ice samples analysed on the same day. As the duplicates from same depths were measured on the same day, we estimated the $e2$ with pooled standard deviation (PSD) between duplicate measurements from entire depths, which yields 3.3 ppb. Taking the $e1$ and $e2$ into account together, the final uncertainty of individual measurement is given as 3.8 ppb by error propagation. The uncertainty for the mean of duplicate results is obtained by dividing the individual uncertainty by square root of 2, yielding 2.7 ppb."

P42 L10: SEM of n=2???

➔ In the previous version, we meant SEM between duplicate measurements (n=2). However, we re-defined the analytical uncertainty and decided not to use SEM of n=2 for representing individual data uncertainty. Revised analytical uncertainty for individual ice measurement is 3.8 ppb, and uncertainty for the mean of duplicate measurements is 2.7 ppb. Please refer to our response to general comment on analytical uncertainty.

P42 L12: Everything will agree well if you average often enough. Please develop transparent and unbiased approach.

➔ We re-defined our analytical uncertainty by including uncertainty of systematic offset and solubility correction. This yields final uncertainty of 3.8 ppb for individual ice measurement. But as we made duplicate measurements for all samples and took the mean of the duplicates, uncertainty of our data set should be $3.8/2^{1/2}$ = 2.7 ppb. Please refer to our responses above.

P42 L13: If this is your final uncertainty estimate, please state this in a clear and dedicated sentence.

➔ As the reviewer pointed out in general comment, we revised the analytical uncertainty estimate. Please refer to our response to general comment on determining analytical uncertainty.

P42 L17: You might want to consider leaving the "M" at the beginning of the sentence.

➔ Modified.

P42 L34: Please quantify the temporal resolution of your Siple Dome data

➔ We stated temporal resolution of our Siple Dome data and re-worded the sentence as below:

➔ "Our new SDMA  $CH_4$ composite data have mean temporal resolution of ~26 years. The WAIS Divide continuous $CH_4$ records show much higher resolution (~2 years), but does not cover the entire early Holocene period (Rhodes et al., 2015).  record is the one of the high-resolution data set covering ~~the early Holocene after the WAIS Divide continuous (~2 years, Rhodes et al., 2015) and discrete (~20 years, WAIS Divide members, 2015) records.from 11.6 to 8.5 ka, apart from the WAIS Divide records (Rhodes et al., 2015; WAIS Divide members, 2015).~~"

P44 L31: from Asian and Amazon wetlands

➔ We changed the words.

➔ "Sperlich et al. (2015) also suggested  that a sharp $CH_4$ peak at Greenland Interstadial 21.2 (~85 ka) was caused  by emission from Asian and Amazon  wetlands."

P47 L29: Another argument to use NEEN discrete is that these data were measured at OSU as well. No?

➔ The reviewer is right. We added this rationale in the paragraph as below:

➔ "Regarding the Greenland side, we use NEEM discrete records because not only there are discrepancies between continuous- and discrete data in some intervals, but also NEEM

discrete records were measured by similar wet extraction technique at OSU (Chappellaz et al., 2013). ~~we use NEEM discrete records because there are discrepancies between continuous and discrete data in some intervals, but also because the NEEM continuous record is not exactly "continuous". Hence, here we regard the NEEM discrete, Siple Dome discrete, and WAIS Divide continuous data as more reliable ones than the others to reconstruct IPD during the early Holocene. In this study, the IPD was calculated by using our Siple Dome CH$_4$ record and a NEEM high resolution discrete CH$_4$ record (Chappellaz et al., 2013)."~~

P48 L9: Please provide a transparent description of how you calculate 1.4 ppb.

➔ Please refer to our response to general comment above.

P48 L13: "regarded as more accurate", not exactly a scientific term. Haven't the uncertainties of 1.4 ppb and 1.5 ppb just been stated for Siple and WAIS, respectively, in the previous sentence? I strongly recommend to re-think the presented uncertainty model. Accuracy is part of the uncertainty but doesn't seem to be considered in the uncertainties presented in this manuscript. If ice core specific accuracy problems infiltrate IPD analysis, how reliable is the magnitude of IPD reconstructions?

➔ Now the analytical uncertainties of SDMA and WAIS continuous records are 2.7 and 1.5 ppb, respectively. We calibrated WAIS continuous data against to SDMA data because continuous data require to be calibrated against discrete measurements, but also SDMA records show good agreement with OSU measurements. Given that NEEM discrete data were measured in OSU by using similar method as well, by doing this we can rule out any discrepancy between IPD-1 and IPD-2 due to different measurement techniques used. The sentence was changes as below:

➔ "Before IPD calculation, WAIS continuous data were calibrated to SDMA data instead of WAIS discrete record, given the discrete measurements generally have better accuracy than continuous ones."

P48 L17: Describe the calculation of the envelope

➔ Please refer to our response to general comment above.

P49 L3: State lifetime and transport time you assume in model

➔ We used lifetime of 18.7, 8.1, and 26.8 years for N, T, and S box, respectively, and transport time of 9 months following Chappellaz et al. (1997). We added these information in the text:

➔ "Following Chappellaz et al. (1997), we assume the lifetime of 18.7, 8.1, and 26.8 years in N, T, and S-box, respectively, and transport time of 9 months."

P49 L7: 15 Tg

➔ We changed the lower case "t" into the capital "T".

P49 L8: IPD-2, 134 to 115 Tg, I don't see anything >125 Tg in Figure S3. I am confused with some of the quantifications in the following text, often, the stated numbers don't seem to match the values in the figures.

➔ As the Reviewer suggested above, we presented a combined IPD (Fig. R1) and newly calculated box-model result (Fig. R2). As we replaced the discussion on IPD-2 with that for combined IPD, we changed the numbers correspondingly.

P49 L9: What trend? I can't see a trend in Fig. S3.

➔ The Reviewer is right, we revised the paragraph thoroughly with a combined IPD.

P49 L12: Where are these numbers from? Fig S3? Fig 4?

➔ The former Fig. 4 is now Fig. 8. We modified the sentence as below and added "Fig. 8" in the sentence:

➔ "The T-box emission is reduced from ~118 Tg yr$^{-1}$ to ~109 Tg yr$^{-1}$, and the N-box source strength increases from ~60 Tg yr$^{-1}$ to ~71 Tg yr$^{-1}$ during the 11.5 – 9.5 ka interval (Fig. 8).

The tropical emission was elevated by ~98 Tg yr$^{-1}$ from the onset of the Holocene to its maximum at 10.6 ka, followed by ~15 Tg yr$^{-1}$ reduction to ~111 Tg yr$^{-1}$ at 9.5 ka. Tropical emission decrease is also observed in IPD-2 from 134 to 115 Tg yr$^{-1}$ during the 11.5-10.0 ka, but this change is not significant in 95% confidence range (Fig. S3 and Table S2)."

P49 L12: The minima at 10.7 ka is only a feature in IPD-1, not in IPD-2. Again, this is misleading, as

IPD-1 is based on much lower temporal resolution. Please use IPD-2 where possible.

➔ We agree with the Reviewer, and we presented a combined IPD using IPD-2 for younger period and IPD-1 for the rest of the studied interval. Please refer to Figure R1 and R2, and relevant responses above.

P49 L16: "…from 29 to 35% during the 11.5 to 10.0 interval." I cannot even see a value <30% or >34%

in Fig. S3, even including the envelope.

➔ We changed the sentence to be more concise:

➔ "Also plotted in Figure 84 is the boreal source fraction, defined as ratio of N-box emission to total source emissions, showing 5% increase (from 31.5 to 36.5%) during the same interval.

The box model results at 9.0, 9.5, and 11.5 ka time slices are summarised in Table 2. It shows a significant increase from ~30% at 11.5 ka to ~35% at 9.5 ka."

| Ref. | N box | T box | Boreal source fraction N/(N+T+S) |
|---|---|---|---|
| (ka) | (Tg yr$^{-1}$) | | (%) |
| Brook et al., 2000 | 64 ± 5 | 123 ± 8 | 32 ± 3 |

| (9.5-11.5 ka) | | | |
|---|---|---|---|
| Chappellaz et al., 1997 (9.5-11.5 ka) | 66 ± 8 | 120 ± 9 | 33 ±3 |
| This study (9.5 – 11.5 ka) (10.8 ka) | 67 ± 3 66 ± 4 65 ± 2 | 118 ± 5 120 ± 4 122 ± 4 | 33 ± 2 33 ± 2 32 ± 1 |
| This study (11.5 ka) | 60 ± 7 47 ± 9 | 118 ± 12 149 ± 10 | 31 ± 4 22 ± 5 |
| This study (9.9 ka)(9.5 ka) (9.8 ka) | 71 ± 3 65 ± 8 74 ± 2 | 109 ± 5 119 ± 9 110 ± 3 | 36 ± 2 33 ± 5 37 ± 1 |
| This study (9.0 ka) | 66 ± 4 | 112 ± 7 | 34 ± 2 |

P49 L7-16: This entire section needs major revision. The numbers don't seem to match the presentation in the figures, the description jumps back and forth between Figures in supplements and main text, the text flow makes it hard to understand.

➔ We removed this section and replaced it with discussion using the combined IPD, following the Reviewer's suggestion above.

P49 L22: What conclusion?

➔ The conclusion in this sentence means the gradual increase of extratropical emission during the earlier part of the studied period. To avoid misleading, we changed the word "This conclusion" into "Our results".

➔ "This conclusion is Our results are supported by proxy-based temperature reconstructions that indicate a gradual warming in northern high latitude northern extratropical regions (30°N – 90°N) until ~9.6 ka, while tropical temperature remains stable (Marcott et al., 2013)."

P49 L30: Your 13 Tg estimate is based on IPD-1, which matches the 8.2 Tg within uncertainties. What value would you get from IPD-2?

➔ We found ~9 Tg from box model results from IPD-2 (Table S1 in previous manuscript version).

P51 L4: Please find a meaningful section title

➔ We removed the entire section.

P51 L27ff: please generalise less and quantify more.

➔ Please refer to our response to Reviewer's comment on Conclusion.

P62, Figure 2: Please add next to axes what these proxies actually show, e.g. warmer, wetter, colder dryer with arrows.

➔ We modified the figure as below:

[Figure]

Figure R4. Revised Figure 2 in manuscript.

P64, Figure 4: Please add reconstructions based on IPD-2

➔ We replaced Figure 4 with new reconstruction based on combined IPD from IPD-1 and IPD-2.

Please refer to Figure R2 in our response above (Fig. 8 in revised main text).

P65, Figure 5: Please synchronize x-axes directions in top panels

➔ We removed the entire section and corresponding figure.

P66 L2: Uncertainties or errors?

➔ Our intension was indicating uncertainty. This was revised.

P67ff: Please include supplements in main text

➔ We partly moved supplements in main text, but we did not include IPD-2, box model of IPD-2, and uncertainty of Styx glacier data set because these information are not essential for discussions of this manuscript.

**Anonymous Reviewer #2: Suggestions for revision or reasons for rejection (will be published if the paper is accepted for final publication)**

The manuscript presents an interesting new set of discrete CH4 measurements from the Siple Dome ice core, allowing the authors to mostly discuss the early Holocene trend of atmospheric methane and to elaborate on the possible mechanisms involved to explain the trend and variability. The analytical work is substantial and I commend the authors for this. However, like the two previous reviewers, I am puzzled by the claimed analytical error of about 2 ppb, when compared with the variability observed when performing blank tests of the system (5 to 15 ppb). The community usually attributes the blank of CH4 analytical systems to degassing of the glass walls of the containers, itself depending on the variable ambiant CH4 concentration in the laboratory and cold room, and on the thermal history of the container. This can introduce a lot of variability, intra-day and inter-day. It is quite surprising that a small variability can be claimed by the authors at the intra-day level. I understand that the authors argue – for a good reason - on the reproducibility of duplicate measurements conducted many days apart on the Siple Dome samples, to claim that their evaluation of the different sources of erros is correct. But it remains quite puzzling from an experimental point of view… I'd suggest for the future evolution of the analytical procedure – and if not done yet – to consider performing again these blank tests while the containers are kept closed in the cold room, under zero air (or nitrogen) filling, before introduction of the ice sample. I'd suspect that this would considerably reduce the inter-day variability of the blanks.

➔ We appreciate the Reviewer #2 for pinpoint comment and thoughtful suggestions. As suggested by the Reviewer #1, we modified the uncertainty estimation of our data. Taking both uncertainties caused by daily systematic offset and the others into account, we address more conservative and comprehensive analytical error. The uncertainty of systematic offset is determined daily by standard error of the mean (SEM) of the results from four bubble-free ice samples, and we used the average value of the daily SEMs for the systematic offset uncertainty ($e1$) calculation. The uncertainty of the other causes ($e2$) is estimated by pooled standard deviation (PSD) of duplicate ice measurements conducted on the same day. The $e2$ includes uncertainty from solubility correction and $CH_4$ inhomogeneity in ice from the same depth interval. Our final data uncertainty for the individual ice is calculated by $((e1)^2 + (e2)^2)^{1/2}$, resulting 3.8 ppb.

A correction for solubility is applied on each data point. Am I wrong or the blank tests are conducted by adding a standard gas to the blank ice ? If this is the case, then the solubility effect should be accounted for - at least partly - through the blank measurements as part of the standard gas gets into diffusive equilibrium with the blank water during the melting phase, and no (or a small) additional correction should apply.

➔ First, it should be noted that we modified the solubility correction method. Now we calculated the solubility effect from residual gas fraction and $CH_4$ mixing ratio of air remained in refrozen meltwater. The 2$^{nd}$ gas extraction tests using Styx glacier ice core, Antarctica show that $CH_4$ mixing ratio in retrapped air is enriched by 3.11 times for glacial ice and 2.98 times for bubble-free ice. Residual gas fraction was measured from SDMA ice samples during the 10 experimental days. The average residual gas fraction is 1.05 ± 0.13% (1σ, n=60) for SDMA ice samples and 0.38 ± 0.08% (1σ, n=40) for bubble-free ice.

➔ The reviewer is right only if the bubble-free ice is a perfect "blank" ice sample that represent all of the physical- and chemical properties of glacial ice except for gas bubbles. However, our bubble-free ice shows different residual gas fraction and hence different solubility effect from glacial ice. As the bubble-free ice does not represent the solubility effect of SDMA ice, we applied the solubility correction to the bubble-free ice as well in the same manner to the SDMA ice, but using different $CH_4$ mixing ratio in retrapped air and residual gas fraction. After corrected for the solubility effect, the offset of $CH_4$ mixing ratio between the bubble-free ice and standard air is caused by system condition change, such as leakage, contaminants, etc. Therefore, we estimate daily systematic offset with the bubble-free ice measurements. This was carried out by injecting standard air on bubble-free ice. After getting raw data, we corrected the results for re-trapped air during melting-refreeze process (solubility correction). The difference in the $CH_4$ level between the original standard air and that from the corrected bubble-free ice was used for the estimation of the daily systematic error.

Figure 1 shows at ~9.6 kyr BP a CH4 spike which may not represent a true atmospheric feature when taking into account the smoothing of atmospheric variations related with gas enclosure conditions at Siple Dome. So there seems to be other sources of errors that the claimed 2 ppb analytical error do not fully cover. Or a good explanation should be brought on why such a narrow spike is observed in the Siple Dome record.

➔ We agreed with the Reviewer and rejected that point. However, this rejection does not change major findings in our manuscript. The figures were revised correspondingly.

The gas enclosure brings me to another concern : the authors make a big case on the interpolar gradient. This is a tricky signal to obtain and to interpret. Notably, gas trapping conditions are key. An ideal case is to combine northern and southern records affected by similar gas trapping conditions. When combining Greenland records with the Siple Dome one, this is clearly not the case. At least the authors should consider convolving the Siple Dome signal with a log-normal distribution reflecting the gas trapping conditions at Siple Dome, before comparing with Greenland records and calculating an IPD. Or they should restrict on only using the WAIS Divide record, despite the fact that it was previously published and not resulting from the authors' work...

➔ The Reviewer is right. The gas age distribution of NEEM is closer to WAIS Divide than Siple Dome condition, thus Siple Dome record should be smoothed more than WAIS Divide. However, our IPD reconstructions were filtered by 1/1000 year$^{-1}$ low-pass window, so that small changes due to different gas enclosure process were smoothed out and do not affect to our finding. The gas trapping conditions become quite important where rapid $CH_4$ changes occur, for example, 8.2 ka cooling event and pre-boreal oscillation (PBO). As commented by the Reviewer #1, we calculated a combined IPD using both Siple Dome and WAIS Divide records.

Aside from the gas enclosure aspect, I wonder indeed if it makes sense in the end to calculate CH4 source strengths changes with a 3-box model while the source evolution is partly attributed to a shift of the ITCZ. The latter necessarily affects the inter-box exchange time as well as the pertinence of the exact latitudinal « boundaries » used between boxes. Isn't there a circular argument here ?

➔ We did not attribute the IPD change and box model results to ITCZ migration in the main text. We discussed only with NH extratropical temperature change and thermokarst lake emissions.

Detail :

I'm not sure that any reference to variable OH in polar regions really make sense. By far most of atmospheric CH4 is oxidized in the inter-tropical band and at relatively high altitude. The polar component does not really matter here.

➔ We removed the sentence that deals with polar winter, and the paragraph was modified as below:

➔ ''The major sink of atmospheric $CH_4$ is photochemical reactions (oxidation) with the hydroxyl radical (OH), which is mainly controlled by atmospheric temperature, humidity, and  the mixing ratio of $CH_4$ itself and non-methane volatile organic compound (NMVOC) (e.g., Levine et al., 2011 and references therein). Air temperature affects humidity thereby limiting the production of OH. Both NMVOCs and $CH_4$  compete for OH , that is, an increase in NMVOC emission reduces the available OH,  and increases the atmospheric lifetime of $CH_4$  (Valdes et al., 2005). Further, since the OH is produced by photo-dissociation reaction, the $CH_4$ sink strength is affected by light availability and tropospheric ozone (e.g., Levy, 1971).  However, recent model studies suggested  that $CH_4$ changes between glacial- and interglacial conditions were driven mostly by source changes, rather than sink changes (Weber et al., 2010; Levine et al., 2011).  "

Apart from the points raised above, I find that the authors correctly addressed all remarks made by the two reviewers as well as the editor.

References that are not cited in the manuscript:

Anderson, T. V.: Efficient, accurate, and non-Gaussian error propagation through nonlinear, closed-
form, analytical system models, Ph. D. thesis, Brigham Young University, Provo, Utah, 78 pp., 2011.

**Atmospheric methane control mechanisms during the early Holocene**

Ji-Woong Yang[1], Jinho Ahn[1], Edward J. Brook[2], and Yeongjun Ryu[1]

[revised manuscript text omitted]
 2 sigma estimated error of SDMA CH$_4$ gas age scale in this study is estimated to be ~147 years. less than ~150 years (see Supplement and see aboveS3). the Siple Dome age uncertainty is likely less than ~100 years (see above and Fig. S1). The climate teleconnection between North Atlantic and tropical hydrology at 10.2 ka might not have been was not sufficiently strong enough to change the low latitude climate. Weak cooling around the North Atlantic region can be a candidate, given that NGRIP $\delta^{18}$O$_{ice}$ records demonstrate smaller amplitude negative anomaly during ~10.2 ka event than those of 8.2 and 9.3 ka., but this The amplitude of $\delta^{18}$O$_{ice}$ changes at 10.2 ka of the high pass filtered GRIP and GISP2 records does show smaller variability than those at 8.2 and 9.3 ka~~

cooling events, but larger than the variability at 10.9 ka (Fig. 5S2). Hence, the "weak cooling" speculation is not fully supported by the other Greenland ice core records. However, this is not supported by other Greenland ice core records such as Greenland Ice Core Project (GRIP) and Greenland Ice Sheet Project 2 (GISP2)., because the high pass filtered GRIP and GISP2 $\delta^{18}O_{ice}$ records show even smaller variability at ~10.9 ka (
[revised manuscript text omitted]

~~Our new CH$_4$ data confirms the abrupt doubling at the Younger Dryas termination. Previous studies using stable isotopes of CH$_4$ have shown contradictory results. Previous studies that, using the stable isotopic composition of C and H in CH$_4$ that aimed to disentangle the cause of abrupt CH$_4$ increase during the earliest period of the Holocene have shown contradictory results. Schaefer et al. (2006) calculated isotopic ($\delta^{13}$C-CH$_4$) mass balance model to discern major source term that caused a slight enrichment in $^{13}$C during the Younger Dryas termination, suggesting tropical wetland emission as a dominant source. The authors also proposed~~

biomass burning, geologic $CH_4$ and enhanced sink process at marine boundary layer as alternatives, but less probable scenarios. On the other hand, Fischer et al. (2008) argued nearly constant biomass burning emission of ~45 Tg yr$^{-1}$ throughout the last glacial termination with a slight increase in PB, and also showed that the boreal sources were expanded during the YD PB transition. However, Möller et al. (2013) pointed out the possibility of changing isotopic signature of each source itself, and they found that less pyrogenic emission is required for LGM condition if they increased the $\delta^{13}C$ $CH_4$ signatures of tropical wetland and of biomass burning. 
[revised manuscript text omitted]

__________________________

**Supplementary text**

**Reproducibility test from Styx Glacier ice core**

Here we present results of reproducibility test by using different ice core samples in the same manner as used for Siple Dome ices, to demonstrate the reliability of our analytical system. The ice core was drilled at Styx glacier (73° 51.10'S, 163° 41.22'E, 1623 m a.s.l) in 2014-2015 austral summer, and mean snow accumulation rate was estimated as 0.13 Mg m$^{-2}$ yr$^{-1}$ (Han et al., 2015). The replicate measurements were carried out at randomly chosen 7 depths with time interval of 51 to 226 days. Depth difference between the replicate pairs is less than 10 cm. Results show the mean absolute difference between the original  and replicate measurements of 1.9, which is as good as the Siple Dome results. The daily blank offset (see the main text) ranges from ~6 to 19 ppb, and the intra day blank offset is 2.4 ppb (standard error of the mean, n = 4). Again, these results reveal the robustness of our blank correction methods.

**Maximum SDMA gas age unceratinty**

In this paper we used a modified gas age scale from the previous one based on $\delta^{15}N$ measurements (Severinghaus et al., 2009) by setting 3 age tie-points and interpolating the age offset between the tie-points (Fig. S1). To estimate gas age uncertainty, we compared the SDMA modified gas age (this study) to the new gas age determined by $CH_4$-correlation with NEEM discrete $CH_4$ data measured at OSU (Chappellaz et al., 2013). Figure S3 shows the offset between the two age scales, which it should be moved to adjust the NEEM $CH_4$ in GICC05modelext NEEM 1 age. In addition, we take into account the maximum layer counting uncertainty of 99 years (Rasmussen et al., 2006) and delta age uncertainty of 30 years (Rasmussen et al., 2013) during the early Holocene. Error progagation gives us the maximum uncertainty of the early Holocene SDMA gas age of ~147.4 years.

[Figure]

Figure S1. Upper: Comparison between Siple Dome CH₄ anomalies plotted with gas age adjusted to GICC05 (red, solid) and previous gas age (red, dashed; Brook et al., 2005). Lower: NGRIP δ¹⁸O anomaly in GICC05 scale. The horizontal error bars denote the age uncertainty of GICC05 chronology (Rasmussen et al., 2006), and the black triangles are age tie points used to adjusting the Siple Dome age scale to GICC05 scale.

[Figure]

Figure S1. Comparison of Greenland oxygen isotope ratios from NGRIP (blue, Rasmussen et al., 2006), GRIP (grey,
Rasmussen et al., 2006) and GISP2 (red, Stuiver and Grootes, 2000). All time series were high-pass filtered with
1/1800-year window. Note that the cooling amplitude at 10.3 ka is smaller than 8.2 and 9.3 ka events in NGRIP
records, but this is not clear in GRIP and GISP2 ice cores.

[Figure]

[Figure]

Figure S4. Age difference between the new gas age scale adjusted to GICC05 by Monte Carlo matching with NEEM
discrete $CH_4$ (Chappellaz et al., 2013) and the original gas age based on $\delta^{15}N$ records (Severinghaus et al., 2009).

Table S1. 3-box source distribution model results of tropical (green, T) and boreal (red, N) boxes and boreal source
fraction obtained from IPD-2. Errors denote 95% confidence interval.

| Ref. | N box | T box | Boreal source fraction |
|---|---|---|---|
| (ka) | (Tg yr⁻¹) | | (%) |
| This study (11.5 ka) | 60 ± 7 | 134 ± 16 | 29 ± 4 |
| This study (10.0 ka) | 71 ± 7 | 115 ± 11 | 35 ± 4 |

---

## Author Response (AR3)

Dear Editor,

Thank you for careful reading and positive evaluation of our revised manuscript. Below we addressed how we revised our manuscript. We hope this could help our updated manuscript to be published.

**Editor Decision: Publish subject to minor revisions (review by Editor)** (27 Jul

2017) by Eric Wolff

Comments to the Author:

Thank you again for your constructive response to the reviewers. Although I think your method of calculating the uncertainty remains open to question, I think (based also on the data in Table 1) the value you are using is probably reasonable, and that there is no value in pursuing this further. I am therefore not sending the paper back for further review but instead accepting it subject to minor changes.

I have a couple of points about the English (copy editors may have more):

Page 3, line 27, remove "Since" so that the snetence makes sense

Line 36, replace "Despite" with "Although"

➔ The sentences were modified as below:

➔ There is no high-resolution reconstruction of past population and land use area, and consequently large uncertainties of $CH_4$ emission from land use change impede identification of any shorter scale changes.

➔ Although Rhodes et al. (2015) reported a very high-resolution record from WAIS Divide ice core that extends from the last glacial period to the earliest Holocene (~9.8 ka), the authors do not deal with the early Holocene $CH_4$ variability.

Page 10, line 39. This should surely read "Fig 7" not "Fig 6"

➔ The sentences were modified as below:

➔ Given the IPD-2 is better constrained than IPD-1, we use IPD-2 curve from 9.9 to 11.5 ka interval and IPD-

1 for the rest of the studied period (Fig. 7).

You have moved much material from SI into main text but you left duplicate figures in SI.

This is not needed, please remove them: Fig S1 is Fig 3; Fig S2 is Fig 5; the material in

Fig S3 is in Fig 8; and Fig S4 is Fig 4.

➔ Regarding the Supplement, we did not uploaded any material for SI, but it seems that the SI of the old version of this manuscript still remains. We asked this to

Copernicus Editorial Support, and they replied us that we cannot delete the uploaded
SI, but if we do not submit a new one, they will not use the old one.

Table 1: It would be useful to add a column showing the difference between the mean of
the 1st measurement and the mean of the 2nd measurement, as this is the statistic that
indicates that your error estimate might be OK.

➔ We added a column in Table 1 showing the difference between the means of
duplicate samples from the 1st and 2nd measurements The Table 1 was changes as
below:

[revised manuscript text omitted]